# ClinicalBench: Can LLMs Beat Traditional ML Models in Clinical Prediction?

## Abstract

Large Language Models (LLMs) hold great promise to revolutionize current clinical systems for their superior capacities on medical text processing tasks and medical licensing exams. Meanwhile, traditional ML models such as SVM and XGBoost have still been mainly adopted in clinical prediction tasks. An emerging question is *Can LLMs beat traditional ML models in clinical prediction?* Thus, we build a new benchmark ClinicalBench to comprehensively study the clinical predictive modeling capacities of both general-purpose and medical LLMs, and compare them with traditional ML models. ClinicalBench embraces three common clinical prediction tasks, two databases, 14 general-purpose LLMs, 8 medical LLMs, and 11 traditional ML models. Through extensive empirical investigation, we discover that **both general-purpose and medical LLMs, even with different model scales, diverse prompting or fine-tuning strategies, still cannot beat traditional ML models in clinical prediction yet**, shedding light on their surprising but critical deficiency in clinical reasoning. We call for caution when practitioners adopt LLMs in clinical applications. ClinicalBench can be utilized to bridge the gap between LLMs' development for healthcare and real-world clinical practice. Code is here.

## 1 Introduction

Large Language Models (LLMs) have shown great potential to revolutionize existing clinical systems for their superior capacities on a variety of medical text processing tasks including document classification, report generation and summarization, and information extraction (Jahan et al., 2024; Chen et al., 2023a; Zhou et al., 2023). Also, LLMs could even rival human clinician performance in medical licensing exams (Singhal et al., 2023a;b; Gilson et al., 2023) and clinical case challenges (Kanjee et al., 2023; McDuff et al., 2023). However, traditional machine learning (ML) models such as Logistic Regression, SVM, XGBoost, MLP and Transformer have still been predominantly adopted for clinical prediction tasks in real-world practice (van de Water et al., 2024; Moazemi et al., 2023), which play an essential role in modern health systems to improve patient outcomes and support clinical decision making. Thus, considering LLMs' profound world knowledge and strong reasoning capacities, an emerging question is: ***Can LLMs beat traditional ML models in clinical prediction?***

In this paper, we propose to build a new benchmark ClinicalBench to comprehensively and systematically investigate the capacities of LLMs in clinical prediction tasks. More importantly, we compare the performance of LLMs with diverse traditional ML models in a head-to-head way to explore the feasibility of adopting LLMs in real-world clinical prediction. Specifically, following previous works (Wang et al., 2020; van de Water et al., 2024), we adopt three common clinical prediction tasks including Length-of-Stay Prediction, Mortality Prediction and Readmission Prediction and two real clinical databases MIMIC-III (Johnson et al., 2016) and MIMIC-IV (Johnson et al., 2020). As for the benchmarked methods, ClinicalBench embraces 11 traditional ML models and 22 LLMs with different scales, which include 14 *general-purpose LLMs* and 8 *medical LLMs*.

In ClinicalBench, we aim to answer three research questions through extensive empirical studies. The first question is *Can directly prompting LLMs beat traditional ML models?* As shown in Figure 1, to effectively leverage the language understanding capacities of LLMs, we first convert the original clinical codes in datasets to texts and then let LLMs output the predicted label directly. Comparing the performance of both 9 general-purpose LLMs and 6 medical LLMs with around 7B parameters to traditional ML models, we discover that **traditional ML models can mostly outperform LLMs with a regular scale by a large margin across three tasks and two datasets**. Comparing general-purpose

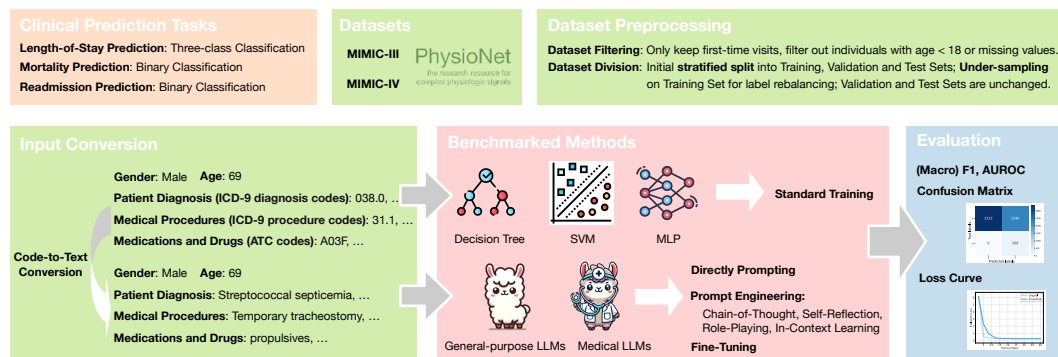

Figure 1: **Overview of** ClinicalBench.

and medical LLMs, we also find that the **medical LLMs do not noticeably surpass general-purpose LLMs with a similar scale in clinical prediction**. We further studied the impact of *decoding temperature* and *model size* on LLMs' predictive modeling capacities. We find that these factors could potentially influence the clinical prediction performance in a negative or positive way for different LLMs. However, **adjusting temperatures or scaling parameters of LLMs still may not reach the performance of typical traditional ML models such as XGBoost, SVM, and RNN**.

The second question is *Can LLMs with prompt engineering beat traditional ML models?* We studied the efficacy of four common prompting strategies including Zero-shot Chain-of-Thought (Kojima et al., 2022), Self-Reflection (Pan et al., 2023), Role-Playing (Chen et al., 2024b), and In-Context Learning (Dong et al., 2023) on both general-purpose LLMs and medical LLMs. In general, we find that **the effectiveness of different prompting strategies is very limited**. Except the improvement brought by In-Context Learning on Llama3-8B, Gemma2-9B, and Internist-7B for Length-of-Stay Prediction, the aforementioned prompting strategies do not enhance the performance of LLMs in clinical prediction across three tasks and two datasets in a noticeable way.

The third question is *Can fine-tuned LLMs beat traditional ML models?* We compare the performance of traditional ML models and LLMs with the same dataset split on training, validation and test sets. With thorough investigation over 2 fine-tuning strategies, 4 LLMs, 3 clinical prediction tasks and 2 datasets, we find that **fine-tuning is clearly effective for certain tasks though the effectiveness varies across different models**. As for Length-of-Stay Prediction and Mortality Prediction, we observe that fine-tuning can evidently enhance the clinical prediction performance of LLMs though the extent of improvement depends on models. As for Readmission Prediction, we do not notice fine-tuning can bring any enhancement. However, **most fine-tuned LLMs still cannot surpass typical traditional ML models such as XGBoost, SVM, Transformer and RNN**.

Overall, the contributions of this paper can be summarized as follows:

- We built a new benchmark ClinicalBench, embracing 14 general-purpose LLMs, 8 medical LLMs, 11 traditional ML models, three tasks and two databases, which made the first attempt to compare the clinical prediction capacities of LLMs and traditional ML models in a head-to-head way.

- We discover that both general-purpose and medical LLMs, even with different model sizes, prompting or fine-tuning strategies, still cannot beat traditional ML models in clinical prediction yet.

- Our findings demonstrate **the surprising but critical deficiency of both general-purpose and medical LLMs in real-world clinical reasoning and decision-making**, which could have almost clinician-level performance in medical licensing exams and clinical case challenges. We call for caution when adopting LLMs in practical clinical applications. ClinicalBench could be leveraged to bridge the gap between the development of LLMs for healthcare and real-world clinical practice.

## 2 ClinicalBench: Benchmarking LLMs and Traditional ML Models in Clinical Prediction

### 2.1 Clinical Prediction Tasks

In ClinicalBench, following previous works (Wang et al., 2020; van de Water et al., 2024), we focus on three common clinical prediction tasks including *Length-of-Stay Prediction*, *Mortality Prediction*,

and *Readmission Prediction*. Specifically, *Length-of-Stay Prediction* is simplified as a three-class classification task aiming to predict the length of the current hospital visit as less than one week, 1 ~ 2 weeks, or more than two weeks. *Mortality Prediction* is a binary classification task which intends to estimate whether the patient will decease in the current visit. *Readmission Prediction* is a binary classification task designed to identify patients who are at high risk of being readmitted to the hospital within a specific time frame. Following PyHealth (Yang et al., 2023), each task requires models to make predictions based on patients' demographic features (*e.g.*, gender and age) and clinical information including diagnosis, medical procedures, medications and drugs for the current visit.

## 2.2 DATASET PROCESSING

We adopt MIMIC-III-v1.4 (Johnson et al., 2016) and MIMIC-IV-v2.2 (Johnson et al., 2020) in ClinicalBench, which are two large and freely accessible databases and widely adopted in the fields of healthcare data science. They contain de-identified data associated with ICU admissions at the Beth Israel Deaconess Medical Center during 2001-2012 and 2008-2019 respectively.

**Dataset Filtering**  We generally follow the filtering criteria in previous literature (Yang et al., 2023; Purushotham et al., 2018; Wang et al., 2020; Tang et al., 2020). First, we only keep the samples of the first-time visits for each patient since other visits require the consideration of patients' history, which desire different prediction strategies compared with first-time visits. Second, we filter out patients who are younger than 18 due to the substantial differences between pediatric and adult physiology. Third, we also did not consider patients with missing values following the literature.

**Dataset Division**  For traditional ML models, we first adopt **stratified split** to divide the original dataset into *training*, *validation* and *test* sets. Due to the label imbalance of *training* set for three tasks, we apply **under-sampling** to rebalance the training set but leave the *validation* and *test* sets unchanged, which can reflect the real-world clinical prediction performance of different models. For prompting-based methods, we directly assess the performance of LLMs on the same *test* set. For fine-tuning based methods, the division is the same as traditional ML models for fair comparison.

**Code-to-Text Conversion**  It is worth noting that the information of patient diagnosis, medical procedures, medications and drugs in original MIMIC-III (MIMIC-IV) dataset is stored in the form of ICD-9 (ICD-10) diagnosis codes, ICD-9 (ICD-10) procedure codes and ATC codes respectively. To effectively leverage the natural language understanding and reasoning capacities of LLMs, as shown in Figure 1, we first convert the codes into texts and then design the specific prompts for LLMs.

## 2.3 BENCHMARKED METHODS

We compare mainstream traditional ML models and LLMs in ClinicalBench. The traditional ML models embrace XGBoost, Logistic Regression, Decision Tree, Random Forest, AdaBoost, SVM, Naive Bayes and neural network models such as MLP, Transformer and RNN. For LLMs, we first investigated directly prompting both general-purpose and medical LLMs with different decoding temperatures and model scales. Then, we explored the effectiveness of diverse prompting and fine-tuning strategies. The examples of specific prompts for the different strategies are in Appendix D. It is worth noting that we only evaluate open-source LLMs in ClinicalBench since closed-source models such as GPT-4 are prohibited for MIMIC-III and MIMIC-IV according to their data use policy [1].

## 3 CAN DIRECTLY PROMPTING LLMS BEAT TRADITIONAL ML MODELS?

In this section, we aim to investigate the question *Can directly prompting LLMs beat traditional ML models?* We first compare the performance of 9 general-purpose and 6 medical LLMs with around 7B parameter scale to a variety of traditional ML models. Then, we further explore the impact of decoding temperatures and parameter scaling on LLMs' clinical prediction performance.

**Main Results**  As shown in Table 1, red and orange represent the performance regarding (Macro) F1% and AUROC% respectively. We can clearly observe that the shades of both red and orange for traditional ML models are darker than those for LLMs. It shows that **traditional ML models generally outperform both general-purpose and medical LLMs with a regular scale in clinical prediction**. Furthermore, we have calculated the ranges of performance with 95% Confidence Interval

---

[1]https://physionet.org/about/licenses/physionet-credentialed-health-data-license-150

| Method | Length-of-Stay Prediction | | Mortality Prediction | | Readmission Prediction | |
|---|---|---|---|---|---|---|
| | Macro F1 (%) 95% CI | AUROC (%) 95% CI | F1 (%) 95% CI | AUROC (%) 95% CI | F1 (%) 95% CI | AUROC (%) 95% CI |
| **Majority** | 23.37 (23.37, 23.37) | 50.00 (50.00, 50.00) | 0.00 (0.00, 0.00) | 50.00 (50.00, 50.00) | 0.00 (0.00, 0.00) | 50.00 (50.00, 50.00) |
| **Minority** | 10.72 (10.72, 10.72) | 50.00 (50.00, 50.00) | 20.88 (20.88, 20.88) | 50.00 (50.00, 50.00) | 13.64 (13.64, 13.64) | 50.00 (50.00, 50.00) |
| *Traditional ML Models* | | | | | | |
| **XGBoost** | 68.05 (67.99, 68.11) | 93.78 (93.72, 93.84) | 64.25 (64.09, 64.41) | 95.59 (95.55, 95.63) | 19.21 (18.98, 19.44) | 68.75 (68.18, 69.32) |
| **LR** | 66.56 (66.46, 66.66) | 93.09 (92.99, 93.19) | 62.57 (62.39, 62.75) | 94.31 (94.24, 94.38) | 19.09 (18.89, 19.29) | 68.24 (67.86, 68.62) |
| **DecisionTree** | 59.17 (59.07, 59.27) | 85.19 (85.11, 85.27) | 51.05 (50.92, 51.18) | 84.21 (84.0, 84.42) | 16.46 (16.21, 16.71) | 58.0 (57.37, 58.63) |
| **RandomForest** | 61.59 (61.55, 61.63) | 91.25 (91.18, 91.32) | 53.40 (53.21, 53.59) | 91.01 (90.92, 91.10) | 18.93 (18.78, 19.08) | 68.91 (68.58, 69.24) |
| **AdaBoost** | 66.72 (66.62, 66.82) | 92.90 (92.83, 92.97) | 60.55 (60.40, 60.70) | 93.25 (93.18, 93.32) | 17.30 (17.06, 17.54) | 62.50 (62.06, 62.94) |
| **SVM** | 70.25 (70.16, 70.34) | 94.78 (94.72, 94.84) | 63.34 (63.22, 63.46) | 95.45 (95.40, 95.50) | 19.47 (19.34, 19.60) | 70.57 (70.13, 71.01) |
| **NaiveBayes** | 42.93 (42.84, 43.02) | 61.55 (61.43, 61.67) | 23.43 (23.41, 23.45) | 56.85 (56.80, 56.90) | 16.86 (16.77, 16.95) | 58.76 (58.56, 58.96) |
| **KNN** | 44.06 (43.98, 44.14) | 74.15 (74.06, 74.24) | 45.04 (44.71, 45.37) | 85.51 (85.35, 85.67) | 15.39 (15.22, 15.56) | 58.81 (58.23, 59.39) |
| **MLP** | 63.16 (63.08, 63.24) | 89.82 (89.74, 89.90) | 57.82 (57.65, 57.99) | 92.41 (92.30, 92.52) | 18.39 (18.28, 18.50) | 66.87 (66.63, 67.11) |
| **Transformer** | 64.48 (64.37, 64.59) | 91.45 (91.36, 91.54) | 56.32 (56.03, 56.61) | 93.34 (93.29, 93.39) | 18.85 (18.74, 18.96) | 67.07 (66.82, 67.32) |
| **RNN** | 67.41 (67.29, 67.53) | 92.86 (92.81, 92.91) | 61.15 (60.84, 61.46) | 92.97 (92.90, 93.04) | 20.94 (20.80, 21.08) | 72.07 (71.82, 72.32) |
| *General-purpose Large Language Models* | | | | | | |
| **Llama3-8B** | 25.78 (25.72, 25.84) | 83.82 (83.74, 83.90) | 25.81 (25.78, 25.84) | 85.40 (85.33, 85.47) | 13.64 (13.64, 13.64) | 43.00 (42.50, 43.50) |
| **Mistral-v0.3-7B** | 11.59 (11.57, 11.61) | 69.73 (69.64, 69.82) | 38.11 (38.06, 38.16) | 86.84 (86.80, 86.88) | 13.44 (13.38, 13.50) | 47.16 (46.69, 47.63) |
| **Gemma2-9B** | 17.30 (17.26, 17.34) | 79.19 (79.08, 79.30) | 43.03 (42.96, 43.10) | 86.46 (86.41, 86.51) | 13.66 (13.66, 13.66) | 47.14 (46.87, 47.41) |
| **Qwen2-7B** | 12.17 (12.14, 12.20) | 73.97 (73.82, 74.12) | 34.98 (34.81, 35.15) | 85.97 (85.91, 86.03) | 2.14 (1.88, 2.40) | 44.04 (43.61, 44.47) |
| **Yi-v1.5-9B** | 10.89 (10.88, 10.90) | 74.51 (74.41, 74.61) | 28.75 (28.72, 28.78) | 76.05 (75.96, 76.14) | 11.35 (11.23, 11.47) | 45.02 (44.76, 45.28) |
| **Vicuna-v1.5-7B** | 22.05 (22.01, 22.09) | 73.20 (73.11, 73.29) | 20.91 (20.91, 20.91) | 77.90 (77.80, 78.00) | 13.64 (13.64, 13.64) | 39.51 (39.18, 39.84) |
| **Phi3.5-mini-3.8B** | 13.17 (13.14, 13.20) | 73.67 (73.63, 73.71) | 26.45 (26.43, 26.47) | 80.37 (80.30, 80.44) | 12.40 (12.30, 12.50) | 46.64 (46.37, 46.91) |
| **InternLM2.5-7B** | 11.53 (11.52, 11.54) | 82.38 (82.28, 82.48) | 33.89 (33.57, 34.21) | 83.89 (83.81, 83.97) | 0.00 (0.00, 0.00) | 45.84 (45.51, 46.17) |
| **MiniCPM3-4B** | 22.25 (22.21, 22.29) | 75.95 (75.86, 76.04) | 24.01 (24.00, 24.02) | 86.28 (86.21, 86.35) | 13.36 (13.31, 13.41) | 42.23 (41.95, 42.51) |
| *Medical Large Language Models* | | | | | | |
| **Meditron-7B** | 33.43 (33.39, 33.47) | 76.63 (76.51, 76.75) | 20.88 (20.88, 20.88) | 77.26 (77.20, 77.32) | 13.64 (13.64, 13.64) | 44.93 (44.43, 45.43) |
| **Medllama3-8B** | 14.88 (14.85, 14.91) | 78.27 (78.22, 78.32) | 24.12 (24.11, 24.13) | 78.74 (78.66, 78.82) | 13.64 (13.64, 13.64) | 49.80 (49.24, 50.36) |
| **BioMistral-7B** | 21.96 (21.89, 22.03) | 40.94 (40.73, 41.15) | 30.53 (30.45, 30.61) | 77.88 (77.81, 77.95) | 13.37 (13.28, 13.46) | 40.93 (40.54, 41.32) |
| **Med42-8B** | 12.82 (12.80, 12.84) | 84.76 (84.70, 84.82) | 28.02 (27.98, 28.06) | 82.55 (82.46, 82.64) | 13.62 (13.60, 13.64) | 42.95 (42.56, 43.34) |
| **BioMedGPT-7B** | 14.11 (14.11, 14.11) | 61.54 (61.45, 61.63) | 22.55 (22.54, 22.56) | 78.93 (78.82, 79.04) | 13.72 (13.65, 13.79) | 42.89 (42.67, 43.11) |
| **Internist-7B** | 11.71 (11.68, 11.74) | 81.10 (81.04, 81.16) | 37.97 (37.92, 38.02) | 87.25 (87.20, 87.30) | 12.07 (11.90, 12.24) | 47.67 (47.36, 47.98) |

Table 1: **Performance of LLMs and Traditional ML Models on Typical Clinical Prediction Tasks**, which include Length-of-Stay Prediction, Mortality Prediction, and Readmission Prediction. Experiments are conducted in **MIMIC-III** dataset. More experiment results on **MIMIC-IV** dataset are in **Appendix B.1**. More experiment results of traditional ML models on different scales of training set are in **Appendix B.2**. (Macro) F1% and AUROC% are adopted as metrics considering label imbalance. Average performance over five runs is reported. The darker shades of red and orange indicate that the scores are closer to 100%. The numbers in bracket are the ranges of performance with **95% Confidence Interval (95% CI)**, which are calculated with the 5-run scores.

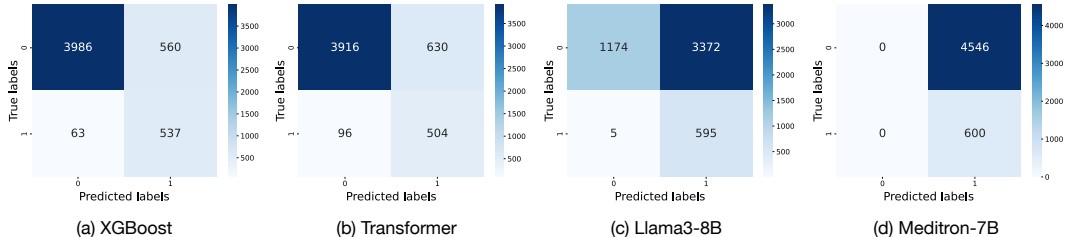

(a) XGBoost     (b) Transformer     (c) Llama3-8B     (d) Meditron-7B

Figure 2: **Examples of Confusion Matrix of LLMs and Traditional ML Models on Mortality Prediction**. Experiments are conducted in MIMIC-III dataset. The complete confusion matrices across different methods, tasks and datasets are in **Appendix B.4**.

through the five-run results, which are shown as the numbers in bracket in Table 1. The lower bounds of traditional ML models' ranges are still mostly larger than the higher bounds of LLMs' ranges, indicating the **high statistical significance of the advantage of traditional ML models over LLMs**. In addition, we have evaluated the performance of traditional ML models with 5%, 10%, 20%, 40% of the original training set in Appendix B.2 and their performance is relatively stable and still surpasses LLMs, which shows **the advantage of traditional ML models in low-resource scenarios**.

Also, we notice that there is generally no notable difference between the performances of general-purpose and medical LLMs, and some medical LLMs can even underperform their general-purpose counterparts. For example, Med42-8B (Christophe et al., 2024) is adapted from Llama3-8B through fine-tuning on medical corpus and can outperform Llama3-8B in various medical knowledge benchmarks such as MedQA (Jin et al., 2021) and MedMCQA (Pal et al., 2022) (The model download links are in Appendix A). However, there is a clear drop for Length-of-Stay Prediction and no substantial difference for the other two tasks when comparing Med42-8B to Llama3-8B regarding their clinical prediction performance, which illustrates that **adapting general-purpose LLMs to medical domain may not improve and could even hurt their clinical reasoning capacities**.

It is noteworthy that the performance of some LLMs (*e.g.*, Llama3-8B and Meditron-7B for Mortality Prediction) is even comparable to that of "Majority" or "Minority" method, which refers to selecting the "Majority" or "Minority" class as the predicted label directly, which further shows **LLMs' deficiency in some clinical prediction tasks**. In more detail, we also explore the confusion matrices of the predictions of both traditional ML models and LLMs. As shown in Figure 2, we can see that **the prediction patterns of LLMs could be distinct from those of traditional ML models**. With under-sampling on the training set, traditional ML models can generally balance precision and recall. However, the predictions of LLMs could be overly biased. For example, Llama3-8B and Meditron-7B have a high rate of false positives, which results in the poor performance on precision and F1.

**The Impact of Decoding Temperature** Since decoding temperature is a key hyperparameter and could have a considerable impact on LLMs' reasoning capacities (Qiu et al., 2024; Renze & Guven, 2024), we explore whether adjusting temperatures can enhance LLMs' clinical prediction performance. As shown in Figure 3, when the tem-

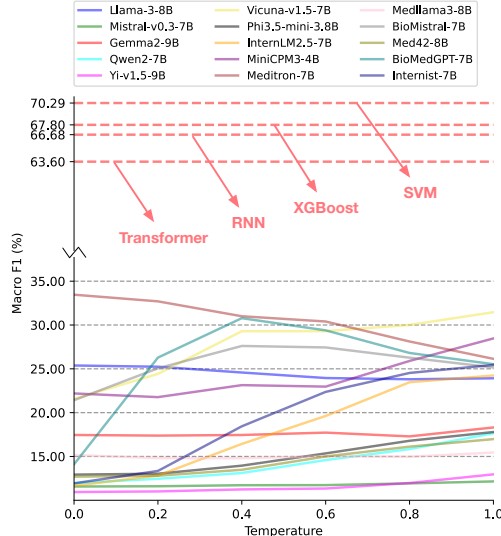

Figure 3: **Performance Comparison Between LLMs with Different Temperatures and Traditional ML Models on Length-of-Stay Prediction.** Experiments are conducted in MIMIC-III dataset. More results on Mortality and Readmission Prediction are in **Appenidix B.3**.

perature increases, **the impact on LLMs' clinical prediction performance could be positive or negative for different LLMs**. For example, when the temperature is higher, the performance of Internist-7B constantly increases but that of Meditron-7B keeps decreasing. However, **only adjusting the decoding temperature of LLMs cannot reach the performance of typical traditional ML models such as XGBoost, SVM, RNN and Transformer**.

| Method | Length-of-Stay Prediction | | Mortality Prediction | | Readmission Prediction | |
|---|---|---|---|---|---|---|
| | **MIMIC-III** | **MIMIC-IV** | **MIMIC-III** | **MIMIC-IV** | **MIMIC-III** | **MIMIC-IV** |
| *Traditional ML Models* | | | | | | |
| **XGBoost** | 67.80 | 64.29 | 63.29 | 45.84 | 19.54 | 27.15 |
| **SVM** | 70.29 | 66.54 | 63.06 | 46.93 | 19.13 | 28.65 |
| **Transformer** | 63.60 | 56.65 | 58.13 | 41.05 | 17.98 | 26.17 |
| **RNN** | 66.68 | 62.17 | 62.43 | 48.79 | 20.79 | 27.12 |
| *General-purpose Large Language Models* | | | | | | |
| **Llama3-8B** | 25.38 | 17.73 | 26.06 | 15.12 | 13.64 | 25.05 |
| **Llama3-70B** | 18.66 | 23.01 | 30.33 | 18.43 | 12.40 | 22.81 |
| **Qwen2-0.5B** | 5.11 | 5.78 | 1.94 | 3.11 | 0.00 | 0.60 |
| **Qwen2-1.5B** | 23.37 | 29.55 | 20.91 | 6.88 | 13.72 | 24.82 |
| **Qwen2-7B** | 11.96 | 6.48 | 35.65 | 43.53 | 4.26 | 0.29 |
| **Yi-v1.5-6B** | 25.81 | 30.18 | 28.35 | 18.26 | 13.69 | 24.66 |
| **Yi-v1.5-9B** | 10.94 | 6.96 | 28.82 | 16.55 | 10.74 | 20.73 |
| **Yi-v1.5-34B** | 18.92 | 18.42 | 41.78 | 33.63 | 13.46 | 24.12 |
| *Medical Large Language Models* | | | | | | |
| **Meditron-7B** | 33.46 | 26.90 | 20.88 | 6.70 | 13.64 | 24.92 |
| **Meditron-70B** | 30.15 | 35.67 | 43.16 | 47.93 | 8.67 | 9.12 |
| **Med42-8B** | 12.69 | 9.97 | 28.59 | 16.79 | 13.59 | 25.06 |
| **Med42-70B** | 15.43 | 22.95 | 42.96 | 31.21 | 12.99 | 23.87 |

Table 2: **Performance Comparison Between LLMs with Different Scales and Traditional ML Models on Clinical Prediction Tasks**. Experiments are conducted in **MIMIC-III** and **MIMIC-IV** datasets. Considering label imbalance, Length-of-Stay Prediction adopts **Macro F1**% and the other two tasks use **F1**% as the metric. The darker shades of red indicate that the scores are closer to 100%.

**The Impact of Parameter Scaling in LLMs**   LLMs with more parameters in the same model series generally perform better across different tasks owing to deeper world knowledge and stronger reasoning capacities. For example, the performance on diverse datasets such as MMLU (Hendrycks et al., 2021) and AGIEval (Zhong et al., 2023) shows a steady growth as scales expand for Yi series models (Young et al., 2024). Thus, after evaluating the performance of LLMs with a around 7B parameter size in Table 1, we explored the impact of parameter scaling on LLMs' clinical prediction performance and investigated whether LLMs with a larger scale could beat traditional ML models.

As shown in Table 2, we have assessed three general-purpose model series including Llama3 (Dubey et al., 2024), Qwen2 (Yang et al., 2024), and Yi-v1.5 (Young et al., 2024) and two medical model series including Meditron (Chen et al., 2023b) and Med42 (Christophe et al., 2024). Comparing LLMs with different scales in the same model series, we can observe that **the parameter scaling does not necessarily lead to better clinical prediction performance**. In Mortality Prediction, LLMs with a larger parameter scale in the same model series tend to perform better. For example, the performance of Qwen2 series models consistently increases on MIMIC-III and MIMIC-IV datasets as the parameter scales grow. However, this tendency does not appear in Length-of-Stay Prediction and Readmission Prediction. Qwen2-1.5B outperforms Qwen2-0.5B and Qwen2-7B for Length-of-Stay Prediction and Readmission Prediction. We also notice that **parameter scaling could even hurt the clinical prediction performance**. For example, Yi-v1.5-6B performs much better than Yi-v1.5-9B and Yi-v1.5-34B for Length-of-Stay Prediction on both MIMIC-III and MIMIC-IV datasets.

However, through the shades of red color, we can clearly see that even though the performance of some LLMs is improved for certain tasks **as the parameter scales expand, they still underperform typical traditional ML models such as XGBoost, SVM, RNN and Transformer**. For example, although the performance is substantially enhanced for Meditron-70B compared to Meditron-7B in Mortality Prediction, there is still a large gap from traditional ML models.

> **Finding 1:** Directly prompting general-purpose and medical LLMs, even with different decoding temperatures or parameter scales, cannot beat traditional ML models in clinical prediction yet.

| Method | Length-of-Stay Prediction | | Mortality Prediction | | Readmission Prediction | |
|---|---|---|---|---|---|---|
| | **MIMIC-III** | **MIMIC-IV** | **MIMIC-III** | **MIMIC-IV** | **MIMIC-III** | **MIMIC-IV** |
| *Traditional ML Models* | | | | | | |
| **XGBoost** | 64.20 | 57.60 | 46.89 | 41.10 | 21.19 | 27.52 |
| **SVM** | 67.19 | 61.82 | 53.85 | 40.00 | 19.61 | 28.00 |
| **Transformer** | 60.25 | 55.38 | 53.00 | 29.36 | 17.24 | 23.86 |
| **RNN** | 65.19 | 58.07 | 53.71 | 24.29 | 19.83 | 23.90 |
| *General-purpose Large Language Models* | | | | | | |
| **Llama3-8B** | 26.36 | 16.14 | 25.84 | 14.35 | 13.78 | 25.40 |
| **+ CoT** | 14.76 | 9.22 | 23.18 | 10.81 | 12.05 | 22.90 |
| **+ Self-Reflection** | 25.88 | 17.53 | 24.10 | 14.29 | 15.70 | 22.80 |
| **+ Role-Playing** | 27.44 | 17.30 | 25.44 | 14.11 | 13.78 | 24.87 |
| **+ ICL** | 51.77 | 54.72 | 28.02 | 14.85 | 11.97 | 24.46 |
| **Gemma2-9B** | 16.22 | 21.66 | 44.78 | 32.99 | 13.78 | 23.32 |
| **+ CoT** | 19.34 | 17.65 | 22.35 | 9.94 | 13.78 | 24.38 |
| **+ Self-Reflection** | 16.67 | 17.89 | 0.00 | 0.00 | 13.96 | 24.07 |
| **+ Role-Playing** | 16.29 | 20.24 | 44.19 | 41.03 | 13.83 | 23.74 |
| **+ ICL** | 44.13 | 46.29 | 23.92 | 12.45 | 14.68 | 23.95 |
| *Medical Large Language Models* | | | | | | |
| **Meditron-7B** | 35.88 | 27.53 | 20.79 | 6.58 | 13.78 | 24.87 |
| **+ CoT** | 16.04 | 12.58 | 13.90 | 5.95 | 12.83 | 24.34 |
| **+ Self-Reflection** | 18.36 | 10.40 | 20.47 | 6.60 | 13.78 | 24.91 |
| **+ Role-Playing** | 27.73 | 17.54 | 20.79 | 6.58 | 13.78 | 24.87 |
| **+ ICL** | 14.09 | 14.62 | 23.72 | 11.37 | 11.64 | 23.27 |
| **Internist-7B** | 11.00 | 14.14 | 38.57 | 29.06 | 11.54 | 17.80 |
| **+ CoT** | 22.86 | 17.03 | 21.67 | 8.86 | 14.26 | 25.48 |
| **+ Self-Reflection** | 23.57 | 15.09 | 21.23 | 10.49 | 12.16 | 21.36 |
| **+ Role-Playing** | 10.87 | 12.61 | 25.44 | 15.38 | 14.42 | 23.08 |
| **+ ICL** | 47.08 | 34.33 | 31.15 | 16.59 | 11.59 | 22.86 |

Table 3: **Performance Comparison Between LLMs with Prompt Engineering and Traditional ML Models on Clinical Prediction Tasks**. Experiments are conducted in sampled **MIMIC-III** and **MIMIC-IV** datasets. Length-of-Stay Prediction adopts **Macro F1**% and the other two tasks use **F1**% as the metric for label imbalance. The darker shades of red indicate the scores are closer to $100\%$.

## 4 CAN LLMs WITH PROMPTING ENGINEERING BEAT TRADITIONAL ML MODELS?

It has been shown that different prompting engineering techniques can exert a notable influence on LLMs' performance across various tasks (Schulhoff et al., 2024; Sahoo et al., 2024). Thus, we investigate the effectiveness of different prompting methods on both general-purpose and medical LLMs for clinical prediction, and also compare them with traditional ML models in this section.

We studied four typical prompting engineering techniques including Zero-shot Chain-of-Thought (Kojima et al., 2022), Self-Reflection (Pan et al., 2023), Role-Playing (Chen et al., 2024b), and In-Context Learning (Dong et al., 2023). The examples of the specific prompt design for different strategies across three tasks are shown in Appendix D. As shown in Table 3, we can observe that **the effectiveness of different prompting strategies is generally very limited though it may vary across models and tasks**. Except that In-Context Learning can explicitly enhance the performance of Llama3-8B, Gemma2-9B, and Internist-7B on Length-of-Stay Prediction, other strategies do not bring a notable improvement and could even degrade the original performance. More importantly, through the shades of red color, we can clearly see that **traditional ML models such as XGBoost, SVM, RNN and Transformer still outperform LLMs with different prompting strategies**.

> **Finding 2:** The effectiveness of typical prompting engineering techniques is generally limited and they still cannot make general-purpose and medical LLMs beat traditional ML models yet.

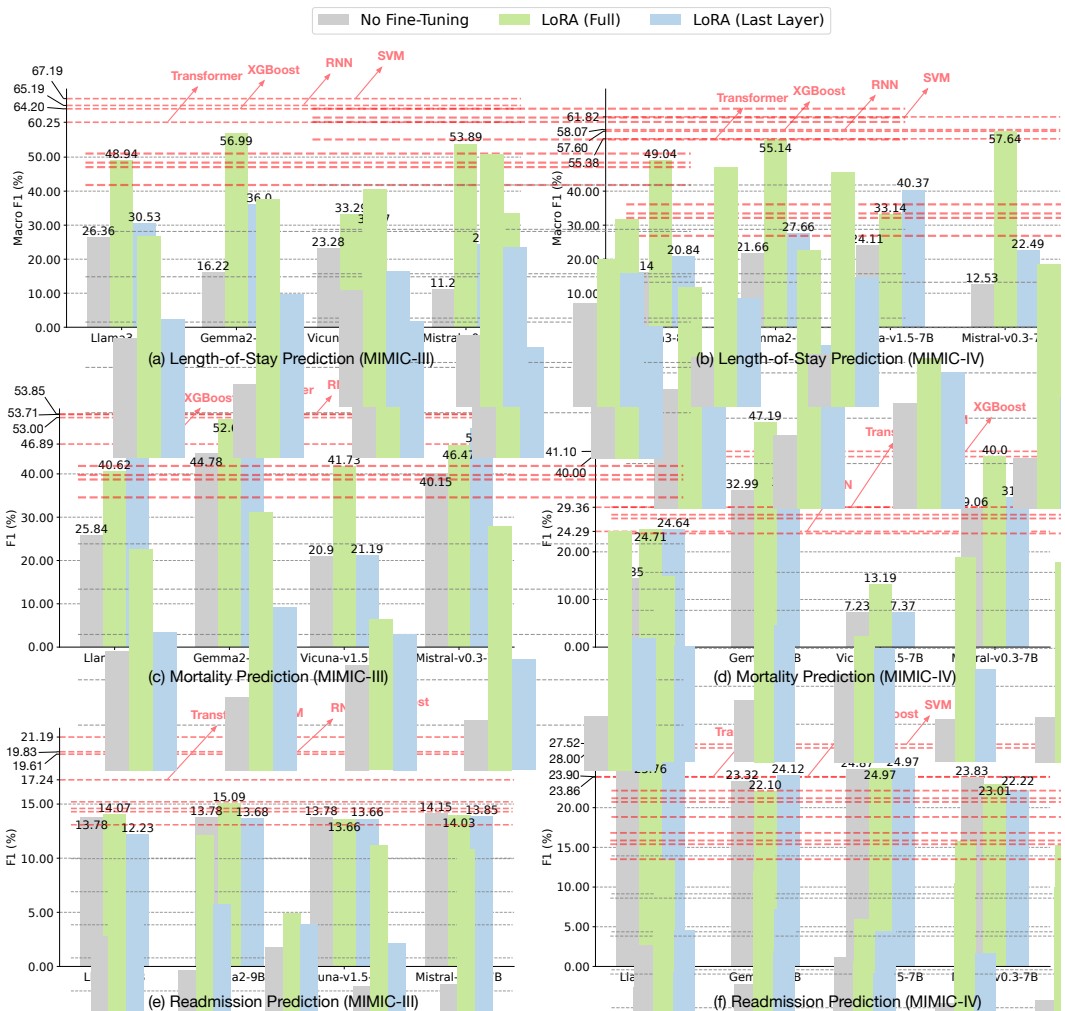

Figure 4: **Performance Comparison Between Fine-tuned LLMs and Traditional ML Models on Clinical Prediction Tasks**. Experiments are conducted on MIMIC-III and MIMIC-IV. Length-of-Stay Prediction adopts **Macro F1**% and the others use **F1**% as the metric. **LoRA (Full)** and **LoRA (Last Layer)** refer to applying LoRA (Hu et al., 2022) to full layers and only last layer respectively.

## 5 CAN FINE-TUNED LLMs BEAT TRADITIONAL ML MODELS?

Fine-tuning is a common paradigm to adapt LLMs to specific tasks or domains (Zhang et al., 2023; Lou et al., 2024). Our results on medical LLMs in Table 1 have shown that only adapting general-purpose LLMs to medical domain may not improve and could even degrade the clinical reasoning capacities. In this section, we explored whether directly adapting general-purpose LLMs to clinical prediction tasks can enhance their performance and also compare them with traditional ML models.

For a fair comparison, we adopted the same dataset division of *training*, *validation* and *test* sets for fine-tuning LLMs and training traditional ML models. The details of fine-tuning data construction for the three clinical prediction tasks are in Appendix C. Considering the relatively small scale of the *training* set, we adjusted the original LoRA (Hu et al., 2022) into two fine-tuning strategies named LoRA (Full) and LoRA (Last Layer). We set the training epochs for both LLMs and traditional ML models as 20. The loss curves of fine-tuning in Appendix B.5 show that LLMs can mostly converge within 20 epochs and we leveraged the *validation* set to select the optimal fine-tuned checkpoint.

As shown in Figure 4, we can observe that **fine-tuning can clearly enhance the clinical prediction performance of LLMs on Length-of-Stay Prediction and Mortality Prediction**, though it may not improve the performance on Readmission Prediction. More specifically, LoRA (Full) brings

a more evident improvement than LoRA (Last Layer) across different LLMs on Length-of-Stay Prediction. The enhancement from fine-tuning on Mortality Prediction is generally less notable than that on Length-of-Stay Prediction. However, **most fine-tuned LLMs still cannot surpass typical traditional ML models such as XGBoost, SVM, RNN and Transformer**. In particular, even with the substantial improvement brought by fine-tuning on Length-of-Stay Prediction, there is still an explicit gap between the performance of most LLMs and typical traditional ML models. Although the fine-tuned Gemma2-9B with LoRA (Full) has been demonstrated to surpass the aforementioned four traditional ML models on MIMIC-IV for Mortality Prediction, the same fine-tuning strategy cannot bring a similar improvement to Gemma2-9B on MIMIC-III for Mortality Prediction.

> **Finding 3:** Fine-tuning can clearly enhance LLMs' performance on certain clinical prediction tasks but most fine-tuned LLMs still cannot beat traditional ML models yet.

## 6 IMPLICATIONS AND HYPOTHESIS

It has attracted increasing attention to transform the existing healthcare systems with LLMs (Zhou et al., 2023; Liu et al., 2024b). While the high performance in medical licensing exams and question-answering benchmarks such as MedQA (Jin et al., 2021) and MedMCQA (Pal et al., 2022) has shown that LLMs may have clinician-level general medical knowledge, it is still highly under-explored whether they could perform clinical reasoning and decision-making with real-world complex scenarios. ClinicalBench made the first attempt to systematically and comprehensively benchmark the clinical reasoning capacities of both general-purpose and medical LLMs based on three real-world clinical prediction tasks. By comparing their performance with traditional ML models, we have shown LLMs' surprising but critical deficiency in performing real-world clinical reasoning, implying their **safety and reliability risks in practical adoption** and **the challenges of integrating LLMs into clinical workflows**. Our discoveries also echo the recent discussions on LLMs' potential limitations (Hager et al., 2024; Han et al., 2024; Gallifant et al., 2024). We hypothesize that LLMs' limited clinical reasoning abilities could be attributed to **the lack of realistic and relevant data in both of the pre-training and post-training stages**, considering the sensitivity nature of patients' information. The recent emerging clinical digital twin (Katsoulakis et al., 2024; Das et al., 2023; Sun et al., 2023) and data synthesis (Liu et al., 2024d; Bauer et al., 2024; Tan et al., 2024) techniques could be explored in the future to address this challenge. ClinicalBench could be adopted to facilitate the progress in enhancing LLMs' clinical reasoning capacities and minimize the gap between the development of LLMs for healthcare and clinical practice in the real world.

## 7 RELATED WORK

**Clinical Prediction** Clinical prediction tasks play a critical role in current healthcare systems with multifaceted significance including improving patient outcomes, optimizing hospital resources, and supporting clinical decision-making (Rajkomar et al., 2019). Length-of-Stay Prediction (Stone et al., 2022), Mortality Prediction (Jentzer et al., 2021), and Readmission Prediction (Artetxe et al., 2018) are among the most common ones. While traditional ML models such as XGBoost, SVM, and RNN have still been widely adopted in these tasks (Moazemi et al., 2023; van de Water et al., 2024), many advanced models have also been developed for clinical predictive modeling (Wang et al., 2024b). For example, Xu et al. (2023) proposed to leverage hypergraph transformers with patients as hyperedges and medical codes as nodes for predictive tasks. Jiang et al. (2024) leveraged personalized knowledge graphs and attention-augmented graph neural networks for enhancing the prediction performance. Cui et al. (2024) conducted diffusion-based data augmentation to further improve health risk prediction. Although the emerging LLMs may have been demonstrated to perform well on medical question-answering and hold great promise to transform healthcare systems, their capacities on clinical prediction tasks are largely under-explored. ClinicalBench shows that LLMs cannot beat traditional ML models yet, suggesting their critical limitations in clinical applications.

**Clinical Benchmarks for LLMs** The majority of existing clinical benchmarks for LLMs can generally be categorized from two perspectives. The first one aims to assess LLMs' capacities in *clinical text processing tasks* (Harris et al., 2024; Jahan et al., 2024; Feng et al., 2024; Chen et al., 2023a; Wang et al., 2023a; Luo et al., 2024). For example, Jahan et al. (2024) has comprehensively

evaluated four LLMs in different typical biomedical text tasks (*e.g.*, named entity recognition, relation extraction, entity linking, text classification and text summarization with biomedical texts). The second one intends to evaluate LLMs performance in ***clinical question-answering tasks***. Besides MedQA (Jin et al., 2021) and MedMCQA (Pal et al., 2022), many recent benchmarks have been built to test the medical knowledge of LLMs in different aspects (Korgul et al., 2023; Chen et al., 2024a; Vladika et al., 2024; Shoham & Rappoport, 2024). For example, Chen et al. (2024c) and Wang et al. (2024a) have developed QA benchmarks to assess the diagnostic performance of LLMs in rare diseases. Kweon et al. (2024), Kasai et al. (2023), Rosoł et al. (2023), Alonso et al. (2024), Cai et al. (2024a), Wang et al. (2023b), and Liu et al. (2024a;c) designed QA benchmarks with languages beyond English such as Korean, Japanese, Polish and Chinese. However, benchmarks on LLMs' clinical reasoning capacities with real-world complex scenarios are relatively lacking. ClinicalBench filled the gap through a comprehensive investigation involving three common clinical prediction tasks, two databases, 14 general-purpose LLMs, 8 medical LLMs and has provided valuable insights.

## 8 CONCLUSION

In this paper, we have built a new benchmark ClinicalBench to comprehensively and systematically compare the effectiveness of traditional ML models and LLMs in typical clinical prediction tasks including Length-of-Stay Prediction, Mortality Prediction, and Readmission Prediction. With extensive empirical evidence, we find that general-purpose and medical LLMs, even with different scales of parameters, diverse prompting or fine-tuning strategies, still cannot beat traditional ML models in clinical prediction yet. Our findings have illustrated the critical limitations of LLMs in performing real-world clinical reasoning and decision-making with complex scenarios. We urge practitioners to exercise caution when adopting LLMs in real-world clinical applications.

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

# Content of Appendix

## A    REPRODUCIBILITY STATEMENT

We conduct the experiments on eight NVIDIA RTX A6000 GPUs. We have benchmarked 14 general-prupose LLMs including Llama3-8B (Dubey et al., 2024), Llama3-70B (Dubey et al., 2024), Mistral-v0.3-7B (Jiang et al., 2023), Gemma2-9B (Team et al., 2024), Qwen2-0.5B (Yang et al., 2024), Qwen2-1.5B (Yang et al., 2024), Qwen2-7B (Yang et al., 2024), Yi-v1.5-6B (Young et al., 2024), Yi-v1.5-9B (Young et al., 2024), Yi-v1.5-34B (Young et al., 2024), Vicuna-v1.5-7B (Zheng et al., 2023), Phi3.5-mini-3.8B (Abdin et al., 2024), InternLM2.5-7B (Cai et al., 2024b), MiniCPM3-4B (Hu et al., 2024) and 8 medical LLMs including Meditron-7B (Chen et al., 2023b), Meditron-70B (Chen et al., 2023b), Medllama3-8B, BioMistral-7B (Labrak et al., 2024), Med42-8B (Christophe et al., 2024), Med42-70B (Christophe et al., 2024), BioMedGPT-7B (Luo et al., 2023) and Internist-7B (Griot et al., 2024). The model checkpoints are downloaded from `https://huggingface.co/`. The specific download links are as follows:

- Llama3-8B: `https://huggingface.co/meta-llama/Meta-Llama-3-8B-Instruct`
- Llama3-70B: `https://huggingface.co/meta-llama/Meta-Llama-3-70B-Instruct`
- Mistral-v0.3-7B: `https://huggingface.co/mistralai/Mistral-7B-Instruct-v0.3`
- Gemma2-9B: `https://huggingface.co/google/gemma-2-9b-it`
- Qwen2-0.5B: `https://huggingface.co/Qwen/Qwen2-0.5B-Instruct`
- Qwen2-1.5B: `https://huggingface.co/Qwen/Qwen2-1.5B-Instruct`
- Qwen2-7B: `https://huggingface.co/Qwen/Qwen2-7B-Instruct`
- Yi-v1.5-6B: `https://huggingface.co/01-ai/Yi-1.5-6B-Chat`
- Yi-v1.5-9B: `https://huggingface.co/01-ai/Yi-1.5-9B-Chat`
- Yi-v1.5-34B: `https://huggingface.co/01-ai/Yi-1.5-34B-Chat`
- Vicuna-v1.5-7B: `https://huggingface.co/lmsys/vicuna-7b-v1.5`
- Phi3.5-mini-3.8B: `https://huggingface.co/microsoft/Phi-3.5-mini-instruct`
- InternLM2.5-7B: `https://huggingface.co/internlm/internlm2_5-7b-chat`
- MiniCPM3-4B: `https://huggingface.co/openbmb/MiniCPM3-4B`
- Meditron-7B: `https://huggingface.co/epfl-llm/meditron-7b`
- Meditron-70B: `https://huggingface.co/epfl-llm/meditron-70b`
- Medllama3-8B: `https://huggingface.co/ProbeMedicalYonseiMAILab/medllama3-v20`
- BioMistral-7B: :`https://huggingface.co/BioMistral/BioMistral-7B`
- Med42-8B: `https://huggingface.co/m42-health/Llama3-Med42-8B`
- Med42-70B: `https://huggingface.co/m42-health/Llama3-Med42-70B`
- BioMedGPT-7B: `https://huggingface.co/PharMolix/BioMedGPT-LM-7B`
- Internist-7B: `https://huggingface.co/internistai/base-7b-v0.2`

# B  MORE EXPERIMENT RESULTS

## B.1  RESULTS OF TRADITIONAL ML MODELS AND DIRECTLY PROMPTING LLMS ON MIMIC-IV

| Method | Length-of-Stay Prediction | | Mortality Prediction | | Readmission Prediction | |
|---|---|---|---|---|---|---|
| | **Macro F1 (%)** 95% CI | **AUROC (%)** 95% CI | **F1 (%)** 95% CI | **AUROC (%)** 95% CI | **F1 (%)** 95% CI | **AUROC (%)** 95% CI |
| **Majority** | 29.56 (29.56, 29.56) | 50.0 (50.0, 50.0) | 0.0 (0.0, 0.0) | 50.0 (50.0, 50.0) | 0.0 (0.0, 0.0) | 50.0 (50.0, 50.0) |
| **Minority** | 4.56 (4.56, 4.56) | 50.0 (50.0, 50.0) | 6.7 (6.7, 6.7) | 50.0 (50.0, 50.0) | 24.92 (24.92, 24.92) | 50.0 (50.0, 50.0) |
| *Traditional ML Models* | | | | | | |
| **XGBoost** | 63.33 (63.22, 63.44) | 95.43 (95.40, 95.46) | 48.85 (48.59, 49.11) | 98.24 (98.20, 98.28) | 27.60 (27.46, 27.74) | 60.37 (60.16, 60.58) |
| **LR** | 61.78 (61.73, 61.83) | 94.52 (94.49, 94.55) | 50.90 (50.78, 51.02) | 98.12 (98.08, 98.16) | 26.67 (26.52, 26.82) | 58.71 (58.48, 58.94) |
| **DecisionTree** | 55.48 (55.18, 55.78) | 89.23 (89.11, 89.35) | 32.65 (32.43, 32.87) | 88.78 (88.58, 88.98) | 26.14 (26.07, 26.21) | 56.46 (56.32, 56.6) |
| **RandomForest** | 57.76 (57.61, 57.91) | 92.23 (92.14, 92.32) | 34.01 (33.66, 34.36) | 95.08 (95.01, 95.15) | 28.03 (27.93, 28.13) | 61.41 (61.32, 61.5) |
| **AdaBoost** | 62.17 (62.05, 62.29) | 93.23 (93.16, 93.30) | 48.00 (47.50, 48.50) | 97.08 (97.02, 97.14) | 26.90 (26.78, 27.02) | 58.83 (58.63, 59.03) |
| **SVM** | 66.06 (65.98, 66.14) | 96.24 (96.18, 96.30) | 46.28 (46.07, 46.49) | 98.11 (98.09, 98.13) | 28.82 (28.70, 28.94) | 62.44 (62.27, 62.61) |
| **NaiveBayes** | 48.03 (47.98, 48.08) | 66.52 (66.32, 66.72) | 18.83 (18.62, 19.04) | 82.04 (81.81, 82.27) | 25.51 (25.47, 25.55) | 52.95 (52.88, 53.02) |
| **KNN** | 47.24 (47.08, 47.40) | 75.29 (74.97, 75.61) | 46.89 (46.73, 47.05) | 89.70 (89.63, 89.77) | 24.85 (24.73, 24.97) | 55.40 (55.16, 55.64) |
| **MLP** | 58.23 (58.15, 58.31) | 91.61 (91.55, 91.67) | 46.33 (46.13, 46.53) | 97.54 (97.50, 97.58) | 25.38 (25.26, 25.50) | 56.7 (56.49, 56.91) |
| **Transformer** | 57.06 (56.95, 57.17) | 92.78 (92.72, 92.84) | 43.95 (43.43, 44.47) | 97.87 (97.84, 97.9) | 26.22 (26.16, 26.28) | 57.3 (57.12, 57.48) |
| **RNN** | 63.58 (63.47, 63.69) | 93.66 (93.63, 93.69) | 49.49 (49.33, 49.65) | 97.32 (97.27, 97.37) | 26.64 (26.48, 26.8) | 57.65 (57.42, 57.88) |
| *General-purpose Large Language Models* | | | | | | |
| **LLama3-8B** | 18.08 (18.02, 18.14) | 91.72 (91.64, 91.8) | 14.97 (14.95, 14.99) | 95.45 (95.4, 95.5) | 25.1 (25.09, 25.11) | 49.6 (49.48, 49.72) |
| **Mistral-v0.3-7B** | 12.29 (12.22, 12.36) | 86.1 (86.07, 86.13) | 28.97 (28.89, 29.05) | 95.64 (95.6, 95.68) | 24.7 (24.6, 24.8) | 51.22 (51.06, 51.38) |
| **Gemma2-9B** | 19.83 (19.79, 19.87) | 89.68 (89.6, 89.76) | 32.91 (32.86, 32.96) | 95.82 (95.78, 95.86) | 24.83 (24.81, 24.85) | 49.92 (49.79, 50.05) |
| **Qwen2-7B** | 6.57 (6.54, 6.6) | 83.46 (83.28, 83.64) | 48.05 (47.63, 48.47) | 95.46 (95.42, 95.5) | 0.23 (0.21, 0.25) | 48.83 (48.72, 48.94) |
| **Yi-v1.5-9B** | 6.97 (6.96, 6.98) | 84.35 (84.3, 84.4) | 16.87 (16.82, 16.92) | 88.89 (88.79, 88.99) | 19.14 (19.0, 19.28) | 50.33 (50.23, 50.43) |
| **Vicuna-v1.5-7B** | 23.25 (23.17, 23.33) | 81.12 (80.99, 81.25) | 7.22 (7.22, 7.22) | 88.39 (88.33, 88.45) | 24.92 (24.92, 24.92) | 51.92 (51.87, 51.97) |
| **Phi3.5-mini-3.8B** | 11.56 (11.51, 11.61) | 82.84 (82.70, 82.98) | 16.72 (16.68, 16.76) | 93.55 (93.50, 93.60) | 21.52 (21.40, 21.64) | 50.87 (50.73, 51.01) |
| **InternLM2.5-7B** | 11.40 (11.35, 11.45) | 90.42 (90.31, 90.53) | 40.68 (40.39, 40.97) | 95.29 (95.25, 95.33) | 0.00 (0.00, 0.00) | 49.05 (48.89, 49.21) |
| **MiniCPM3-4B** | 17.80 (17.71, 17.89) | 86.41 (86.29, 86.53) | 12.28 (12.26, 12.30) | 94.54 (94.47, 94.61) | 22.93 (22.89, 22.97) | 49.90 (49.80, 50.00) |
| *Medical Large Language Models* | | | | | | |
| **Meditron-7B** | 28.08 (27.96, 28.2) | 86.17 (86.03, 86.31) | 6.7 (6.7, 6.7) | 91.95 (91.9, 92.0) | 24.92 (24.92, 24.92) | 48.92 (48.75, 49.09) |
| **Medllama3-8B** | 8.31 (8.27, 8.35) | 80.73 (80.57, 80.89) | 12.86 (12.84, 12.88) | 91.48 (91.46, 91.5) | 24.92 (24.92, 24.92) | 48.22 (48.04, 48.4) |
| **BioMistral-7B** | 13.53 (13.46, 13.6) | 39.12 (38.99, 39.25) | 22.81 (22.75, 22.87) | 91.89 (91.81, 91.97) | 22.03 (21.94, 22.12) | 48.54 (48.45, 48.63) |
| **Med42-8B** | 9.95 (9.9, 10.0) | 90.86 (90.76, 90.96) | 16.7 (16.67, 16.73) | 93.62 (93.56, 93.68) | 25.61 (25.55, 25.67) | 50.78 (50.67, 50.89) |
| **BioMedGPT-7B** | 7.65 (7.65, 7.65) | 70.48 (70.22, 70.74) | 12.01 (12.00, 12.02) | 93.51 (93.49, 93.53) | 23.97 (23.92, 24.02) | 50.35 (50.30, 50.40) |
| **Internist-7B** | 14.24 (14.18, 14.30) | 89.39 (89.32, 89.46) | 28.81 (28.75, 28.87) | 96.57 (96.54, 96.60) | 15.72 (15.59, 15.85) | 50.59 (50.43, 50.75) |

Table 4: **Performance of LLMs and Traditional ML Models on Typical Clinical Prediction Tasks**. Experiments are conducted in **MIMIC-IV** dataset. The numbers in bracket are the ranges of performance with **95% Confidence Interval (95% CI)**, which are calculated with the 5-run scores.

## B.2  RESULTS OF TRADITIONAL ML MODELS ON DIFFERENT SCALES OF TRAINING SET

| Method | Length-of-Stay Prediction | | Mortality Prediction | | Readmission Prediction | |
|---|---|---|---|---|---|---|
| | Macro F1 (%) 95% CI | AUROC (%) 95% CI | F1 (%) 95% CI | AUROC (%) 95% CI | F1 (%) 95% CI | AUROC (%) 95% CI |
| **Majority** | 23.37 (23.37, 23.37) | 50.00 (50.00, 50.00) | 0.00 (0.00, 0.00) | 50.00 (50.00, 50.00) | 0.00 (0.00, 0.00) | 50.00 (50.00, 50.00) |
| **Minority** | 10.72 (10.72, 10.72) | 50.00 (50.00, 50.00) | 20.88 (20.88, 20.88) | 50.00 (50.00, 50.00) | 13.64 (13.64, 13.64) | 50.00 (50.00, 50.00) |
| *Traditional ML Models with 40% of Original Training Set from MIMIC-III* | | | | | | |
| **XGBoost** | 63.91 (63.8, 64.02) | 91.6 (91.52, 91.68) | 60.53 (60.41, 60.65) | 94.2 (94.15, 94.25) | 17.68 (17.51, 17.85) | 64.3 (64.0, 64.6) |
| **LR** | 62.51 (62.35, 62.67) | 90.64 (90.5, 90.78) | 59.42 (59.29, 59.55) | 92.42 (92.37, 92.47) | 17.63 (17.44, 17.82) | 65.37 (64.89, 65.85) |
| **DecisionTree** | 56.18 (56.03, 56.33) | 83.24 (83.03, 83.45) | 47.13 (47.01, 47.25) | 81.39 (81.18, 81.6) | 14.76 (14.72, 14.8) | 54.29 (54.07, 54.51) |
| **RandomForest** | 60.83 (60.72, 60.94) | 90.86 (90.81, 90.91) | 51.74 (51.54, 51.94) | 89.96 (89.83, 90.09) | 18.03 (17.89, 18.17) | 65.65 (65.36, 65.94) |
| **AdaBoost** | 62.52 (62.31, 62.73) | 89.43 (89.15, 89.71) | 55.79 (55.59, 55.99) | 90.25 (90.06, 90.44) | 17.35 (17.15, 17.55) | 62.0 (61.53, 62.47) |
| **SVM** | 65.08 (64.95, 65.21) | 91.68 (91.6, 91.76) | 59.75 (59.53, 59.97) | 93.68 (93.62, 93.74) | 18.3 (18.07, 18.53) | 65.44 (65.02, 65.86) |
| **NaiveBayes** | 45.57 (45.48, 45.66) | 62.48 (62.28, 62.68) | 24.86 (24.81, 24.91) | 59.1 (59.02, 59.18) | 16.41 (16.31, 16.51) | 57.66 (57.45, 57.87) |
| **KNN** | 44.78 (44.71, 44.85) | 74.84 (74.69, 74.99) | 41.53 (41.22, 41.84) | 83.81 (83.54, 84.08) | 14.94 (14.89, 14.99) | 59.45 (59.19, 59.71) |
| **MLP** | 59.08 (58.96, 59.2) | 87.01 (86.86, 87.16) | 56.1 (55.99, 56.21) | 90.17 (90.11, 90.23) | 19.34 (19.11, 19.57) | 65.63 (65.22, 66.04) |
| **Transformer** | 59.44 (59.26, 59.62) | 86.93 (86.8, 87.06) | 54.59 (54.26, 54.92) | 91.17 (91.05, 91.29) | 18.26 (18.08, 18.44) | 64.38 (64.03, 64.73) |
| **RNN** | 62.44 (62.26, 62.62) | 88.5 (88.37, 88.63) | 58.26 (58.14, 58.38) | 89.44 (89.32, 89.56) | 19.61 (19.38, 19.84) | 67.23 (66.89, 67.57) |
| *Traditional ML Models with 20% of Original Training Set from MIMIC-III* | | | | | | |
| **XGBoost** | 62.26 (62.14, 62.38) | 90.93 (90.84, 91.02) | 56.98 (56.61, 57.35) | 92.34 (92.23, 92.45) | 15.8 (15.59, 16.01) | 60.55 (60.18, 60.92) |
| **LR** | 60.52 (60.27, 60.77) | 89.69 (89.5, 89.88) | 55.27 (54.94, 55.6) | 90.99 (90.87, 91.11) | 16.98 (16.76, 17.2) | 62.37 (61.91, 62.83) |
| **DecisionTree** | 54.73 (54.57, 54.89) | 80.14 (79.94, 80.34) | 44.94 (44.43, 45.45) | 76.99 (76.71, 77.27) | 14.66 (14.55, 14.77) | 55.62 (55.31, 55.93) |
| **RandomForest** | 60.36 (60.2, 60.52) | 90.65 (90.52, 90.78) | 50.52 (50.31, 50.73) | 89.23 (89.14, 89.32) | 16.15 (16.02, 16.28) | 62.38 (62.11, 62.65) |
| **AdaBoost** | 60.79 (60.65, 60.93) | 88.14 (87.96, 88.32) | 51.95 (51.51, 52.39) | 87.71 (87.48, 87.94) | 16.02 (15.89, 16.15) | 58.35 (58.02, 58.68) |
| **SVM** | 63.04 (62.93, 63.15) | 91.02 (90.94, 91.1) | 56.06 (55.74, 56.38) | 92.26 (92.17, 92.35) | 17.17 (16.99, 17.35) | 63.7 (63.35, 64.05) |
| **NaiveBayes** | 43.82 (43.77, 43.87) | 62.88 (62.7, 63.06) | 25.63 (25.57, 25.69) | 60.44 (60.34, 60.54) | 16.82 (16.65, 16.99) | 58.21 (57.87, 58.55) |
| **KNN** | 44.39 (44.2, 44.58) | 74.48 (74.27, 74.69) | 41.03 (40.77, 41.29) | 82.86 (82.66, 83.06) | 14.72 (14.65, 14.79) | 59.46 (58.95, 59.97) |
| **MLP** | 57.17 (56.86, 57.48) | 85.85 (85.63, 86.07) | 53.01 (52.68, 53.34) | 89.27 (89.11, 89.43) | 16.58 (16.3, 16.86) | 61.92 (61.47, 62.37) |
| **Transformer** | 58.11 (57.84, 58.38) | 87.64 (87.38, 87.9) | 52.56 (52.11, 53.01) | 90.29 (90.11, 90.47) | 16.82 (16.69, 16.95) | 61.36 (61.07, 61.65) |
| **RNN** | 61.67 (61.38, 61.96) | 88.45 (88.26, 88.64) | 55.31 (54.94, 55.68) | 88.42 (88.2, 88.64) | 19.14 (18.9, 19.38) | 63.92 (63.69, 64.15) |

Table 5: **Performance of Traditional ML Models on Typical Clinical Prediction Tasks with Different Scales of Training Set**. Experiments are conducted in **MIMIC-III** dataset. The training set is obtained through stratified sampling from the original training set in Table 1. The numbers in bracket are the ranges of performance with **95% Confidence Interval (95% CI)**, which are calculated with the 5-run scores.

| Method | Length-of-Stay Prediction | | Mortality Prediction | | Readmission Prediction | |
| --- | --- | --- | --- | --- | --- | --- |
| | **Macro F1 (%)** 95% CI | **AUROC (%)** 95% CI | **F1 (%)** 95% CI | **AUROC (%)** 95% CI | **F1 (%)** 95% CI | **AUROC (%)** 95% CI |
| **Majority** | 23.37 (23.37, 23.37) | 50.00 (50.00, 50.00) | 0.00 (0.00, 0.00) | 50.00 (50.00, 50.00) | 0.00 (0.00, 0.00) | 50.00 (50.00, 50.00) |
| **Minority** | 10.72 (10.72, 10.72) | 50.00 (50.00, 50.00) | 20.88 (20.88, 20.88) | 50.00 (50.00, 50.00) | 13.64 (13.64, 13.64) | 50.00 (50.00, 50.00) |
| *Traditional ML Models with 10% of Original Training Set from MIMIC-III* | | | | | | |
| **XGBoost** | 61.38 (61.22, 61.54) | 90.25 (90.13, 90.37) | 50.83 (50.46, 51.2) | 89.61 (89.43, 89.79) | 16.84 (16.6, 17.08) | 61.41 (60.83, 61.99) |
| **LR** | 60.77 (60.59, 60.95) | 90.19 (90.05, 90.33) | 53.21 (52.91, 53.51) | 89.94 (89.81, 90.07) | 17.36 (17.11, 17.61) | 62.26 (61.96, 62.56) |
| **DecisionTree** | 53.08 (52.97, 53.19) | 77.14 (76.78, 77.5) | 40.69 (40.43, 40.95) | 75.1 (74.77, 75.43) | 14.95 (14.81, 15.09) | 55.91 (55.34, 56.48) |
| **RandomForest** | 60.05 (59.89, 60.21) | 90.27 (90.18, 90.36) | 48.62 (48.31, 48.93) | 88.18 (87.99, 88.37) | 16.58 (16.44, 16.72) | 61.47 (61.27, 61.67) |
| **AdaBoost** | 59.9 (59.8, 60.0) | 87.2 (87.0, 87.4) | 48.94 (48.62, 49.26) | 85.3 (85.11, 85.49) | 15.51 (15.37, 15.65) | 57.14 (56.71, 57.57) |
| **SVM** | 62.6 (62.48, 62.72) | 90.72 (90.66, 90.78) | 51.91 (51.55, 52.27) | 90.73 (90.59, 90.87) | 16.84 (16.63, 17.05) | 45.66 (43.96, 47.36) |
| **NaiveBayes** | 41.54 (41.39, 41.69) | 63.82 (63.65, 63.99) | 26.48 (26.32, 26.64) | 62.18 (61.81, 62.55) | 15.44 (15.33, 15.55) | 55.3 (55.09, 55.51) |
| **KNN** | 43.35 (43.13, 43.57) | 73.5 (73.23, 73.77) | 40.12 (39.82, 40.42) | 82.63 (82.47, 82.79) | 14.48 (14.39, 14.57) | 56.86 (56.58, 57.14) |
| **MLP** | 56.93 (56.68, 57.18) | 85.96 (85.75, 86.17) | 50.54 (50.23, 50.85) | 88.87 (88.79, 88.95) | 16.4 (16.22, 16.58) | 59.36 (59.09, 59.63) |
| **Transformer** | 57.98 (57.78, 58.18) | 88.5 (88.28, 88.72) | 50.76 (50.46, 51.06) | 89.33 (89.21, 89.45) | 15.15 (15.0, 15.3) | 57.59 (57.41, 57.77) |
| **RNN** | 61.51 (61.25, 61.77) | 88.57 (88.41, 88.73) | 53.81 (53.51, 54.11) | 88.23 (88.09, 88.37) | 9.13 (8.38, 9.88) | 61.97 (61.73, 62.21) |
| *Traditional ML Models with 5% of Original Training Set from MIMIC-III* | | | | | | |
| **XGBoost** | 59.22 (59.0, 59.44) | 89.27 (89.12, 89.42) | 48.62 (48.47, 48.77) | 87.47 (87.41, 87.53) | 16.03 (15.85, 16.21) | 58.28 (57.99, 58.57) |
| **LR** | 60.01 (59.87, 60.15) | 89.97 (89.81, 90.13) | 52.56 (52.28, 52.84) | 88.97 (88.81, 89.13) | 16.05 (15.87, 16.23) | 59.63 (59.37, 59.89) |
| **DecisionTree** | 52.41 (52.29, 52.53) | 76.2 (75.96, 76.44) | 39.33 (38.89, 39.77) | 71.41 (70.92, 71.9) | 16.0 (15.88, 16.12) | 56.57 (56.35, 56.79) |
| **RandomForest** | 60.07 (59.86, 60.28) | 90.35 (90.27, 90.43) | 50.14 (49.84, 50.44) | 88.39 (88.22, 88.56) | 15.9 (15.78, 16.02) | 61.34 (61.06, 61.62) |
| **AdaBoost** | 56.82 (56.56, 57.08) | 86.79 (86.64, 86.94) | 44.83 (44.66, 45.0) | 83.22 (83.02, 83.42) | 15.67 (15.51, 15.83) | 58.59 (58.32, 58.86) |
| **SVM** | 61.63 (61.53, 61.73) | 90.1 (90.01, 90.19) | 51.29 (50.94, 51.64) | 89.46 (89.33, 89.59) | 16.25 (16.06, 16.44) | 61.02 (60.68, 61.36) |
| **NaiveBayes** | 42.91 (42.71, 43.11) | 66.12 (65.9, 66.34) | 29.89 (29.72, 30.06) | 66.44 (66.25, 66.63) | 14.58 (14.4, 14.76) | 53.94 (53.75, 54.13) |
| **KNN** | 40.78 (40.69, 40.87) | 71.91 (71.63, 72.19) | 39.87 (39.31, 40.43) | 82.65 (82.52, 82.78) | 14.14 (14.09, 14.19) | 55.36 (54.92, 55.8) |
| **MLP** | 56.19 (55.95, 56.43) | 85.5 (85.3, 85.7) | 49.36 (49.12, 49.6) | 87.51 (87.34, 87.68) | 15.37 (15.26, 15.48) | 57.12 (56.89, 57.35) |
| **Transformer** | 56.8 (56.6, 57.0) | 87.99 (87.77, 88.21) | 48.81 (48.38, 49.24) | 87.6 (87.4, 87.8) | 13.77 (13.61, 13.93) | 52.6 (52.35, 52.85) |
| **RNN** | 60.77 (60.53, 61.01) | 87.05 (86.86, 87.24) | 51.75 (51.44, 52.06) | 87.05 (86.86, 87.24) | 12.9 (12.17, 13.63) | 59.04 (58.81, 59.27) |

Table 6: **Performance of Traditional ML Models on Typical Clinical Prediction Tasks with Different Scales of Training Set**. Experiments are conducted in **MIMIC-III** dataset. The training set is obtained through stratified sampling from the original training set in Table 1. The numbers in bracket are the ranges of performance with **95% Confidence Interval (95% CI)**, which are calculated with the 5-run scores.

| Method | Length-of-Stay Prediction | | Mortality Prediction | | Readmission Prediction | |
|---|---|---|---|---|---|---|
| | **Macro F1 (%)** 95% CI | **AUROC (%)** 95% CI | **F1 (%)** 95% CI | **AUROC (%)** 95% CI | **F1 (%)** 95% CI | **AUROC (%)** 95% CI |
| **Majority** | 29.56 (29.56, 29.56) | 50.0 (50.0, 50.0) | 0.0 (0.0, 0.0) | 50.0 (50.0, 50.0) | 0.0 (0.0, 0.0) | 50.0 (50.0, 50.0) |
| **Minority** | 4.56 (4.56, 4.56) | 50.0 (50.0, 50.0) | 6.7 (6.7, 6.7) | 50.0 (50.0, 50.0) | 24.92 (24.92, 24.92) | 50.0 (50.0, 50.0) |
| *Traditional ML Models with 40% of Original Training Set from MIMIC-IV* | | | | | | |
| **XGBoost** | 62.05 (61.92, 62.18) | 94.51 (94.46, 94.56) | 44.35 (44.07, 44.63) | 97.27 (97.22, 97.32) | 26.73 (26.7, 26.76) | 58.34 (58.22, 58.46) |
| **LR** | 60.69 (60.49, 60.89) | 93.7 (93.64, 93.76) | 47.2 (46.99, 47.41) | 97.45 (97.41, 97.49) | 26.19 (26.11, 26.27) | 58.04 (57.86, 58.22) |
| **DecisionTree** | 53.14 (52.85, 53.43) | 86.59 (86.51, 86.67) | 31.34 (30.88, 31.8) | 84.88 (84.66, 85.1) | 26.09 (26.04, 26.14) | 54.92 (54.8, 55.04) |
| **RandomForest** | 57.63 (57.51, 57.75) | 92.04 (91.95, 92.13) | 33.86 (33.32, 34.4) | 94.79 (94.64, 94.94) | 26.97 (26.92, 27.02) | 59.84 (59.71, 59.97) |
| **AdaBoost** | 60.81 (60.71, 60.91) | 92.3 (92.2, 92.4) | 43.24 (42.8, 43.68) | 95.33 (95.15, 95.51) | 26.33 (26.23, 26.43) | 56.44 (56.27, 56.61) |
| **SVM** | 64.64 (64.51, 64.77) | 95.59 (95.54, 95.64) | 42.78 (42.49, 43.07) | 97.36 (97.33, 97.39) | 28.44 (28.31, 28.57) | 60.9 (60.72, 61.08) |
| **NaiveBayes** | 42.47 (42.37, 42.57) | 70.0 (69.84, 70.16) | 16.13 (15.99, 16.27) | 77.43 (77.32, 77.54) | 25.27 (25.15, 25.39) | 52.69 (52.48, 52.9) |
| **KNN** | 44.5 (44.29, 44.71) | 71.41 (71.17, 71.65) | 42.8 (42.5, 43.1) | 89.51 (89.31, 89.71) | 23.39 (23.09, 23.69) | 53.7 (53.36, 54.04) |
| **MLP** | 57.11 (56.97, 57.25) | 89.83 (89.77, 89.89) | 42.14 (41.91, 42.37) | 96.8 (96.77, 96.83) | 25.79 (25.63, 25.95) | 56.82 (56.64, 57.0) |
| **Transformer** | 57.82 (57.77, 57.87) | 91.02 (90.84, 91.2) | 40.47 (40.14, 40.8) | 97.24 (97.21, 97.27) | 25.56 (25.46, 25.66) | 56.39 (56.31, 56.47) |
| **RNN** | 61.97 (61.86, 62.08) | 91.51 (91.44, 91.58) | 46.1 (45.89, 46.31) | 96.31 (96.27, 96.35) | 25.52 (25.39, 25.65) | 56.99 (56.87, 57.11) |
| *Traditional ML Models with 20% of Original Training Set from MIMIC-IV* | | | | | | |
| **XGBoost** | 60.24 (60.16, 60.32) | 93.49 (93.41, 93.57) | 39.34 (38.94, 39.74) | 95.9 (95.81, 95.99) | 25.84 (25.77, 25.91) | 56.76 (56.57, 56.95) |
| **LR** | 60.04 (59.88, 60.2) | 92.98 (92.82, 93.14) | 43.39 (43.03, 43.75) | 96.68 (96.63, 96.73) | 25.48 (25.37, 25.59) | 56.25 (56.13, 56.37) |
| **DecisionTree** | 51.88 (51.6, 52.16) | 83.38 (83.19, 83.57) | 26.88 (26.1, 27.66) | 82.63 (82.29, 82.97) | 21.59 (21.01, 22.17) | 52.31 (52.08, 52.54) |
| **RandomForest** | 57.32 (57.13, 57.51) | 91.71 (91.6, 91.82) | 33.69 (33.22, 34.16) | 94.41 (94.25, 94.57) | 27.09 (27.03, 27.15) | 58.76 (58.52, 59.0) |
| **AdaBoost** | 58.46 (58.27, 58.65) | 92.23 (92.1, 92.36) | 38.71 (38.36, 39.06) | 94.1 (93.93, 94.27) | 24.58 (24.46, 24.7) | 54.43 (54.31, 54.55) |
| **SVM** | 63.1 (63.0, 63.2) | 94.99 (94.92, 95.06) | 40.0 (39.62, 40.38) | 96.36 (96.3, 96.42) | 26.6 (26.43, 26.77) | 58.55 (58.32, 58.78) |
| **NaiveBayes** | 39.13 (39.0, 39.26) | 70.63 (70.41, 70.85) | 14.53 (14.4, 14.66) | 75.68 (75.44, 75.92) | 24.64 (24.45, 24.83) | 53.75 (53.52, 53.98) |
| **KNN** | 43.72 (43.57, 43.87) | 72.58 (72.31, 72.85) | 40.05 (39.64, 40.46) | 86.05 (85.74, 86.36) | 23.05 (22.95, 23.15) | 53.07 (52.88, 53.26) |
| **MLP** | 55.42 (55.28, 55.56) | 88.33 (88.16, 88.5) | 39.63 (39.3, 39.96) | 96.11 (96.05, 96.17) | 24.58 (24.48, 24.68) | 54.86 (54.73, 54.99) |
| **Transformer** | 56.93 (56.81, 57.05) | 90.98 (90.8, 91.16) | 37.91 (37.47, 38.35) | 96.29 (96.22, 96.36) | 24.79 (24.7, 24.88) | 54.86 (54.72, 55.0) |
| **RNN** | 60.43 (60.27, 60.59) | 90.04 (89.89, 90.19) | 43.05 (42.75, 43.35) | 95.49 (95.42, 95.56) | 25.01 (24.86, 25.16) | 55.59 (55.44, 55.74) |

Table 7: **Performance of Traditional ML Models on Typical Clinical Prediction Tasks with Different Scales of Training Set**. Experiments are conducted in **MIMIC-IV** dataset. The training set is obtained through stratified sampling from the original training set in Table 1. The numbers in bracket are the ranges of performance with **95% Confidence Interval (95% CI)**, which are calculated with the 5-run scores.

| Method | Length-of-Stay Prediction | | Mortality Prediction | | Readmission Prediction | |
|---|---|---|---|---|---|---|
| | **Macro F1 (%)** 95% CI | **AUROC (%)** 95% CI | **F1 (%)** 95% CI | **AUROC (%)** 95% CI | **F1 (%)** 95% CI | **AUROC (%)** 95% CI |
| **Majority** | 29.56 (29.56, 29.56) | 50.0 (50.0, 50.0) | 0.0 (0.0, 0.0) | 50.0 (50.0, 50.0) | 0.0 (0.0, 0.0) | 50.0 (50.0, 50.0) |
| **Minority** | 4.56 (4.56, 4.56) | 50.0 (50.0, 50.0) | 6.7 (6.7, 6.7) | 50.0 (50.0, 50.0) | 24.92 (24.92, 24.92) | 50.0 (50.0, 50.0) |
| *Traditional ML Models with 10% of Original Training Set from MIMIC-IV* | | | | | | |
| **XGBoost** | 56.69 (56.62, 56.76) | 92.01 (91.89, 92.13) | 33.33 (32.75, 33.91) | 93.82 (93.65, 93.99) | 25.09 (24.98, 25.2) | 55.63 (55.37, 55.89) |
| **LR** | 58.49 (58.28, 58.7) | 92.3 (92.15, 92.45) | 40.54 (39.92, 41.16) | 96.02 (95.93, 96.11) | 24.6 (24.43, 24.77) | 55.13 (54.89, 55.37) |
| **DecisionTree** | 49.88 (49.5, 50.26) | 80.31 (80.06, 80.56) | 29.06 (28.42, 29.7) | 80.44 (79.97, 80.91) | 24.79 (24.64, 24.94) | 52.8 (52.63, 52.97) |
| **RandomForest** | 57.15 (56.99, 57.31) | 91.72 (91.58, 91.86) | 36.3 (35.56, 37.04) | 94.94 (94.78, 95.1) | 25.57 (25.42, 25.72) | 56.15 (55.93, 56.37) |
| **AdaBoost** | 53.7 (53.58, 53.82) | 90.74 (90.61, 90.87) | 41.85 (41.52, 42.18) | 94.61 (94.54, 94.68) | 23.5 (23.26, 23.74) | 53.42 (53.24, 53.6) |
| **SVM** | 62.19 (62.03, 62.35) | 94.25 (94.17, 94.33) | 37.97 (37.45, 38.49) | 95.59 (95.46, 95.72) | 24.91 (24.67, 25.15) | 54.26 (53.82, 54.7) |
| **NaiveBayes** | 36.28 (36.2, 36.36) | 71.22 (71.14, 71.3) | 13.66 (13.58, 13.74) | 75.26 (75.19, 75.33) | 23.85 (23.69, 24.01) | 53.29 (53.2, 53.38) |
| **KNN** | 41.1 (40.78, 41.42) | 71.1 (70.69, 71.51) | 34.55 (34.27, 34.83) | 84.03 (83.7, 84.36) | 22.67 (22.46, 22.88) | 53.06 (52.84, 53.28) |
| **MLP** | 54.22 (54.08, 54.36) | 86.64 (86.61, 86.67) | 35.58 (35.21, 35.95) | 95.34 (95.26, 95.42) | 24.0 (23.91, 24.09) | 54.24 (54.08, 54.4) |
| **Transformer** | 55.25 (55.14, 55.36) | 90.8 (90.66, 90.94) | 32.36 (31.5, 33.22) | 95.32 (95.24, 95.4) | 23.85 (23.74, 23.96) | 54.06 (53.85, 54.27) |
| **RNN** | 59.13 (59.04, 59.22) | 88.22 (88.09, 88.35) | 29.23 (28.87, 29.59) | 94.99 (94.92, 95.06) | 24.5 (24.34, 24.66) | 55.15 (54.92, 55.38) |
| *Traditional ML Models with 5% of Original Training Set from MIMIC-IV* | | | | | | |
| **XGBoost** | 54.57 (54.51, 54.63) | 90.43 (90.22, 90.64) | 27.26 (26.95, 27.57) | 90.19 (90.11, 90.27) | 24.48 (24.27, 24.69) | 54.31 (53.9, 54.72) |
| **LR** | 57.82 (57.67, 57.97) | 92.01 (91.88, 92.14) | 36.06 (35.64, 36.48) | 94.7 (94.63, 94.77) | 24.16 (23.92, 24.4) | 54.2 (53.78, 54.62) |
| **DecisionTree** | 49.19 (48.83, 49.55) | 74.38 (73.38, 75.38) | 25.61 (24.95, 26.27) | 79.92 (79.53, 80.31) | 21.38 (20.78, 21.98) | 52.16 (51.89, 52.43) |
| **RandomForest** | 57.24 (57.06, 57.42) | 91.38 (91.18, 91.58) | 37.01 (36.3, 37.72) | 94.14 (93.97, 94.31) | 25.47 (25.28, 25.66) | 54.89 (54.46, 55.32) |
| **AdaBoost** | 49.63 (49.4, 49.86) | 90.36 (90.21, 90.51) | 36.65 (35.85, 37.45) | 93.11 (92.94, 93.28) | 24.68 (24.51, 24.85) | 53.55 (53.27, 53.83) |
| **SVM** | 60.81 (60.55, 61.07) | 93.21 (93.07, 93.35) | 33.91 (33.46, 34.36) | 93.6 (93.44, 93.76) | 24.93 (24.67, 25.19) | 52.22 (51.71, 52.73) |
| **NaiveBayes** | 34.42 (34.32, 34.52) | 70.08 (69.74, 70.42) | 12.74 (12.58, 12.9) | 73.99 (73.69, 74.29) | 23.12 (23.0, 23.24) | 52.13 (52.0, 52.26) |
| **KNN** | 38.38 (37.98, 38.78) | 67.31 (66.88, 67.74) | 23.2 (22.69, 23.71) | 78.08 (77.6, 78.56) | 22.3 (22.0, 22.6) | 52.66 (52.42, 52.9) |
| **MLP** | 51.86 (51.69, 52.03) | 83.12 (82.98, 83.26) | 33.01 (32.68, 33.34) | 94.05 (93.94, 94.16) | 23.93 (23.75, 24.11) | 53.48 (53.14, 53.82) |
| **Transformer** | 53.19 (52.98, 53.4) | 89.04 (88.94, 89.14) | 24.19 (23.95, 24.43) | 92.4 (92.23, 92.57) | 23.83 (23.67, 23.99) | 53.34 (53.0, 53.68) |
| **RNN** | 46.33 (46.02, 46.64) | 90.15 (89.92, 90.38) | 12.49 (11.97, 13.01) | 89.77 (89.35, 90.19) | 25.14 (24.95, 25.33) | 54.8 (54.39, 55.21) |

Table 8: **Performance of Traditional ML Models on Typical Clinical Prediction Tasks with Different Scales of Training Set**. Experiments are conducted in **MIMIC-IV** dataset. The training set is obtained through stratified sampling from the original training set in Table 1. The numbers in bracket are the ranges of performance with **95% Confidence Interval (95% CI)**, which are calculated with the 5-run scores.

## B.3 RESULTS OF LLMS WITH DIFFERENT TEMPERATURES OF DECODING

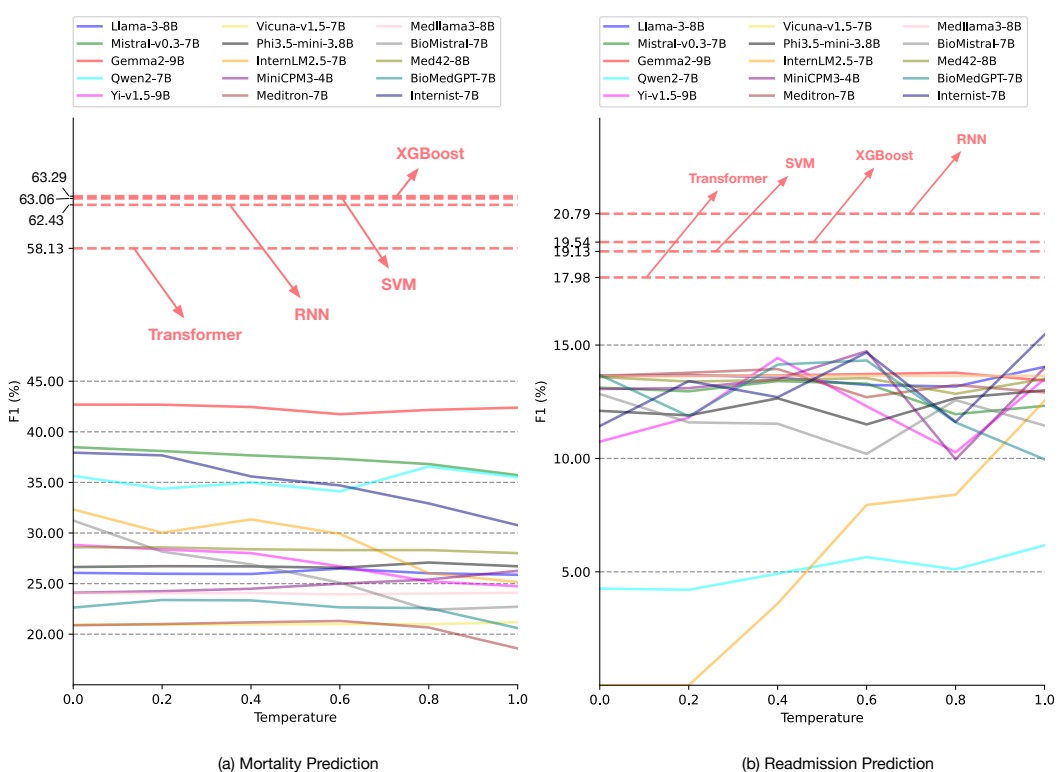

(a) Mortality Prediction
(b) Readmission Prediction

Figure 5: **Performance Comparison Between LLMs with Different Temperatures and Traditional ML Models on Mortality Prediction and Readmission Prediction**. Experiments are conducted in MIMIC-III dataset.

B.4 CONFUSION MATRIX OF TRADITIONAL ML MODELS AND DIRECTLY PROMPTING LLMS

Figure 6: **Confusion Matrix of Traditional ML Models and Directly Prompting LLMs for Length-of-Stay Prediction on MIMIC-III Dataset**.

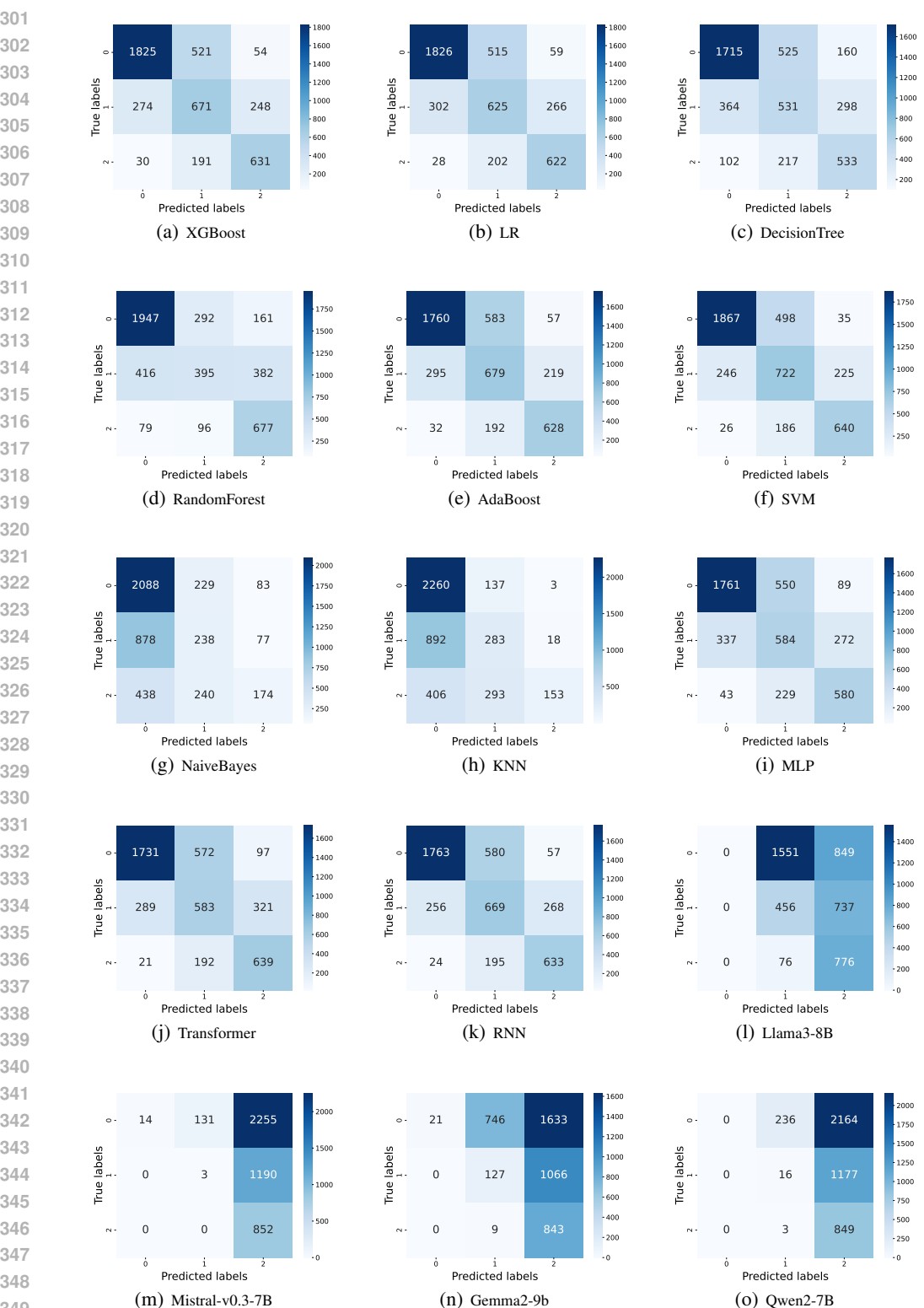

Figure 7: **Confusion Matrix of Traditional ML Models and Directly Prompting LLMs for Length-of-Stay Prediction on MIMIC-III Dataset**.

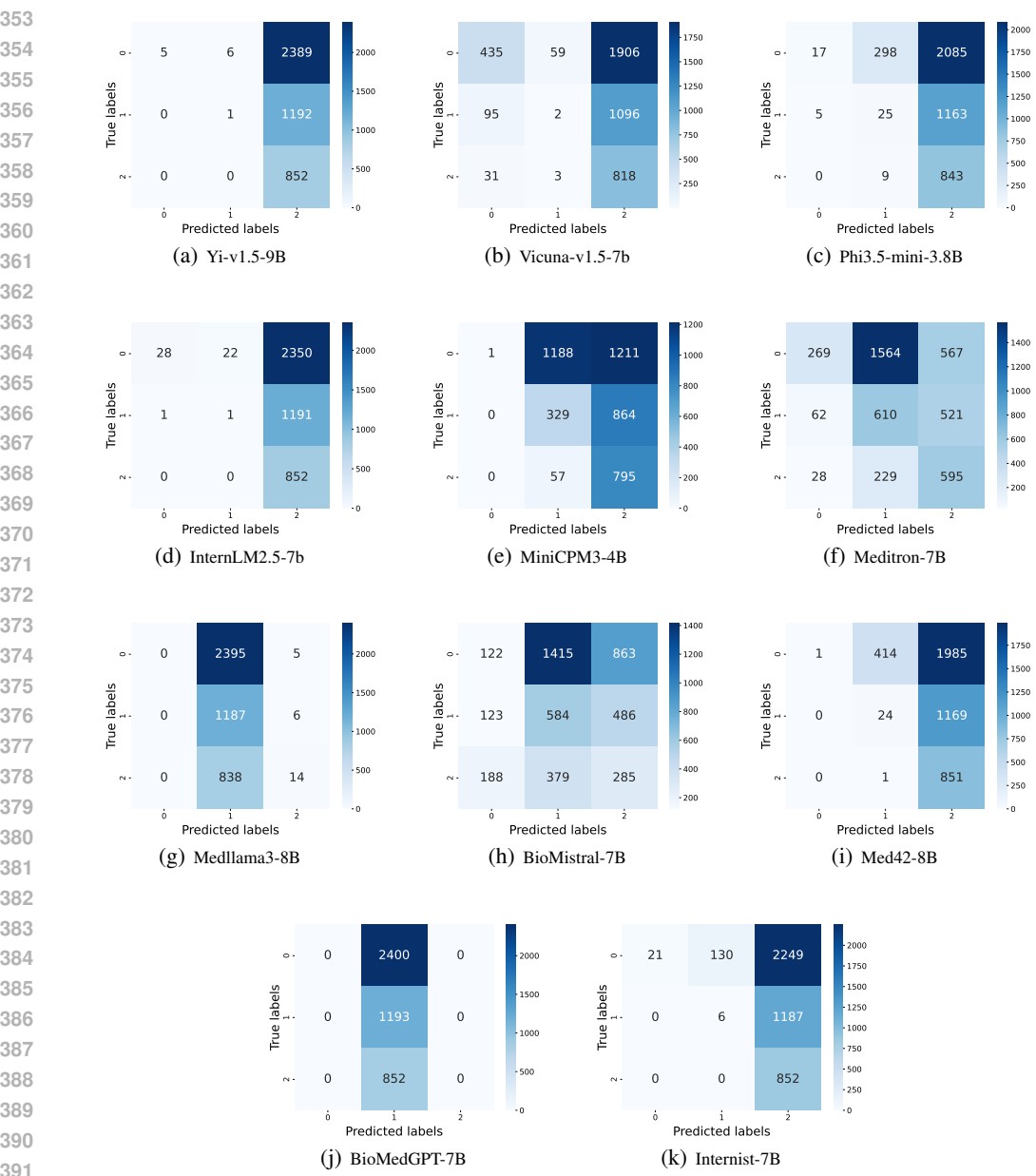

Figure 8: **Confusion Matrix of Traditional ML Models and Directly Prompting LLMs for Mortality Prediction on MIMIC-III Dataset**.

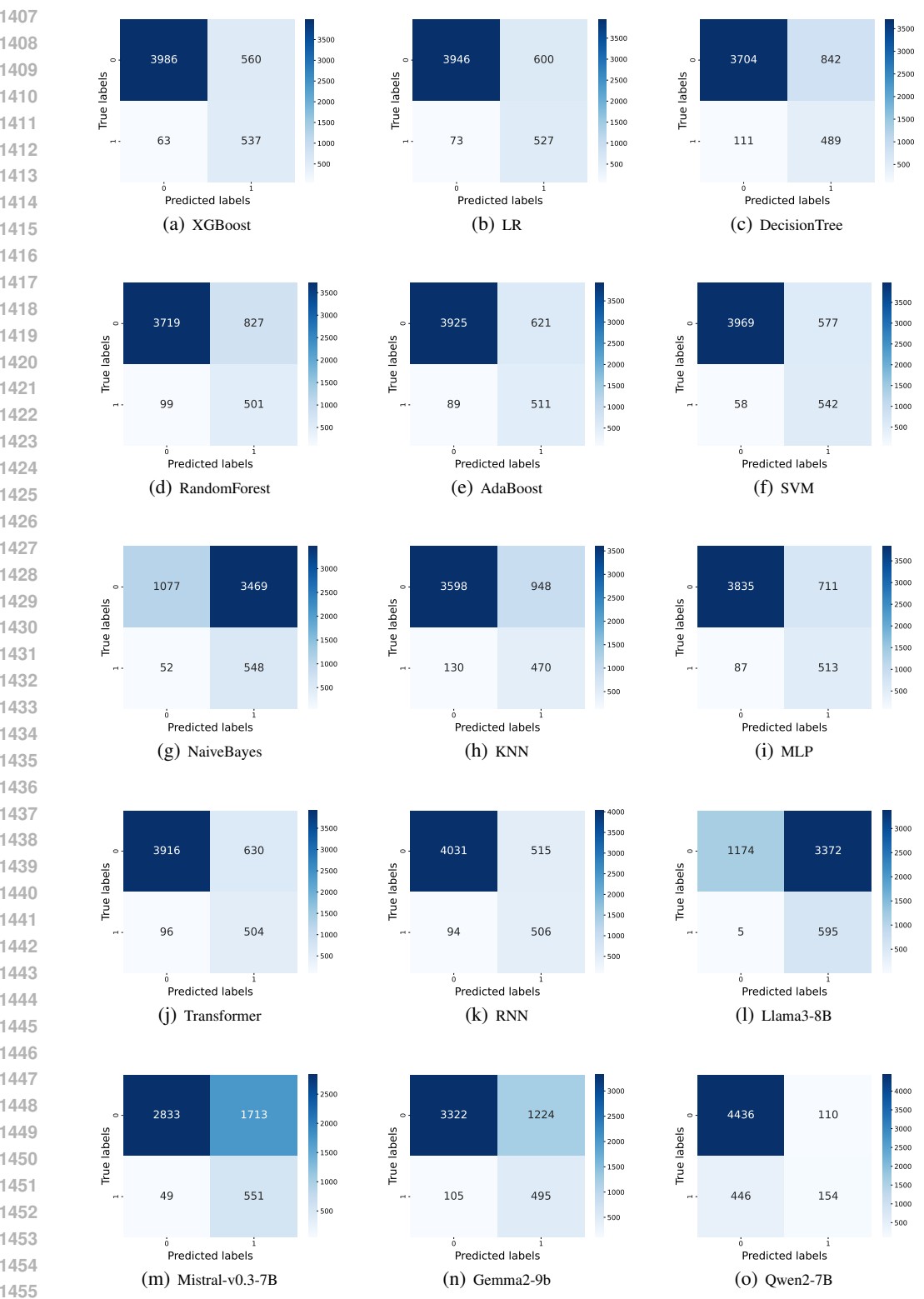

Figure 9: **Confusion Matrix of Traditional ML Models and Directly Prompting LLMs for Mortality Prediction on MIMIC-III Dataset**.

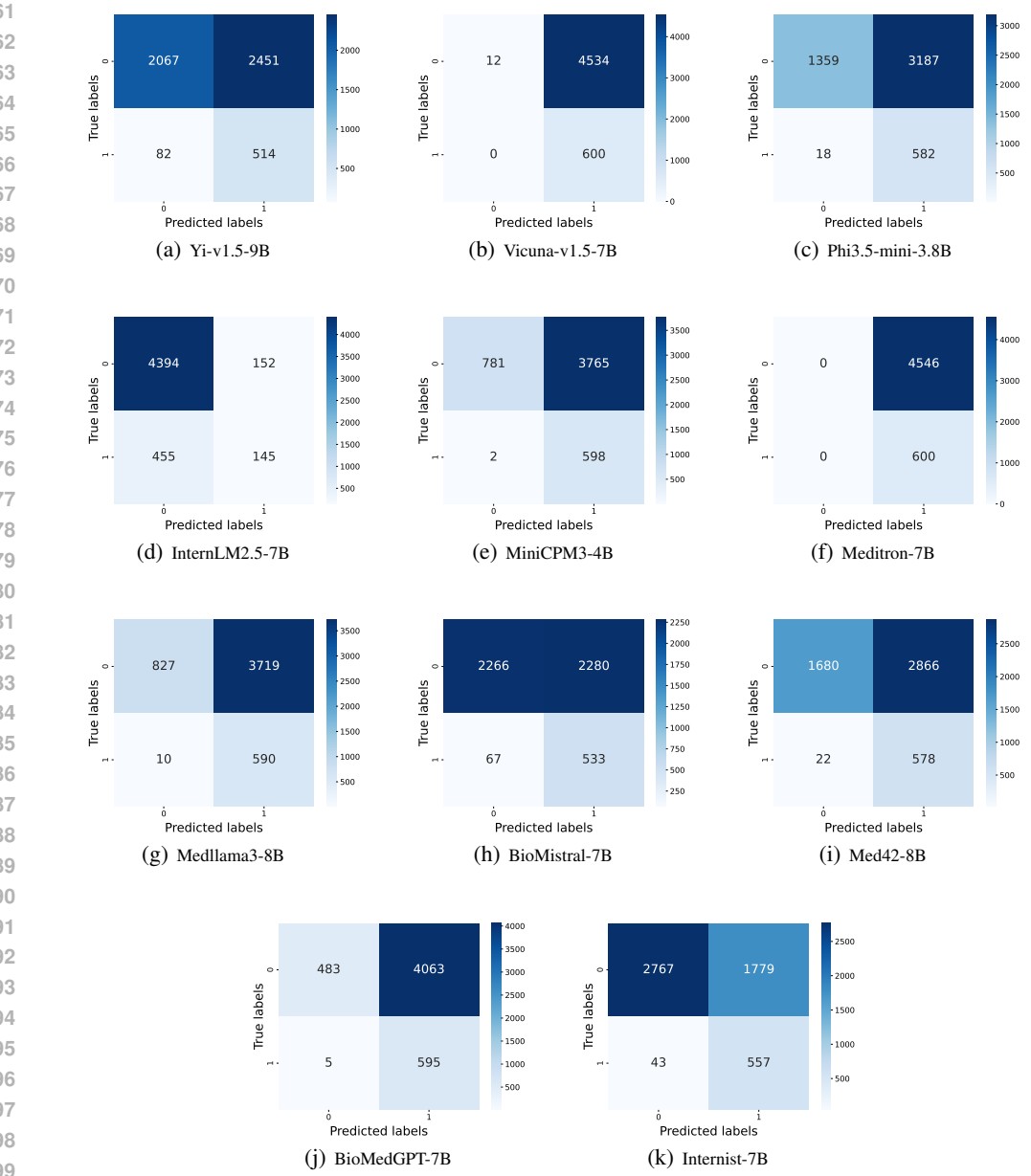

Figure 10: **Confusion Matrix of Traditional ML Models and Directly Prompting LLMs for Readmission Prediction on MIMIC-III Dataset**.

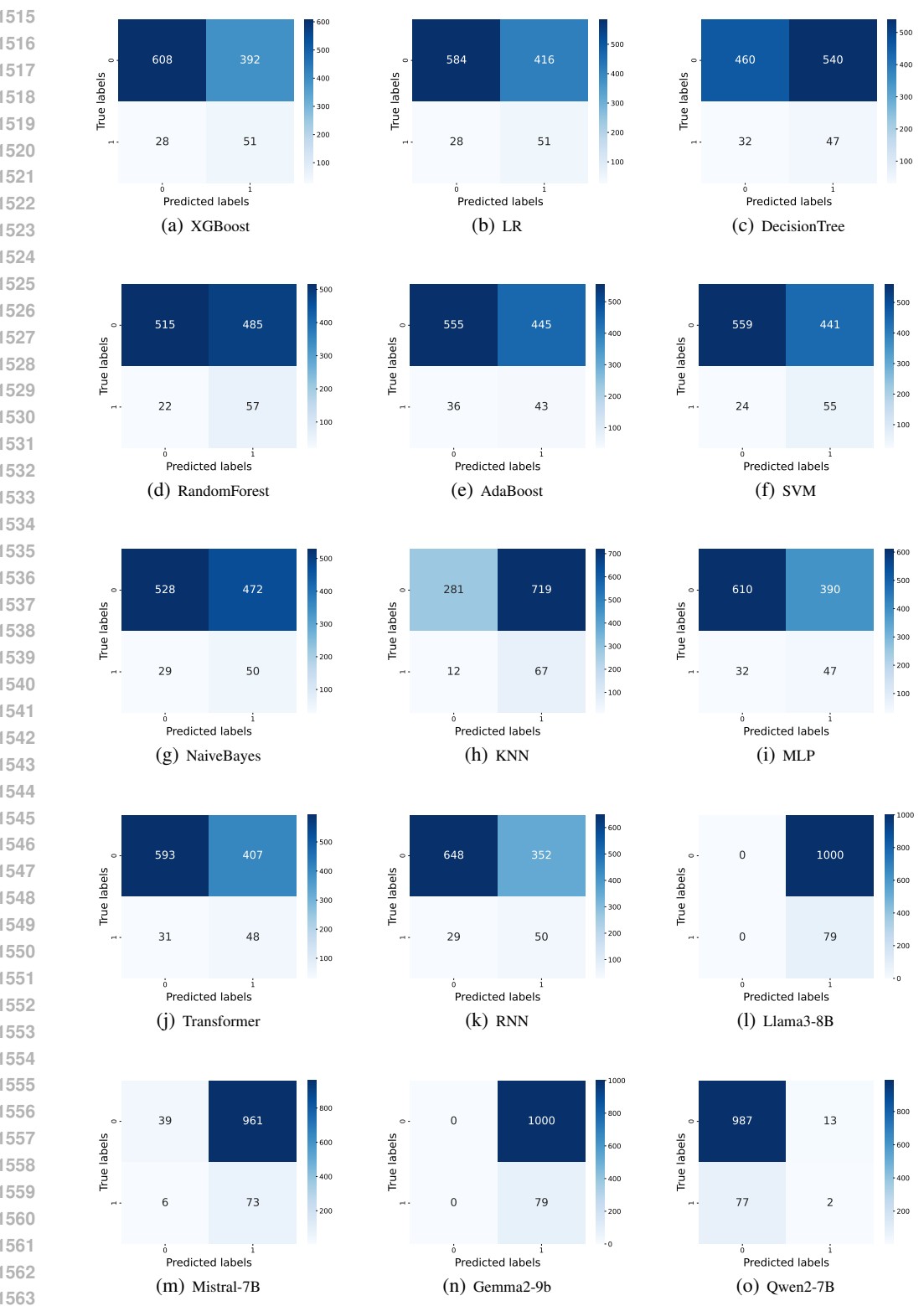

Figure 11: **Confusion Matrix of Traditional ML Models and Directly Prompting LLMs for Readmission Prediction on MIMIC-III Dataset**.

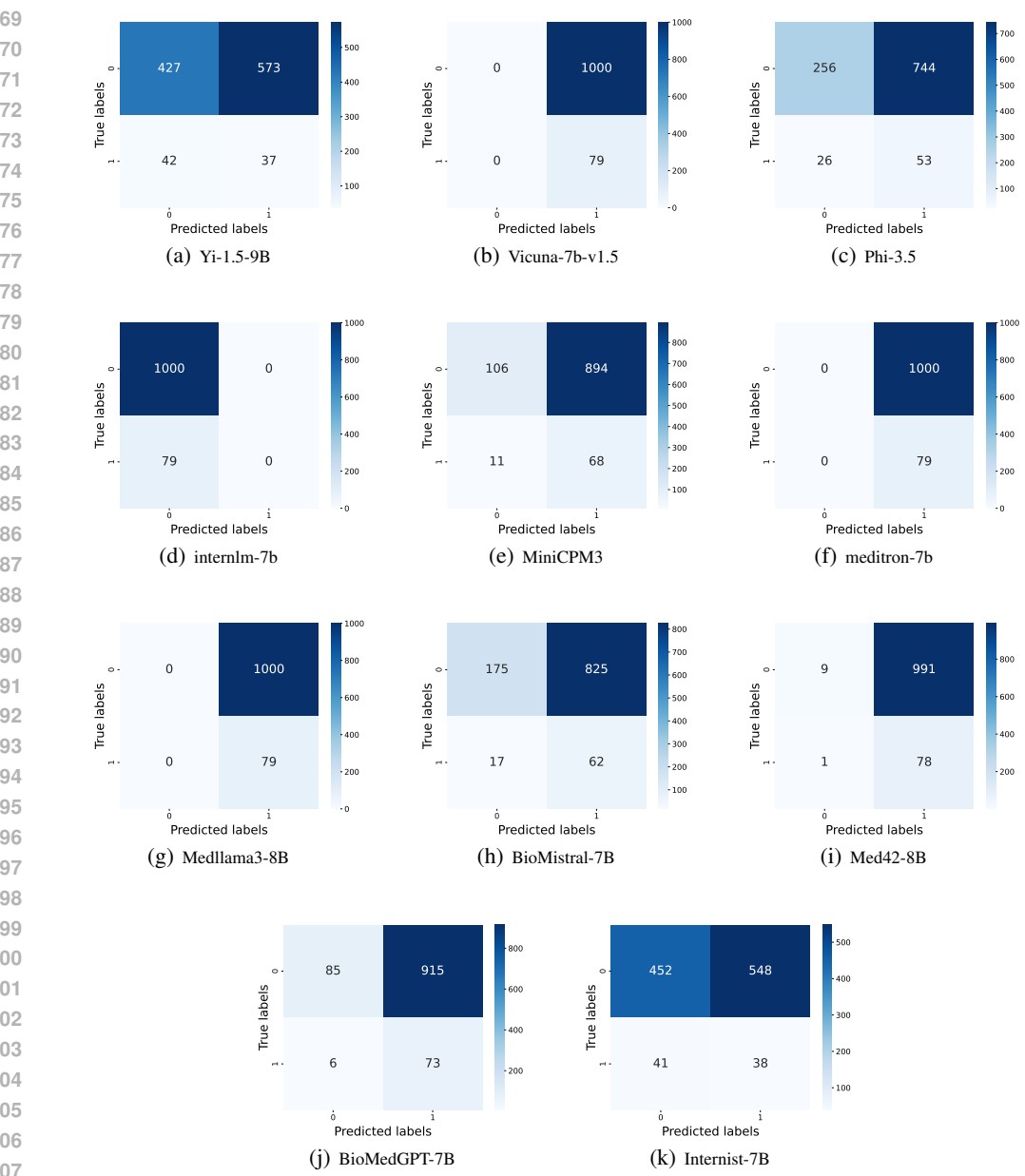

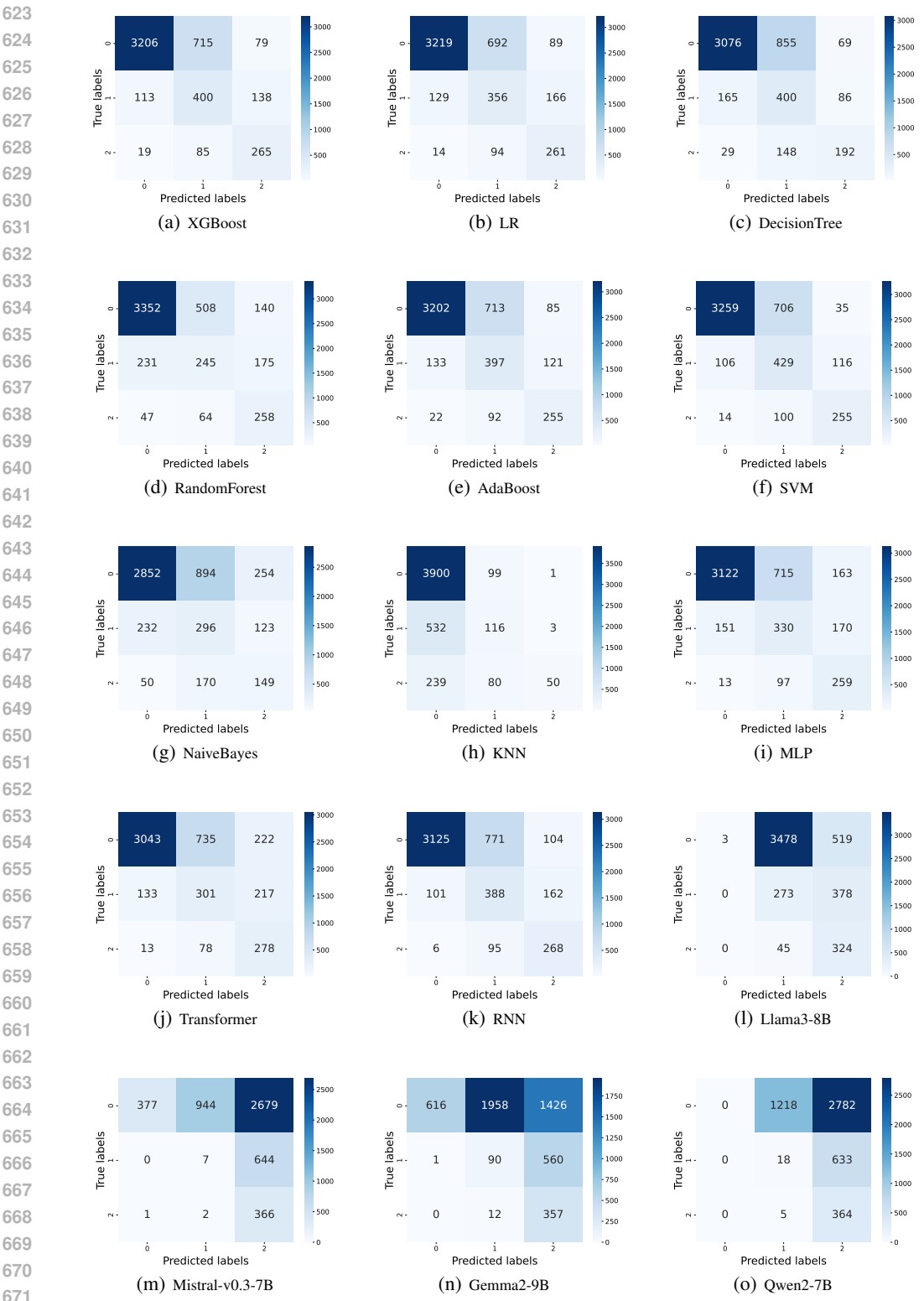

Figure 12: **Confusion Matrix of Traditional ML Models and Directly Prompting LLMs for Length-of-Stay Prediction on MIMIC-IV Dataset**.

Figure 13: **Confusion Matrix of Traditional ML Models and Directly Prompting LLMs for Length-of-Stay Prediction on MIMIC-IV Dataset**.

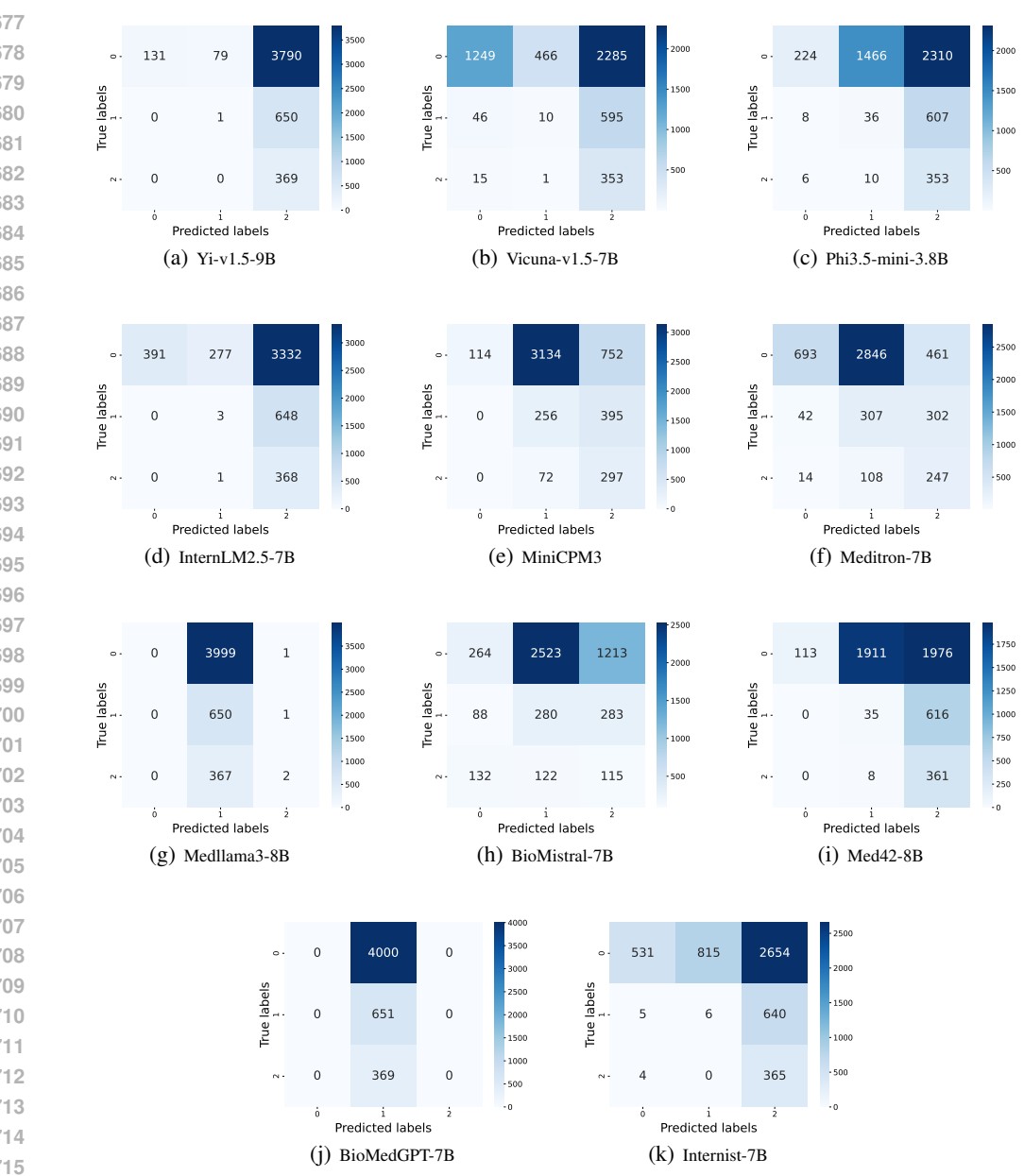

Figure 14: **Confusion Matrix of Traditional ML Models and Directly Prompting LLMs for Mortality Prediction on MIMIC-IV Dataset**.

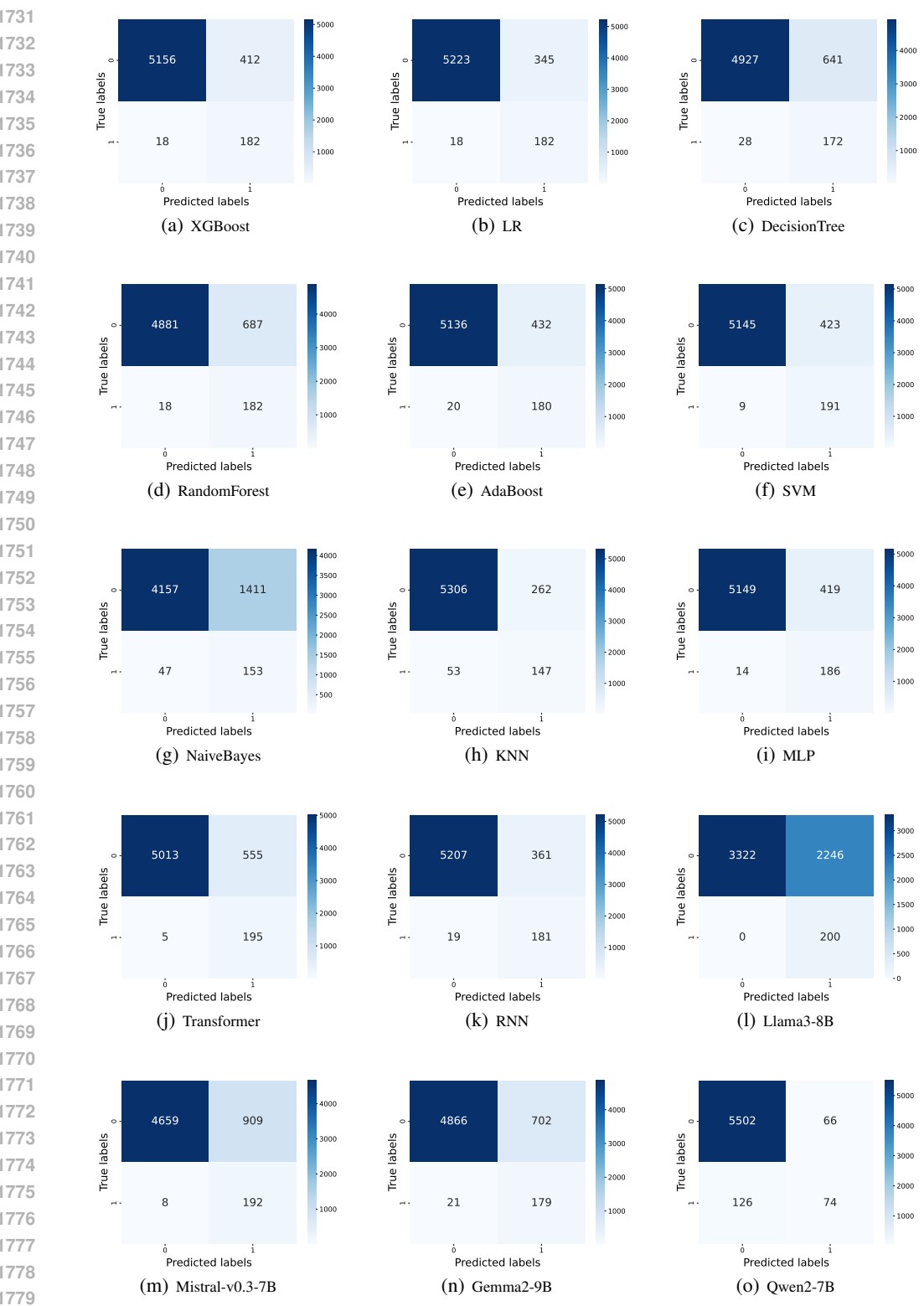

Figure 15: **Confusion Matrix of Traditional ML Models and Directly Prompting LLMs for Mortality Prediction on MIMIC-IV Dataset**.

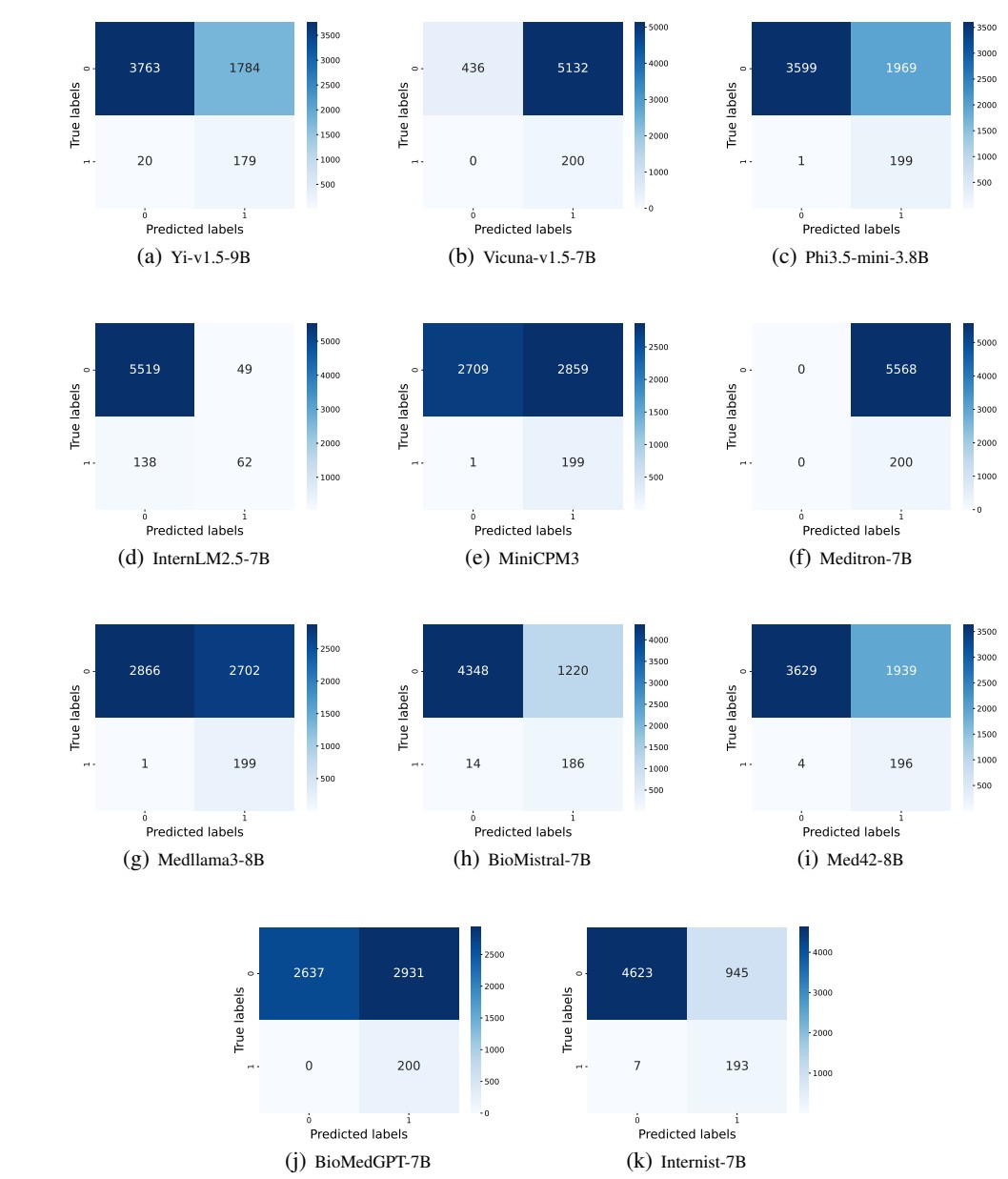

Figure 16: **Confusion Matrix of Traditional ML Models and Directly Prompting LLMs for Readmission Prediction on MIMIC-IV Dataset**.

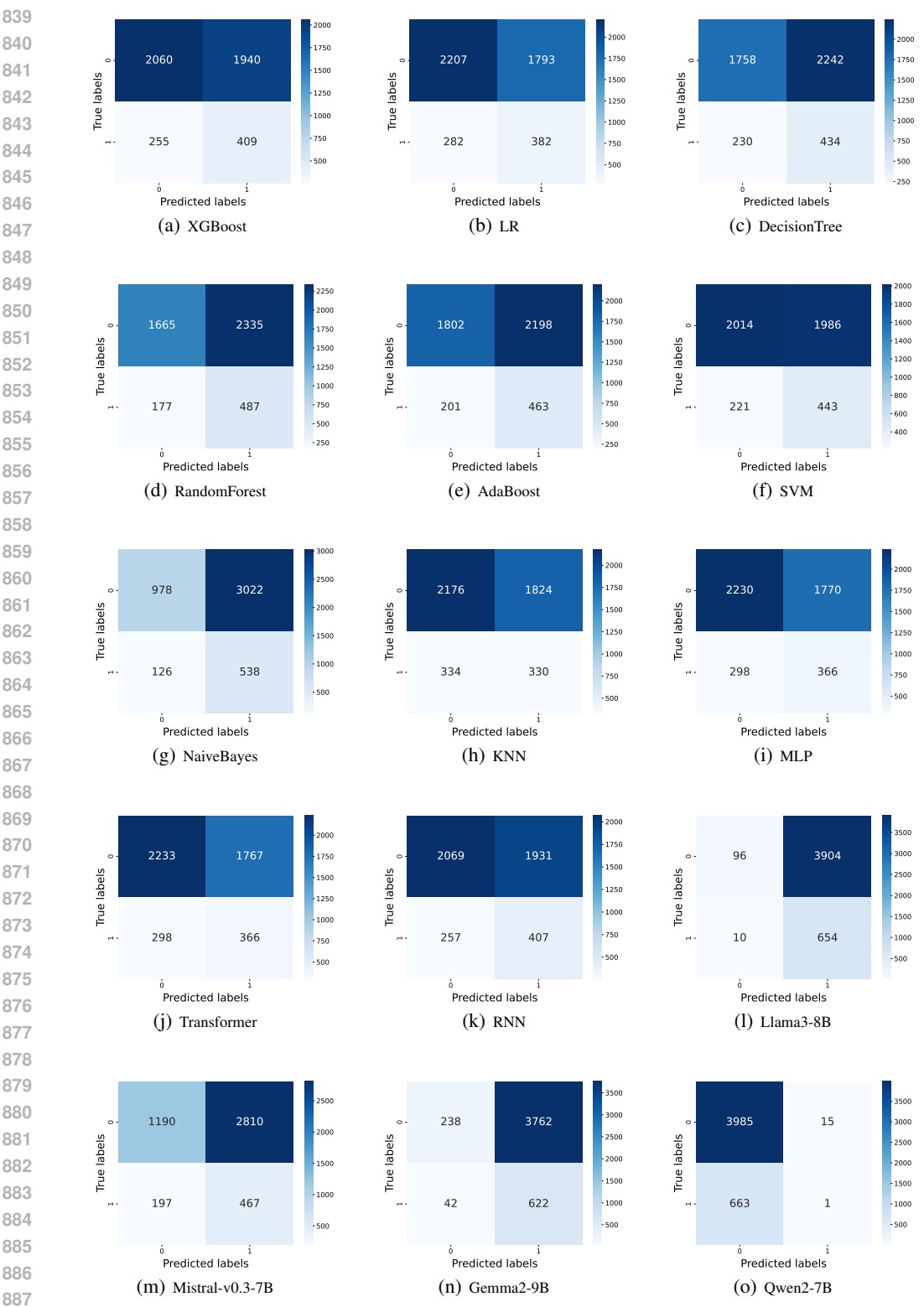

Figure 17: **Confusion Matrix of Traditional ML Models and Directly Prompting LLMs for Readmission Prediction on MIMIC-IV Dataset**.

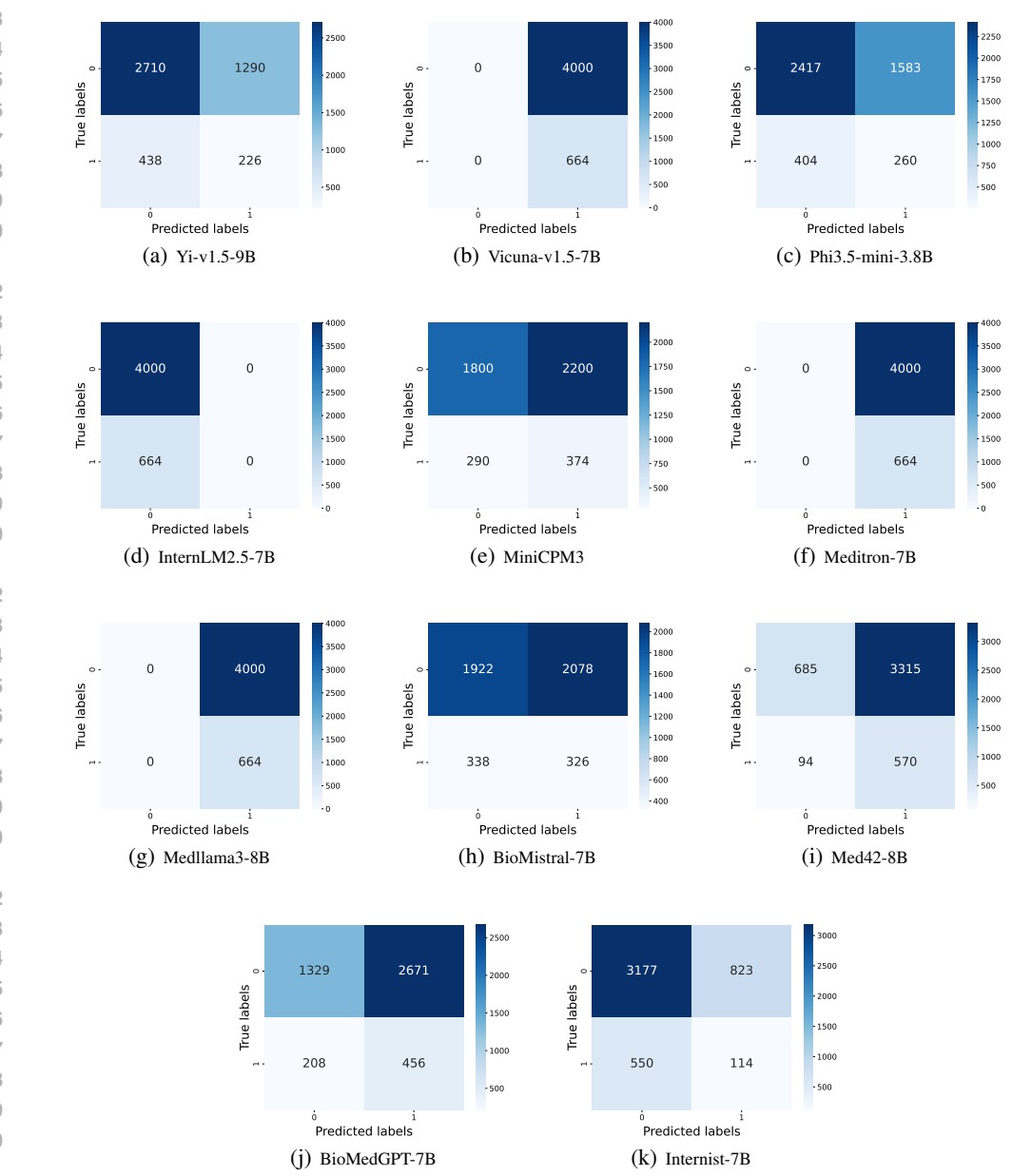

## B.5 LOSS CURVES OF FINE-TUNING LLMS

Figure 18: **Loss Curves of LoRA (full) for Length-of-Stay Prediction on MIMIC-III**.

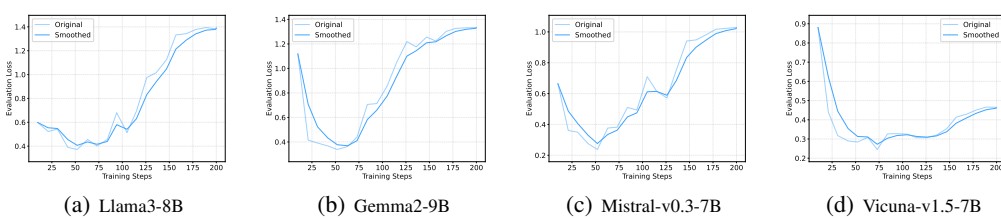

    (a) Llama3-8B           (b) Gemma2-9B          (c) Mistral-v0.3-7B       (d) Vicuna-v1.5-7B

Figure 19: **Loss Curves of LoRA (last layer) for Length-of-Stay Prediction on MIMIC-III**.

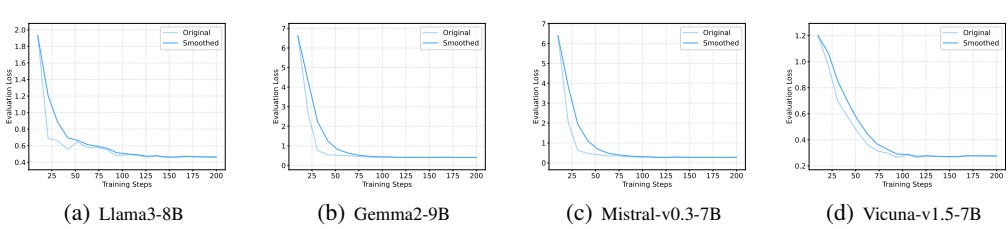

    (a) Llama3-8B           (b) Gemma2-9B          (c) Mistral-v0.3-7B       (d) Vicuna-v1.5-7B

Figure 20: **Loss Curves of LoRA (full) for Mortality Prediction on MIMIC-III**.

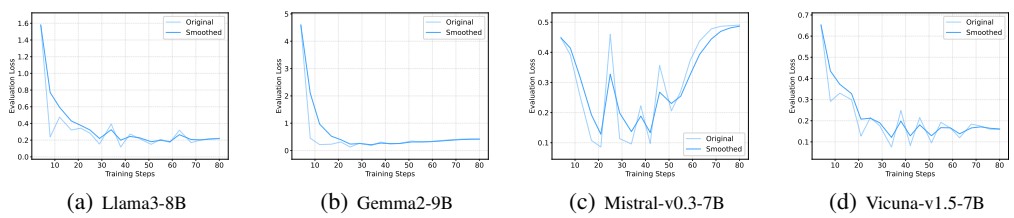

    (a) Llama3-8B           (b) Gemma2-9B          (c) Mistral-v0.3-7B       (d) Vicuna-v1.5-7B

Figure 21: **Loss Curves of LoRA (last layer) for Mortality Prediction on MIMIC-III**.

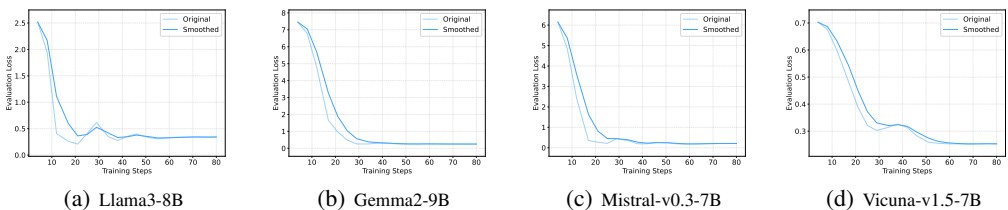

    (a) Llama3-8B           (b) Gemma2-9B          (c) Mistral-v0.3-7B       (d) Vicuna-v1.5-7B

Figure 22: **Loss Curves of LoRA (full) for Readmission Prediction on MIMIC-III**.

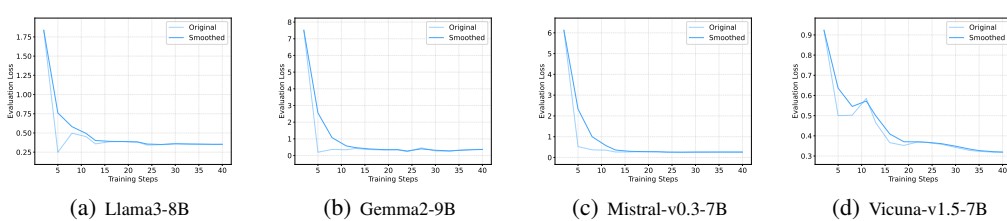

(a) Llama3-8B  (b) Gemma2-9B  (c) Mistral-v0.3-7B  (d) Vicuna-v1.5-7B

Figure 23: **Loss Curves of LoRA (last layer) for Readmission Prediction on MIMIC-III**.

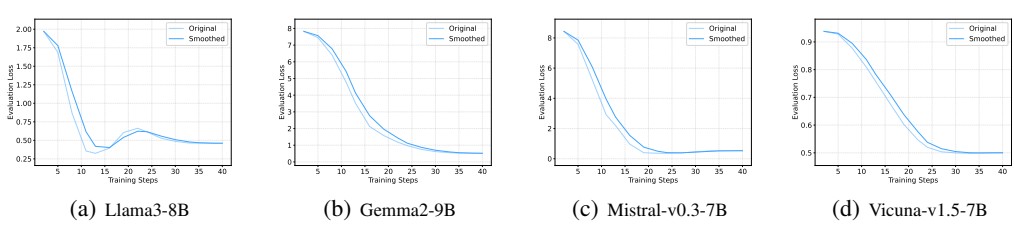

(a) Llama3-8B  (b) Gemma2-9B  (c) Mistral-v0.3-7B  (d) Vicuna-v1.5-7B

Figure 24: **Loss Curves of LoRA (full) for Length-of-Stay Prediction on MIMIC-IV**.

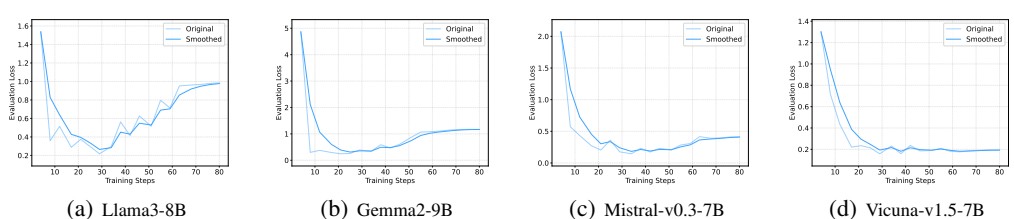

(a) Llama3-8B  (b) Gemma2-9B  (c) Mistral-v0.3-7B  (d) Vicuna-v1.5-7B

Figure 25: **Loss Curves of LoRA (last layer) for Length-of-Stay Prediction on MIMIC-IV**.

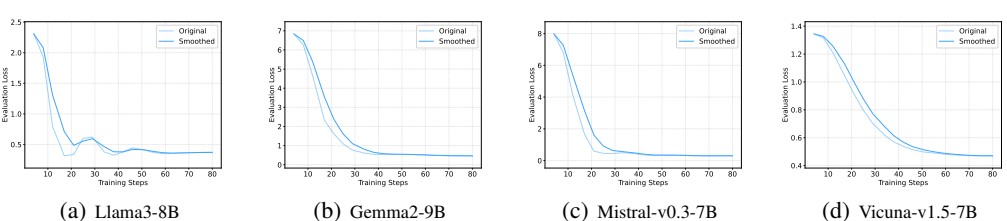

(a) Llama3-8B  (b) Gemma2-9B  (c) Mistral-v0.3-7B  (d) Vicuna-v1.5-7B

Figure 26: **Loss Curves of LoRA (full) for Mortality Prediction on MIMIC-IV**.

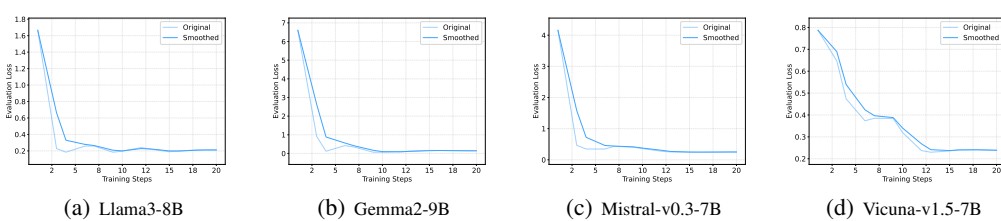

(a) Llama3-8B      (b) Gemma2-9B      (c) Mistral-v0.3-7B      (d) Vicuna-v1.5-7B

Figure 27: **Loss Curves of LoRA (last layer) for Mortality Prediction on MIMIC-IV**.

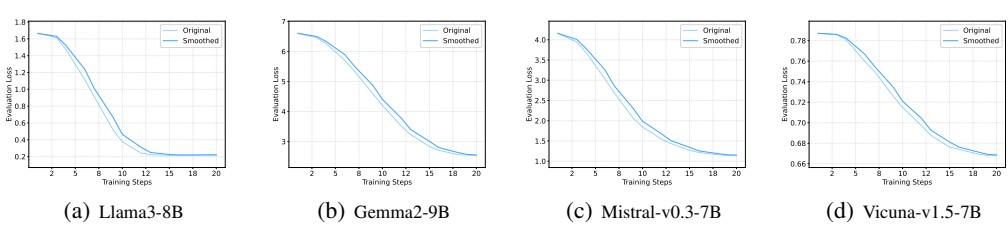

(a) Llama3-8B      (b) Gemma2-9B      (c) Mistral-v0.3-7B      (d) Vicuna-v1.5-7B

Figure 28: **Loss Curves of LoRA (full) for Readmission Prediction on MIMIC-IV**.

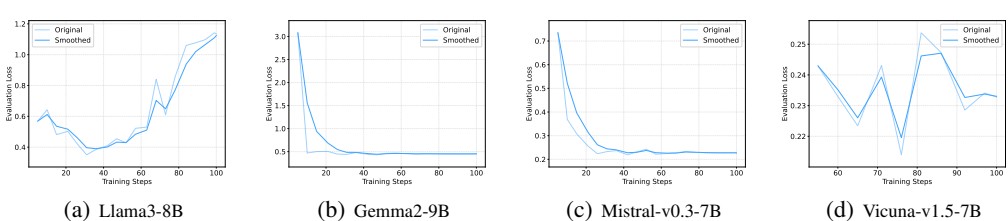

(a) Llama3-8B      (b) Gemma2-9B      (c) Mistral-v0.3-7B      (d) Vicuna-v1.5-7B

Figure 29: **Loss Curves of LoRA (last layer) for Readmission Prediction on MIMIC-IV**.

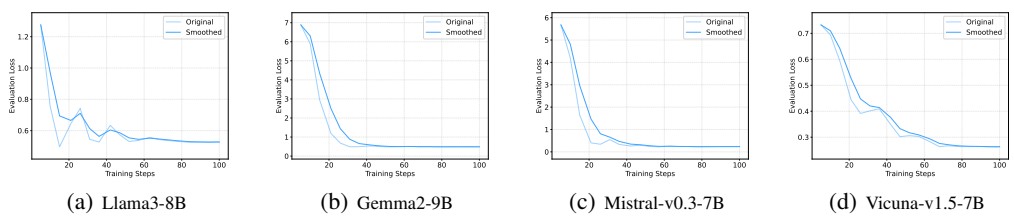

(a) Llama3-8B      (b) Gemma2-9B      (c) Mistral-v0.3-7B      (d) Vicuna-v1.5-7B

# C  MORE DETAILS OF FINE-TUNING DATA CONSTRUCTION FOR LLMS

Table 9: Example of Fine-Tuning Data Construction for Length-of-Stay Prediction

| |
| --- |
| *Instruction:*
Given the patient information, predict the number of weeks of stay in hospital.
Answer 1 if no more than one week,
Answer 2 if more than one week but not more than two weeks,
Answer 3 if more than two weeks.
Answer with only the number |
| *Input:*
Patient information:
Age: 78
Gender: male
Conditions: Coronary atherosclerosis of native coronary artery, Chronic total occlusion of coronary artery, Malignant neoplasm of rectum, Personal history of antineoplastic chemotherapy, Person boarding or alighting a pedal cycle injured in collision with railway train or railway vehicle
Procedures: Percutaneous transluminal coronary angioplasty [PTCA], Insertion of non-drug-eluting coronary artery stent(s), Left heart cardiac catheterization, Coronary arteriography using two catheters, Insertion of two vascular stents, Procedure on two vessels
Using Drugs: other analgesics and antipyretics, lipid modifying agents, plain, antithrombotic agents, i.v. solution additives, antidepressants, potassium, viral vaccines, vasodilators used in cardiac diseases, belladonna and derivatives, plain
Answer: |
| *Output:*
1 |

Table 10: Example of Fine-Tuning Data Construction for Mortality Prediction

| |
| --- |
| *Instruction:*
Given the patient information, predict the mortality of the patient.
Answer 1 if the patient will die, answer 0 otherwise.
Answer with only the number |
| *Input:*
Patient information:
Age: 64
Gender: female
Conditions: Osteoarthrosis, localized, not specified whether primary or secondary, lower leg, Other and unspecified angina pectoris, Unspecified essential hypertension, Mitral valve disorders, Psoriatic arthropathy
Procedures: Total knee replacement
Using Drugs: selective calcium channel blockers with mainly vascular effects, beta blocking agents, lipid modifying agents, plain, vitamin a and d, incl. combinations of the two, antihistamines for systemic use, other nutrients, macrolides, lincosamides and streptogramins, anticholinergic agents, drugs for peptic ulcer and gastro-oesophageal reflux disease (gord), drugs for constipation, opioids, antidepressants, other analgesics and antipyretics, i.v. solutions, antivaricose therapy, antiemetics and antinauseants, antiinflammatory and antirheumatic products, non-steroids, antacids, calcium, angiotensin ii receptor blockers (arbs), plain, antithrombotic agents
Will the patient die because of the above situation?
Answer: |
| *Output:*
0 |

Table 11: Example of Fine-Tuning Data Construction for Readmission Prediction

*Instruction:*
Given the patient information, predict the readmission of the patient.
Answer 1 if the patient will be readmitted to the hospital within two weeks, answer 0 otherwise.
Answer with only the number

*Input:*
Patient information:
Age: 91
Gender: female
Conditions: Open wound of forehead, without mention of complication, Hyposmolality and/or hyponatremia, Open wound of knee, leg [except thigh], and ankle, without mention of complication, Contusion of thigh, Contusion of scapular region, Open wound of scalp, without mention of complication, Unspecified essential hypertension, Insomnia, unspecified, Unspecified deficiency anemia, Acquired absence of both cervix and uterus, Acquired absence of intestine (large) (small), Personal history of malignant melanoma of skin, Fall from other slipping, tripping, or stumbling
Procedures: Closure of skin and subcutaneous tissue of other sites
Using Drugs: selective calcium channel blockers with mainly vascular effects, vitamin a and d, incl. combinations of the two, antidepressants, other analgesics and antipyretics, beta blocking agents, drugs for constipation, antithrombotic agents, antacids, calcium, i.v. solution additives
Will the patient be readmitted to the hospital within two weeks?
Answer:

*Output:*
0

# D EXAMPLES OF LLM-BASED CLINICAL PREDICTION

## D.1 LENGTH-OF-STAY PREDICTION

### D.1.1 DIRECTLY PROMPTING

Table 12: Example of LLMs with Directly Prompting for Length-of-Stay Prediction.

---

*Input:*
Patient information:
Age: 62
Gender: male
Conditions: Streptococcal septicemia, Pressure ulcer, lower back, Acute osteomyelitis, pelvic region and thigh, Unknown Diagnosis, Cocaine abuse, unspecified, Acute kidney failure, unspecified, Hemorrhage complicating a procedure, Hyperpotassemia, Chronic osteomyelitis, pelvic region and thigh, Sepsis, Paraplegia, Unspecified essential hypertension, Colostomy status, Other artificial opening of urinary tract status, Anemia of other chronic disease, Other iatrogenic hypotension, Candidiasis of other urogenital sites, Late effects of injury purposely inflicted by other person
Procedures: Local excision of lesion or tissue of bone, other bones, Local excision of lesion or tissue of bone, femur, Nonexcisional debridement of wound, infection or burn, Venous catheterization, not elsewhere classified, Transfusion of packed cells
Using Drugs: anesthetics, general, i.v. solution additives, other antibacterials, anterior pituitary lobe hormones and analogues, antithrombotic agents, other analgesics and antipyretics, drugs for constipation, drugs for peptic ulcer and gastro-oesophageal reflux disease (gord), antiseptics and disinfectants, antiinfectives and antiseptics, excl. combinations with corticosteroids, other nutrients, opioids, other beta-lactam antibacterials, other mineral supplements, other diagnostic agents, antipsychotics, i.v. solutions, antivaricose therapy, beta-lactam antibacterials, penicillins
Predict the number of weeks of stay in hospital.
Answer 1 if no more than one week,
Answer 2 if more than one week but not more than two weeks,
Answer 3 if more than two weeks.
Answer with only the number. Answer:

---

*Output for Llama3-8B:*
3

- - - - - - - - - - - - - - - - - - - - - - - - - - - - - - - - - - - - - - - -

*Output for Gemma2-9B:*
3

- - - - - - - - - - - - - - - - - - - - - - - - - - - - - - - - - - - - - - - -

*Output for Meditron-7B:*
3

- - - - - - - - - - - - - - - - - - - - - - - - - - - - - - - - - - - - - - - -

*Output for Internist-7B:*
3

### D.1.2 CHAIN-OF-THOUGHT PROMPTING

Table 13: Example of LLMs with Chain of Thought for Length-of-Stay Prediction.

*Input:*
Patient information:
Age: 73
Gender: female
Conditions: Closed fracture of surgical neck of humerus, Other emphysema, Closed fracture of other facial bones, Closed fracture of malar and maxillary bones, Hematoma complicating a procedure, Iatrogenic pulmonary embolism and infarction, Acute posthemorrhagic anemia, Pneumonia, organism unspecified, Fall from other slipping, tripping, or stumbling, Diabetes mellitus without mention of complication, type II or unspecified type, not stated as uncontrolled, Other specified surgical operations and procedures causing abnormal patient reaction, or later complication, without mention of misadventure at time of operation, Personal history of other diseases of circulatory system, Other and unspecified hyperlipidemia, Unspecified essential hypertension
Procedures: Partial shoulder replacement, Transfusion of packed cells
Using Drugs: i.v. solution additives, opioids, selective calcium channel blockers with mainly vascular effects, beta blocking agents, high-ceiling diuretics, antithrombotic agents, lipid modifying agents, plain, insulins and analogues, blood glucose lowering drugs, excl. insulins, other analgesics and antipyretics, antiemetics and antinauseants, i.v. solutions, antivaricose therapy, other beta-lactam antibacterials, antipsychotics, anxiolytics, adrenergics, inhalants, other drugs for obstructive airway diseases, inhalants, other antibacterials, corticosteroids for systemic use, plain, other diagnostic agents, vitamin b12 and folic acid, vitamin b1, plain and in combination with vitamin b6 and b12, other mineral supplements, quinolone antibacterials, antidepressants, potassium, drugs for peptic ulcer and gastro-oesophageal reflux disease (gord), drugs for constipation
Please provide your concise reasoning steps for the prediction(no more than 3 steps), and finally answer 1 if the patient will stay no more than one week, answer 2 if more than one week but not more than two weeks, answer 3 if more than two weeks.

*Output for Llama3-8B:*
Please note that the patient's condition is quite complex and the prediction may not be accurate. This is a hypothetical scenario and not a real patient.
Answer: 3
Reasoning steps:
1. The patient has a closed fracture of the surgical neck of humerus, which is a serious injury that requires surgical intervention. This suggests that the patient will require a significant amount of care and treatment in the hospital.
2. The patient also has a number of other conditions, including pneumonia, acute posthemorrhagic anemia, and iatrogenic pulmonary embolism and infarction, which are all serious and potentially life-threatening. This suggests that the patient will require ongoing medical care and monitoring in the hospital.
3. Given the complexity of the patient's condition and the need for ongoing medical care and monitoring, it is likely that the patient will require more than two weeks of hospitalization.
Note: This is a hypothetical scenario and not a real patient. The actual length of stay may vary depending on a variety of factors, including the patient's response to treatment, the severity of their condition, and the availability of hospital resources.

- - - - - - - - - - - - - - - - - - - - - - - - - - - - - - - - - - - - - - - - - - - - - - - - - - -

*Output for Gemma2-9B:*

*Continued on next page*

Table 13 – continued from previous page

**Reasoning:**
1. **Severity of Conditions:** The patient has multiple serious conditions including congestive heart failure, mitral valve disorders, atrial fibrillation, chronic kidney disease, and diabetes. These require significant medical management.
2. **Complex Procedures:** The patient underwent major surgeries like mitral valve replacement, coronary bypass, and transvenous lead insertion. Recovery from these procedures takes considerable time.
3. **Post-Operative Care:** The patient received multiple transfusions, medications for various conditions, and likely required intensive monitoring post-surgery. This indicates a prolonged hospital stay for recovery and management.
**Answer:** 3

*Output for Meditron-7B:*
"," ## Prediction
## Prediction
## Prediction
## Prediction
## Prediction
## Prediction
## Prediction
## Prediction

*Output for Internist-7B:*
Reasoning:
1. Analyze the patient's conditions and procedures: - The patient has multiple fractures and injuries, indicating a high likelihood of a prolonged hospital stay. - The patient has a history of diabetes and hyperlipidemia, which may require close monitoring and management. - The patient has undergone a partial shoulder replacement, which is a surgical procedure that typically requires a longer recovery period.
2. Consider the patient's medications and drug history: - The patient is taking multiple medications, including opioids, beta-blockers, and antithrombotic agents. - These medications may require close monitoring and adjustment during the hospital stay.
3. Evaluate the patient's overall health and risk factors: - The patient has a history of diabetes and hyperlipidemia, which may increase the risk of complications and prolong the hospital stay. - The patient has a history of other diseases of the circulatory system, which may also contribute to a longer hospital stay.
Based on the above analysis, it is likely that the patient will require a longer hospital stay. Therefore, the answer is 3, indicating a stay of more than two weeks. It is important to note that this prediction is based on the information provided and may vary depending on the specific circumstances and progress of the patient's condition. Regular monitoring and assessment by healthcare professionals will be necessary to determine the actual duration of the hospital stay.
Please note that this is a prediction based on the given information and may not be accurate in all cases. It is important to consider individual patient factors and consult with healthcare professionals for a more accurate assessment.

### D.1.3 Self-Reflection Prompting

Table 14: Example of LLMs with Self Reflection for Length-of-Stay Prediction.

---

*Input:*
Patient information:
Age: 73
Gender: female
Conditions: Closed fracture of surgical neck of humerus, Other emphysema, Closed fracture of other facial bones, Closed fracture of malar and maxillary bones, Hematoma complicating a procedure, Iatrogenic pulmonary embolism and infarction, Acute posthemorrhagic anemia, Pneumonia, organism unspecified, Fall from other slipping, tripping, or stumbling, Diabetes mellitus without mention of complication, type II or unspecified type, not stated as uncontrolled, Other specified surgical operations and procedures causing abnormal patient reaction, or later complication, without mention of misadventure at time of operation, Personal history of other diseases of circulatory system, Other and unspecified hyperlipidemia, Unspecified essential hypertension
Procedures: Partial shoulder replacement, Transfusion of packed cells
Using Drugs: i.v. solution additives, opioids, selective calcium channel blockers with mainly vascular effects, beta blocking agents, high-ceiling diuretics, antithrombotic agents, lipid modifying agents, plain, insulins and analogues, blood glucose lowering drugs, excl. insulins, other analgesics and antipyretics, antiemetics and antinauseants, i.v. solutions, antivaricose therapy, other beta-lactam antibacterials, antipsychotics, anxiolytics, adrenergics, inhalants, other drugs for obstructive airway diseases, inhalants, other antibacterials, corticosteroids for systemic use, plain, other diagnostic agents, vitamin b12 and folic acid, vitamin b1, plain and in combination with vitamin b6 and b12, other mineral supplements, quinolone antibacterials, antidepressants, potassium, drugs for peptic ulcer and gastro-oesophageal reflux disease (gord), drugs for constipation
Answer 1 if no more than one week,
Answer 2 if more than one week but not more than two weeks,
Answer 3 if more than two weeks.
First answer with a number. Then conduct a concise reflection. Finally output your answer again with a number.

---

*Output for Llama3-8B:*
Answer: 3
Reflection: The patient has a complex medical history with multiple conditions, including a closed fracture of the surgical neck of humerus, pneumonia, and iatrogenic pulmonary embolism. They have undergone a partial shoulder replacement and have received multiple medications. Given the severity of their conditions and the need for ongoing medical care, it is likely that they will require a prolonged hospital stay of more than two weeks.
Answer: 3
'''python
......

- - - - - - - - - - - - - - - - - - - - - - - - - - - - - - - - - - - - - - - - -

*Output for Gemma2-9B:*
**Answer:** 3
**Reflection:**
The patient presents with multiple severe injuries, including a closed fracture of the surgical neck of the humerus, facial bones, and a hematoma complicating a procedure. These injuries necessitate surgical intervention (partial shoulder replacement) and extensive post-operative care. The patient also has a history of diabetes, emphysema, and hypertension, which can complicate recovery. The presence of pneumonia and acute posthemorrhagic anemia further suggests a prolonged hospital stay.
**Answer:** 3

- - - - - - - - - - - - - - - - - - - - - - - - - - - - - - - - - - - - - - - - -

*Output for Meditron-7B:*

*Continued on next page*

Table 14 – continued from previous page

Answer 1
Answer 2
Answer 3
Answer 1
Answer 2
Answer 3
Answer 1
Answer 2
Answer 3
Answer 1
Answer 2
Answer 3
Answer 1
Answer 2
Answer 3

*Output for Internist-7B:*

Reasoning:To predict the number of weeks of stay in hospital, we need to analyze the patient's conditions, procedures, and medications.

1. Conditions: The patient has multiple conditions, including a closed fracture of the surgical neck of the humerus, other emphysema, closed fractures of other facial bones, closed fractures of the malar and maxillary bones, hematoma complicating a procedure, iatrogenic pulmonary embolism and infarction, acute posthemorrhagic anemia, pneumonia, and fall from other slipping, tripping, or stumbling. These conditions indicate that the patient may require a longer stay in the hospital.

2. Procedures: The patient has undergone a partial shoulder replacement and a transfusion of packed cells. These procedures may require a longer recovery period and may contribute to a longer hospital stay.

3. Medications: The patient is using various medications, including i.v. solution additives, opioids, selective calcium channel blockers with mainly vascular effects, beta blocking agents, high-ceiling diuretics, antithrombotic agents, lipid modifying agents, insulins and analogues, blood glucose lowering drugs, other analgesics and antipyretics, antiemetics and antinauseants, i.v. solutions, antivaricose therapy, other beta-lactam antibacterials, antipsychotics, anxiolytics, adrenergics, inhalants, other drugs for obstructive airway diseases, inhalants, other antibacterials, corticosteroids for systemic use, plain, other diagnostic agents, vitamin b12 and folic acid, vitamin b1, plain and in combination with vitamin b6 and b12, other mineral supplements, quinolone antibacterials, antidepressants, potassium, drugs for peptic ulcer and gastro-oesophageal reflux disease (gord), drugs for constipation. These medications may also contribute to a longer hospital stay.

Based on the patient's conditions, procedures, and medications, it is likely that the patient will require a longer stay in the hospital. Therefore, the correct answer is 3, indicating a stay of more......

### D.1.4 ROLE-PLAYING PROMPTING

Table 15: Example of LLMs with Role-Playing for Length-of-Stay Prediction.

| |
|---|
| *Input:* |
| Imagine that you are a doctor. Today, you're seeing a patient with the following profile: |
| Patient information: |
| Age: 62 |
| Gender: male |
| Conditions: Streptococcal septicemia, Pressure ulcer, lower back, Acute osteomyelitis, pelvic region and thigh, Unknown Diagnosis, Cocaine abuse, unspecified, Acute kidney failure, unspecified, Hemorrhage complicating a procedure, Hyperpotassemia, Chronic osteomyelitis, pelvic region and thigh, Sepsis, Paraplegia, Unspecified essential hypertension, Colostomy status, Other artificial opening of urinary tract status, Anemia of other chronic disease, Other iatrogenic hypotension, Candidiasis of other urogenital sites, Late effects of injury purposely inflicted by other person |
| Procedures: Local excision of lesion or tissue of bone, other bones, Local excision of lesion or tissue of bone, femur, Nonexcisional debridement of wound, infection or burn, Venous catheterization, not elsewhere classified, Transfusion of packed cells |
| Using Drugs: anesthetics, general, i.v. solution additives, other antibacterials, anterior pituitary lobe hormones and analogues, antithrombotic agents, other analgesics and antipyretics, drugs for constipation, drugs for peptic ulcer and gastro-oesophageal reflux disease (gord), antiseptics and disinfectants, antiinfectives and antiseptics, excl. combinations with corticosteroids, other nutrients, opioids, other beta-lactam antibacterials, other mineral supplements, other diagnostic agents, antipsychotics, i.v. solutions, antivaricose therapy, beta-lactam antibacterials, penicillins |
| Predict the number of weeks of stay in hospital. |
| Answer 1 if no more than one week, |
| Answer 2 if more than one week but not more than two weeks, |
| Answer 3 if more than two weeks. |
| Answer with only the number. Answer: |
| *Output for Llama3-8B:* |
| 3 |
| *Output for Gemma2-9B:* |
| 3 |
| *Output for Meditron-7B:* |
| 3 |
| *Output for Internist-7B:* |
| 3 |

### D.1.5 IN-CONTEXT LEARNING

Table 16: Example of LLMs with In Context Learning for Length-of-Stay Prediction.

*Input:*
 Patient information:
Age: 74
Gender: male
Conditions: Subendocardial infarction, initial episode of care, Intestinal infection due to Clostridium difficile, Congestive heart failure, unspecified, Coronary atherosclerosis of native coronary artery, Unspecified essential hypertension, Pure hypercholesterolemia, Abdominal aneurysm without mention of rupture, Percutaneous transluminal coronary angioplasty status, Personal history of malignant neoplasm of large intestine
Procedures: Unknown Procedure, Combined right and left heart cardiac catheterization, Coronary arteriography using two catheters
Using Drugs: i.v. solution additives, cardiac stimulants excl. cardiac glycosides, other nutrients, antithrombotic agents, potassium, expectorants, excl. combinations with cough suppressants, drugs for peptic ulcer and gastro-oesophageal reflux disease (gord), hypnotics and sedatives, other analgesics and antipyretics, opioids, lipid modifying agents, plain, belladonna and derivatives, plain, other mineral supplements, other diagnostic agents, agents against amoebiasis and other protozoal diseases, anxiolytics, quinolone antibacterials, ace inhibitors, plain, beta blocking agents, other antibacterials
Predict the number of weeks of stay in hospital.
Answer 1 if no more than one ,
Answer 2 if more than one week but not more than two weeks,
Answer 3 if more than two weeks.
Answer with only the number. Answer: 1

Patient information:
Age: 36
Gender: male
Conditions: Thoracic aneurysm without mention of rupture, Aortic valve disorders, Congenital insufficiency of aortic valve, Cardiac complications, not elsewhere classified, Other specified cardiac dysrhythmias, Unspecified essential hypertension
Procedures: Open and other replacement of aortic valve, Resection of vessel with replacement, thoracic vessels, Other operations on vessels of heart, Extracorporeal circulation auxiliary to open heart surgery
Using Drugs: drugs for functional gastrointestinal disorders, antiarrhythmics, class i and iii, drugs for constipation, drugs for peptic ulcer and gastro-oesophageal reflux disease (gord), other analgesics and antipyretics, antiinflammatory and antirheumatic products, non-steroids, opioids, other nutrients, anesthetics, general, antiinfectives and antiseptics, excl. combinations with corticosteroids, calcium, arteriolar smooth muscle, agents acting on, cardiac stimulants excl. cardiac glycosides, i.v. solution additives, other beta-lactam antibacterials, insulins and analogues, propulsives, hypnotics and sedatives, other mineral supplements, other diagnostic agents, beta blocking agents, high-ceiling diuretics, potassium, antithrombotic agents, antacids, urologicals, throat preparations, ace inhibitors, plain, angiotensin ii receptor blockers (arbs), plain
Predict the number of weeks of stay in hospital.
Answer 1 if no more than one week,
Answer 2 if more than one week but not more than two weeks,
Answer 3 if more than two weeks.
Answer with only the number. Answer: 2

Table 17: Example of LLMs with In Context Learning for Length-of-Stay Prediction (Cont.).

*Input (Cont.):*
Patient information:
Age: 73
Gender: male
Conditions: Closed fracture of shaft of femur, Acute posthemorrhagic anemia, Subendocardial infarction, initial episode of care, Other postoperative infection, Unspecified septicemia, Severe sepsis, Septic shock, Unknown Diagnosis, Cardiogenic shock, Cardiac complications, not elsewhere classified, Paroxysmal ventricular tachycardia, Ventricular fibrillation, Congestive heart failure, unspecified, Atrial fibrillation, Unknown Diagnosis, Pneumonia, organism unspecified, Unspecified fall, Coronary atherosclerosis of native coronary artery, Cardiac pacemaker in situ
Procedures: Open reduction of fracture with internal fixation, femur, Closed reduction of fracture without internal fixation, femur, Left heart cardiac catheterization, Coronary arteriography using two catheters, Continuous invasive mechanical ventilation for 96 consecutive hours or more, Venous catheterization, not elsewhere classified, Enteral infusion of concentrated nutritional substances
Using Drugs: i.v. solution additives, opioids, other analgesics and antipyretics, beta blocking agents, selective calcium channel blockers with direct cardiac effects, antithrombotic agents, vasodilators used in cardiac diseases, agents for treatment of hemorrhoids and anal fissures for topical use, angiotensin ii receptor blockers (arbs), plain, antidepressants, drugs for constipation, drugs for peptic ulcer and gastro-oesophageal reflux disease (gord), anti-dementia drugs, lipid modifying agents, plain, antipsychotics, antipruritics, incl. antihistamines, anesthetics, etc., anxiolytics, antiarrhythmics, class i and iii, quinolone antibacterials, antacids, urologicals, expectorants, excl. combinations with cough suppressants, antiinfectives and antiseptics, excl. combinations with corticosteroids, calcium, other beta-lactam antibacterials, i.v. solutions, antivaricose therapy, other mineral supplements, other diagnostic agents, high-ceiling diuretics, anesthetics, general, muscle relaxants, peripherally acting agents, other antibacterials, other nutrients, hypnotics and sedatives, beta-lactam antibacterials, penicillins, cardiac stimulants excl. cardiac glycosides, anterior pituitary lobe hormones and analogues, other drugs for obstructive airway diseases, inhalants, adrenergics, inhalants, potassium, antifungals for topical use, ace inhibitors, plain
Predict the number of weeks of stay in hospital.
Answer 1 if no more than one week,
Answer 2 if more than one week but not more than two weeks,
Answer 3 if more than two weeks.
Answer with only the number. Answer: 3

Patient information:
Age: 73
Gender: female
Conditions: Closed fracture of surgical neck of humerus, Other emphysema, Closed fracture of other facial bones, Closed fracture of malar and maxillary bones, Hematoma complicating a procedure, Iatrogenic pulmonary embolism and infarction, Acute posthemorrhagic anemia, Pneumonia, organism unspecified, Fall from other slipping, tripping, or stumbling, Diabetes mellitus without mention of complication, type II or unspecified type, not stated as uncontrolled, Other specified surgical operations and procedures causing abnormal patient reaction, or later complication, without mention of misadventure at time of operation, Personal history of other diseases of circulatory system, Other and unspecified hyperlipidemia, Unspecified essential hypertension

Table 18: Example of LLMs with In Context Learning for Length-of-Stay Prediction (Cont.).

*Input (Cont.):*
Procedures: Partial shoulder replacement, Transfusion of packed cells
Using Drugs: i.v. solution additives, opioids, selective calcium channel blockers with mainly vascular effects, beta blocking agents, high-ceiling diuretics, antithrombotic agents, lipid modifying agents, plain, insulins and analogues, blood glucose lowering drugs, excl. insulins, other analgesics and antipyretics, antiemetics and antinauseants, i.v. solutions, antivaricose therapy, other beta-lactam antibacterials, antipsychotics, anxiolytics, adrenergics, inhalants, other drugs for obstructive airway diseases, inhalants, other antibacterials, corticosteroids for systemic use, plain, other diagnostic agents, vitamin b12 and folic acid, vitamin b1, plain and in combination with vitamin b6 and b12, other mineral supplements, quinolone antibacterials, antidepressants, potassium, drugs for peptic ulcer and gastro-oesophageal reflux disease (gord), drugs for constipation
Predict the number of weeks of stay in hospital.
Answer 1 if no more than one week,
Answer 2 if more than one week but not more than two weeks,
Answer 3 if more than two weeks.
Answer with only the number. Answer:

*Output for Llama3-8B:*
2

*Output for Gemma2-9B:*
2

*Output for Meditron-7B:*
2

*Output for Internist-7B:*
1

## D.2 MORTALITY PREDICTION

### D.2.1 DIRECTLY PROMPTING

Table 19: Example of LLMs with Directly Prompting for Mortality Prediction.

---

*Input:*
Patient information:
Age: 73
Gender: female
Conditions: Closed fracture of surgical neck of humerus, Other emphysema, Closed fracture of other facial bones, Closed fracture of malar and maxillary bones, Hematoma complicating a procedure, Iatrogenic pulmonary embolism and infarction, Acute posthemorrhagic anemia, Pneumonia, organism unspecified, Fall from other slipping, tripping, or stumbling, Diabetes mellitus without mention of complication, type II or unspecified type, not stated as uncontrolled, Other specified surgical operations and procedures causing abnormal patient reaction, or later complication, without mention of misadventure at time of operation, Personal history of other diseases of circulatory system, Other and unspecified hyperlipidemia, Unspecified essential hypertension
Procedures: Partial shoulder replacement, Transfusion of packed cells
Using Drugs: i.v. solution additives, opioids, selective calcium channel blockers with mainly vascular effects, beta blocking agents, high-ceiling diuretics, antithrombotic agents, lipid modifying agents, plain, insulins and analogues, blood glucose lowering drugs, excl. insulins, other analgesics and antipyretics, antiemetics and antinauseants, i.v. solutions, antivaricose therapy, other beta-lactam antibacterials, antipsychotics, anxiolytics, adrenergics, inhalants, other drugs for obstructive airway diseases, inhalants, other antibacterials, corticosteroids for systemic use, plain, other diagnostic agents, vitamin b12 and folic acid, vitamin b1, plain and in combination with vitamin b6 and b12, other mineral supplements, quinolone antibacterials, antidepressants, potassium, drugs for peptic ulcer and gastro-oesophageal reflux disease (gord), drugs for constipation
Will the patient die because of the above situation?
Answer 1 if yes, 0 if no. Answer with only the number.
Answer:

---

*Output for Llama3-8B:*
1

---

*Output for Gemma2-9B:*
0

---

*Output for Meditron-7B:*
1

---

*Output for Internist-7B:*
0

---

### D.2.2 CHAIN-OF-THOUGHT PROMPTING

Table 20: Example of LLMs with Chain of Thought for Length-of-Stay Prediction.

*Input:*
Patient information:
Age: 61
Gender: male
Conditions: Unspecified septicemia, Acute pancreatitis, Alcohol abuse, unspecified, Acute kidney failure, unspecified, Acute and subacute necrosis of liver, Acidosis, Defibrination syndrome, Shock, unspecified, Unknown Diagnosis, Unspecified protein-calorie malnutrition, Acute respiratory failure, Severe sepsis, Anemia, unspecified, Acute alcoholic hepatitis, Personal history of noncompliance with medical treatment, presenting hazards to health, Diabetes mellitus without mention of complication, type II or unspecified type, not stated as uncontrolled, Long-term (current) use of insulin, Abdominal pain, other specified site
Procedures: Continuous invasive mechanical ventilation for less than 96 consecutive hours, Insertion of endotracheal tube, Arterial catheterization, Venous catheterization, not elsewhere classified, Transfusion of packed cells
Using Drugs: insulins and analogues, other nutrients, i.v. solution additives, other beta-lactam antibacterials, anxiolytics, anterior pituitary lobe hormones and analogues, quinolone antibacterials, other antibacterials, vitamin b1, plain and in combination with vitamin b6 and b12, other diagnostic agents, vitamin b12 and folic acid, vitamin k and other hemostatics, other mineral supplements, beta-lactam antibacterials, penicillins, drugs for peptic ulcer and gastro-oesophageal reflux disease (gord), hypnotics and sedatives, anesthetics, general, posterior pituitary lobe hormones, calcium, cardiac stimulants excl. cardiac glycosides
Will the patient die because of the above situation?
Please provide your concise reasoning steps for the prediction(no more than 3 steps), and finally answer 1 if the patient will die and 0 otherwise."
Please provide your concise reasoning steps for the prediction(no more than 3 steps), and finally answer 1 if the patient will die and 0 otherwise

*Output for Llama3-8B:*
Here is the reasoning steps:
Step 1: The patient has severe sepsis, acute kidney failure, and liver failure, which are all life-threatening conditions.
Step 2: The patient also has a history of non-compliance with medical treatment, which may worsen the patient's condition.
Step 3: Given the severity of the patient's conditions and the potential for non-compliance with treatment, it is likely that the patient will die.
Answer: 1

- - - - - - - - - - - - - - - - - - - - - - - - - - - - - - - - - - - - - - - - - -

*Output for Gemma2-9B:*
**Reasoning:**
1. **Severe Sepsis and Multi-Organ Failure:** The patient presents with severe sepsis, a life-threatening condition, and multiple organ failures (pancreas, liver, kidneys, respiratory system).
2. **Complicating Factors:** Alcohol abuse, diabetes, and malnutrition further weaken the patient's immune system and ability to fight infection.
3. **Aggressive Treatment:** The patient is receiving intensive care interventions (ventilation, transfusions, multiple medications), indicating a critical condition requiring urgent medical attention.
**Answer:** 1
**Disclaimer:** I am an AI chatbot and cannot provide medical advice. This prediction is based on the information provided and should not be considered a definitive prognosis. The patient's actual outcome depends on various factors and individual medical management.

- - - - - - - - - - - - - - - - - - - - - - - - - - - - - - - - - - - - - - - - - -

*Output for Meditron-7B:*

Table 20 – continued from previous page

## 1. The patient is 83 years old.
## 2. The patient has pneumonia.
## 3. The patient has organism unspecified.
## 4. The patient has postinflammatory pulmonary fibrosis.
## 5. The patient has atrial fibrillation.
## 6. The patient has congestive heart failure, unspecified.
## 7. The patient has acute respiratory failure.
## 8. The patient has coronary atherosclerosis of native coronary artery.
## 9. The patient has esophageal reflux.
## 10. The patient is using macrolides, lincosamides and streptogramins.
## 11. The patient is using other beta-lactam antibacterials.
## 12. The patient is using expectorants, excl. combinations with cough suppressants.

*Output for Internist-7B:*

Reasoning:

1. Assess the patient's overall condition: The patient has multiple comorbidities, including septicemia, acute pancreatitis, alcohol abuse, acute kidney failure, acute and subacute necrosis of liver, acidosis, defibrination syndrome, shock, unspecified, unknown diagnosis, unspecified protein-calorie malnutrition, acute respiratory failure, severe sepsis, anemia, unspecified, acute alcoholic hepatitis, personal history of noncompliance with medical treatment, presenting hazards to health, diabetes mellitus without mention of complication, type II or unspecified type, not stated as uncontrolled, long-term (current) use of insulin, and abdominal pain.

2. Consider the patient's medical history and current condition: The patient has a history of noncompliance with medical treatment, presenting hazards to health, and long-term use of insulin. The patient also has multiple comorbidities, including septicemia, acute pancreatitis, alcohol abuse, acute kidney failure, acute and subacute necrosis of liver, acidosis, defibrination syndrome, shock, unspecified, unknown diagnosis, unspecified protein-calorie malnutrition, acute respiratory failure, severe sepsis, anemia, unspecified, acute alcoholic hepatitis, diabetes mellitus without mention of complication, type II or unspecified type, not stated as uncontrolled, and abdominal pain.

3. Evaluate the patient's current treatment and procedures: The patient has undergone continuous invasive mechanical ventilation for less than 96 consecutive hours, insertion of an endotracheal tube, arterial catheterization, venous catheterization, not elsewhere classified, transfusion of packed cells, and has been using various medications, including insulins and analogues, other nutrients, i.v. solution additives, other beta-lactam antibacterials, anxiolytics, anterior pituitary lobe hormones and analogues, quinolone antibacterials, other antibacterials, vitamin b1, plain and in combination with vitamin b.....

### D.2.3 SELF-REFLECTION PROMPTING

Table 21: Example of LLMs with Self Reflection for Mortality Prediction.

*Input:*
Patient information:
Age: 61
Gender: male
Conditions: Unspecified septicemia, Acute pancreatitis, Alcohol abuse, unspecified, Acute kidney failure, unspecified, Acute and subacute necrosis of liver, Acidosis, Defibrination syndrome, Shock, unspecified, Unknown Diagnosis, Unspecified protein-calorie malnutrition, Acute respiratory failure, Severe sepsis, Anemia, unspecified, Acute alcoholic hepatitis, Personal history of noncompliance with medical treatment, presenting hazards to health, Diabetes mellitus without mention of complication, type II or unspecified type, not stated as uncontrolled, Long-term (current) use of insulin, Abdominal pain, other specified site
Procedures: Continuous invasive mechanical ventilation for less than 96 consecutive hours, Insertion of endotracheal tube, Arterial catheterization, Venous catheterization, not elsewhere classified, Transfusion of packed cells
Using Drugs: insulins and analogues, other nutrients, i.v. solution additives, other beta-lactam antibacterials, anxiolytics, anterior pituitary lobe hormones and analogues, quinolone antibacterials, other antibacterials, vitamin b1, plain and in combination with vitamin b6 and b12, other diagnostic agents, vitamin b12 and folic acid, vitamin k and other hemostatics, other mineral supplements, beta-lactam antibacterials, penicillins, drugs for peptic ulcer and gastro-oesophageal reflux disease (gord), hypnotics and sedatives, anesthetics, general, posterior pituitary lobe hormones, calcium, cardiac stimulants excl. cardiac glycosides
Will the patient die because of the above situation?
Answer 1 if yes, 0 if no.
First answer with a number. Then conduct a concise reflection. Finally output your answer again with a number.

*Output for Llama3-8B:*
 Answer: 1
Reflection: The patient has a multitude of severe conditions, including septicemia, acute pancreatitis, and acute kidney failure, which are all life-threatening. Additionally, the patient has a history of non-compliance with medical treatment, which may have contributed to the severity of their condition. The patient is also receiving multiple medications, including antibiotics, insulin, and other treatments, which may be helping to manage their condition. However, the patient's overall prognosis is poor, and it is likely that they will die due to the severity of their condition.
Answer: 1
```python
# Define the patient information as a dictionary
patient_info = {
""age"": 61,
......
```

- - - - - - - - - - - - - - - - - - - - - - - - - - - - - - - - - - - - - - -

*Output for Gemma2-9B:*
 0
It is impossible to determine with certainty whether a patient will die based solely on the provided information. While the patient presents with a multitude of severe and life-threatening conditions, including septicemia, pancreatitis, kidney failure, liver necrosis, and shock, the course of these conditions and the effectiveness of treatment are crucial factors. The patient's age, medical history, and response to interventions all play a role in the prognosis. A definitive answer requires a comprehensive medical evaluation and ongoing monitoring.
0

- - - - - - - - - - - - - - - - - - - - - - - - - - - - - - - - - - - - - - -

*Output for Meditron-7B:*

Table 21 – continued from previous page

## 1
## 2
## 3
## 4
## 5
## 6
## 7
## 8
## 9
## 10
## 11
## 12
## 13
## 14

*Output for Internist-7B:*

Reasoning:The patient in this scenario has multiple medical conditions and has undergone several procedures. The question asks if the patient will die because of these factors.

To determine the answer, we need to consider the patient's medical history, current conditions, and the procedures they have undergone.

The patient has a history of alcohol abuse, acute pancreatitis, acute kidney failure, acute and subacute necrosis of the liver, acidosis, defibrination syndrome, shock, unspecified protein-calorie malnutrition, acute respiratory failure, severe sepsis, anemia, acute alcoholic hepatitis, and a personal history of noncompliance with medical treatment.

The patient has also undergone several procedures, including continuous invasive mechanical ventilation for less than 96 consecutive hours, insertion of an endotracheal tube, arterial catheterization, venous catheterization, and transfusion of packed cells.

Given the patient's multiple medical conditions and the procedures they have undergone, it is likely that the patient's overall health is compromised. The combination of these factors increases the risk of complications and potentially death.

Therefore, the answer is 1, indicating that the patient may die due to the above situation.

Reflecting on the answer, it is important to consider the patient's medical history, current conditions, and the procedures they have undergone. The combination of these factors increases the risk of complications and potentially death. It is crucial to provide appropriate medical care and support to the patient to improve their chances of survival.

Outputting the answer again with a number, the answer is still 1, indicating that the patient may die due to the above situation.

In summary, based on the patient's medical history, current conditions, and the procedures they have undergone, there is a high risk of complications and potentially death. It is important to provide appropriate medical care and support to the patient to improve their chances of survival. The answer is 1.

Note: It is important to consider the patient's overall health and medical history when determining the likelihood of death. This answer is based on the information provided in the scenario and may vary in real-life situations. Always consult with a healthcare professional for accurate medical......

### D.2.4 ROLE-PLAYING PROMPTING

Table 22: Example of LLMs with Role-Playing for Mortality Prediction.

---

*Input:*
Imagine that you are a doctor. Today, you're seeing a patient with the following profile:
Patient information:
Age: 73
Gender: female
Conditions: Closed fracture of surgical neck of humerus, Other emphysema, Closed fracture of other facial bones, Closed fracture of malar and maxillary bones, Hematoma complicating a procedure, Iatrogenic pulmonary embolism and infarction, Acute posthemorrhagic anemia, Pneumonia, organism unspecified, Fall from other slipping, tripping, or stumbling, Diabetes mellitus without mention of complication, type II or unspecified type, not stated as uncontrolled, Other specified surgical operations and procedures causing abnormal patient reaction, or later complication, without mention of misadventure at time of operation, Personal history of other diseases of circulatory system, Other and unspecified hyperlipidemia, Unspecified essential hypertension
Procedures: Partial shoulder replacement, Transfusion of packed cells
Using Drugs: i.v. solution additives, opioids, selective calcium channel blockers with mainly vascular effects, beta blocking agents, high-ceiling diuretics, antithrombotic agents, lipid modifying agents, plain, insulins and analogues, blood glucose lowering drugs, excl. insulins, other analgesics and antipyretics, antiemetics and antinauseants, i.v. solutions, antivaricose therapy, other beta-lactam antibacterials, antipsychotics, anxiolytics, adrenergics, inhalants, other drugs for obstructive airway diseases, inhalants, other antibacterials, corticosteroids for systemic use, plain, other diagnostic agents, vitamin b12 and folic acid, vitamin b1, plain and in combination with vitamin b6 and b12, other mineral supplements, quinolone antibacterials, antidepressants, potassium, drugs for peptic ulcer and gastro-oesophageal reflux disease (gord), drugs for constipation
Will the patient die because of the above situation?
Answer 1 if yes, 0 if no. Answer with only the number.
Answer:

---

*Output for Llama3-8B:*
1

- - - - - - - - - - - - - - - - - - - - - - - - - - - - - - - - - - - - - - - - - -

*Output for Gemma2-9B:*
1

- - - - - - - - - - - - - - - - - - - - - - - - - - - - - - - - - - - - - - - - - -

*Output for Meditron-7B:*
1

- - - - - - - - - - - - - - - - - - - - - - - - - - - - - - - - - - - - - - - - - -

*Output for Internist-7B:*
1

### D.2.5 IN-CONTEXT LEARNING

Table 23: Example of LLMs with In Context Learning for Mortality Prediction.

*Input:*

Patient information:
Age: 43
Gender: female
Conditions: Coronary atherosclerosis of native coronary artery, Intermediate coronary syndrome, Diabetes mellitus without mention of complication, type I [juvenile type], not stated as uncontrolled, Unspecified essential hypertension, Pure hypercholesterolemia, Tobacco use disorder
Procedures: (Aorto)coronary bypass of two coronary arteries, Left heart cardiac catheterization, Extracorporeal circulation auxiliary to open heart surgery, Coronary arteriography using two catheters, Angiocardiography of left heart structures
Using Drugs: other analgesics and antipyretics, antipsychotics, vasodilators used in cardiac diseases, antacids, urologicals, anxiolytics, antidepressants, potassium, iron preparations, lipid modifying agents, plain, beta blocking agents, drugs for peptic ulcer and gastro-oesophageal reflux disease (gord), dopaminergic agents, thyroid preparations, opioids, other nutrients, cardiac stimulants excl. cardiac glycosides, drugs for constipation, i.v. solution additives, calcium, propulsives, anti-inflammatory and antirheumatic products, non-steroids, other antibacterials, high-ceiling diuretics, antithrombotic agents, other beta-lactam antibacterials, other mineral supplements Will the patient die because of the above situation?
Answer 1 if yes, 0 if no. Answer with only the number.
Answer: 0

Patient information:
Age: 86
Gender: male
Conditions: Intracerebral hemorrhage, Pneumonitis due to inhalation of food or vomitus, Unspecified essential hypertension, Aortocoronary bypass status, Coronary atherosclerosis of unspecified type of vessel, native or graft
Procedures: Continuous invasive mechanical ventilation for less than 96 consecutive hours
Using Drugs: beta blocking agents, vitamin b1, plain and in combination with vitamin b6 and b12, i.v. solution additives, antiepileptics, quinolone antibacterials, other antibacterials, drugs for peptic ulcer and gastro-oesophageal reflux disease (gord), other mineral supplements, other diagnostic agents, anxiolytics, anesthetics, general, opioids, antiemetics and antinauseants, hypnotics and sedatives
Will the patient die because of the above situation?
Answer 1 if yes, 0 if no. Answer with only the number.
Answer: 1

Patient information:
Age: 73
Gender: female
Conditions: Closed fracture of surgical neck of humerus, Other emphysema, Closed fracture of other facial bones, Closed fracture of malar and maxillary bones, Hematoma complicating a procedure, Iatrogenic pulmonary embolism and infarction, Acute posthemorrhagic anemia, Pneumonia, organism unspecified, Fall from other slipping, tripping, or stumbling, Diabetes mellitus without mention of complication, type II or unspecified type, not stated as uncontrolled, Other specified surgical operations and procedures causing abnormal patient reaction, or later complication, without mention of misadventure at time of operation, Personal history of other diseases of circulatory system, Other and unspecified hyperlipidemia, Unspecified essential hypertension
Procedures: Partial shoulder replacement, Transfusion of packed cells

*Continued on next page*

Table 23 – continued from previous page

Using Drugs: i.v. solution additives, opioids, selective calcium channel blockers with mainly vascular effects, beta blocking agents, high-ceiling diuretics, antithrombotic agents, lipid modifying agents, plain, insulins and analogues, blood glucose lowering drugs, excl. insulins, other analgesics and antipyretics, antiemetics and antinauseants, i.v. solutions, antivaricose therapy, other beta-lactam antibacterials, antipsychotics, anxiolytics, adrenergics, inhalants, other drugs for obstructive airway diseases, inhalants, other antibacterials, corticosteroids for systemic use, plain, other diagnostic agents, vitamin b12 and folic acid, vitamin b1, plain and in combination with vitamin b6 and b12, other mineral supplements, quinolone antibacterials, antidepressants, potassium, drugs for peptic ulcer and gastro-oesophageal reflux disease (gord), drugs for constipation
Will the patient die because of the above situation?
Answer 1 if yes, 0 if no. Answer with only the number.
Answer:

*Output for Llama3-8B:*
1

*Output for Gemma2-9B:*
1

*Output for Meditron-7B:*
1

*Output for Internist-7B:*
1

## D.3 READMISSION PREDICTION

### D.3.1 DIRECTLY PROMPTING

Table 24: Example of LLMs with Directly Prompting for Readmission Prediction.

---

*Input:*
Patient information:
Age: 76
Gender: female
Conditions: Other specified intestinal obstruction, Malignant neoplasm of anus, unspecified site, Stenosis of rectum and anus, Unknown Diagnosis, Coronary atherosclerosis of native coronary artery, Esophageal reflux, Unspecified essential hypertension, Personal history of malignant neoplasm of rectum, rectosigmoid junction, and anus
Procedures: Temporary colostomy, Combined right and left heart cardiac catheterization, Coronary arteriography using two catheters, Venous catheterization, not elsewhere classified
Using Drugs: opioids, antithrombotic agents, antiinfectives and antiseptics, excl. combinations with corticosteroids, drugs for peptic ulcer and gastro-oesophageal reflux disease (gord), other antibacterials, other beta-lactam antibacterials, beta blocking agents, other mineral supplements, i.v. solution additives, other diagnostic agents, other nutrients, other analgesics and antipyretics, belladonna and derivatives, plain, i.v. solutions, antivaricose therapy, antiemetics and antinauseants, drugs for constipation
Will the patient be readmitted to the hospital within two weeks?
Answer 1 for yes, 0 for no. Answer with only the number.
Answer:

---

*Output for Llama3-8B:*
1

- - - - - - - - - - - - - - - - - - - - - - - - - - - - - - - - - - - - - - - - - - - - - -

*Output for Gemma2-9B:*
1

- - - - - - - - - - - - - - - - - - - - - - - - - - - - - - - - - - - - - - - - - - - - - -

*Output for Meditron-7B:*
1

- - - - - - - - - - - - - - - - - - - - - - - - - - - - - - - - - - - - - - - - - - - - - -

*Output for Internist-7B:*
1

---

### D.3.2 CHAIN-OF-THOUGHT PROMPTING

Table 25: Example of LLMs with Chain of Thought for Readmission Prediction.

| |
|---|
| *Input:* |
| Patient information: |
| Age: 68 |
| Gender: male |
| Conditions: Malignant neoplasm of other specified sites of bladder, Hemorrhage complicating a procedure |
| Procedures: Other transurethral excision or destruction of lesion or tissue of bladder, Control of (postoperative) hemorrhage of bladder |
| Using Drugs: antiemetics and antinauseants, opioids, belladonna and derivatives, plain, antipsychotics, antihistamines for systemic use, quinolone antibacterials, i.v. solutions, antivaricose therapy, hypnotics and sedatives, other analgesics and antipyretics, urologicals, other mineral supplements, i.v. solution additives, other diagnostic agents |
| Will the patient be readmitted to the hospital within two weeks? |
| Please provide your concise reasoning steps for the prediction(no more than 3 steps), and finally answer 1 if the patient will die and 0 otherwise |
| *Output for Llama3-8B:* |
| Here is the solution: |
| Step 1: The patient has undergone a transurethral excision or destruction of lesion or tissue of bladder, which is a surgical procedure. This increases the likelihood of complications and readmission. |
| Step 2: The patient has a history of hemorrhage complicating a procedure, which is a significant complication that may require further treatment and hospitalization. |
| Step 3: The patient is taking a variety of medications, including opioids, antihistamines, and antipsychotics, which may increase the risk of adverse reactions and complications. |
| Answer: 1 (The patient will be readmitted to the hospital within two weeks) |
| Note: This prediction is based on the patient's medical history and procedures, and may not be accurate in all cases. A more detailed review of the patient's medical records and current condition would be necessary to make a more accurate prediction. |
| *Output for Gemma2-9B:* |
| **Reasoning:** |
| 1. **Procedure:** The patient underwent a procedure for bladder cancer (""Other transurethral excision or destruction of lesion or tissue of bladder"") and experienced postoperative hemorrhage. |
| 2. **Complications:** Postoperative hemorrhage is a potential complication of bladder cancer procedures and can require further intervention. |
| 3. **Drug Usage:** The patient is on a variety of medications, including opioids and antiemetics, which are often used for pain management and nausea after surgery. This suggests ongoing recovery and potential need for further care. |
| **Answer:** 1 |
| *Output for Meditron-7B:* |
| *Continued on next page* |

Table 25 – continued from previous page

## 1. The patient is male.
## 2. The patient is 68 years old.
## 3. The patient has a malignant neoplasm of other specified sites of bladder.
## 4. The patient has a hemorrhage complicating a procedure.
## 5. The patient has other transurethral excision or destruction of lesion or tissue of bladder.
## 6. The patient has control of (postoperative) hemorrhage of bladder.
## 7. The patient has antiemetics and antinauseants.
## 8. The patient has opioids.
## 9. The patient has belladonna and derivatives.
## 10. The patient has plain. ......

- - - - - - - - - - - - - - - - - - - - - - - - - - - - - - - - - - - - -

*Output for Internist-7B:*

Reasoning:

1. Assess the patient's condition: The patient is a 68-year-old male with a malignant neoplasm of other specified sites of the bladder and hemorrhage complicating a procedure. This indicates that the patient has undergone a procedure related to the bladder and is experiencing bleeding as a complication.

2. Consider the procedures and medications: The patient has undergone a transurethral excision or destruction of lesion or tissue of the bladder and control of postoperative hemorrhage of the bladder. The patient is also using various medications, including antiemetics and antinauseants, opioids, belladonna and derivatives, antipsychotics, antihistamines, quinolone antibacterials, i.v. solutions, antivaricose therapy, hypnotics and sedatives, other analgesics and antipyretics, urologicals, other mineral supplements, i.v. solution additives, and other diagnostic agents.

3. Evaluate the risk factors: The patient's age, gender, and medical history are important factors to consider. The patient is 68 years old, which may increase the risk of complications and readmission. The patient's gender and medical history are also relevant, as they may contribute to the overall risk assessment.

4. Analyze the likelihood of readmission: Based on the patient's condition, procedures, medications, and risk factors, it is important to assess the likelihood of readmission within two weeks. This may involve considering the severity of the patient's condition, the effectiveness of the procedures and medications, and the potential for complications or recurrence of the bleeding.

5. Make the prediction: Based on the analysis, if the likelihood of readmission within two weeks is high, the answer would be 1. If the likelihood is low, the answer would be 0.

In this case, the patient's condition, procedures, medications, and risk factors suggest a high likelihood of readmission within two weeks. Therefore, the answer is 1.

Note: It is important to consider the specific patient's condition, procedures, medications, and risk factors when making a prediction. This explanation is a general guide and may not apply to every individual case.

### D.3.3 SELF-REFLECTION PROMPTING

Table 26: Example of LLMs with Self Reflection for Readmission Prediction.

---

*Input:*
Patient information:
Age: 68
Gender: male
Conditions: Malignant neoplasm of other specified sites of bladder, Hemorrhage complicating a procedure
Procedures: Other transurethral excision or destruction of lesion or tissue of bladder, Control of (postoperative) hemorrhage of bladder
Using Drugs: antiemetics and antinauseants, opioids, belladonna and derivatives, plain, antipsychotics, antihistamines for systemic use, quinolone antibacterials, i.v. solutions, antivaricose therapy, hypnotics and sedatives, other analgesics and antipyretics, urologicals, other mineral supplements, i.v. solution additives, other diagnostic agents
Will the patient be readmitted to the hospital within two weeks?
Answer 1 for yes, 0 for no. Answer with only the number.
First answer with a number. Then conduct a concise reflection. Finally output your answer again with a number.

---

*Output for Llama3-8B:*
 Please enter your answer.
```

1
```

Reflection: Based on the patient's conditions and procedures, it is likely that the patient will be readmitted to the hospital within two weeks. The patient has a malignant neoplasm of the bladder, which may require further treatment or monitoring. Additionally, the patient has undergone a transurethral excision or destruction of lesion or tissue of the bladder, which may lead to complications or side effects that require hospitalization. The patient is also taking a number of medications, which may increase the risk of readmission.
Answer:
```

1
``` ``` 1
``` ``` 1
``` ``` 1
``` ``` 1
``` ``` 1
``` ``` 1
``` ``` 1
``` ``` 1
``` ``` 1
``` ``` 1"

- - - - - - - - - - - - - - - - - - - - - - - - - - - - - - - - - - - - - - - - -

*Output for Gemma2-9B:*
 **Reflection:**
The patient underwent a procedure for a malignant bladder tumor and experienced post-operative hemorrhage. This suggests a potentially complex recovery and a higher risk of complications. The use of various medications, including opioids and antiemetics, points towards ongoing pain management and potential nausea/vomiting.
While it's impossible to say for certain without more information, the patient's condition and the nature of the procedure increase the likelihood of readmission within two weeks.
**Answer:** 1

Table 26 – continued from previous page

*Output for Meditron-7B:*
## 1
## 2
## 3
## 4
## 5
## 6
## 7
## 8
## 9
## 10
## 11
## 12
## 13
## 14
## 15

*Output for Internist-7B:*
Reasoning:The patient in this scenario is a 68-year-old male with a malignant neoplasm of other specified sites of the bladder and hemorrhage complicating a procedure. The patient has undergone a transurethral excision or destruction of lesion or tissue of the bladder and control of postoperative hemorrhage of the bladder. The patient is also using various medications, including antiemetics and antinauseants, opioids, belladonna and derivatives, antipsychotics, antihistamines for systemic use, quinolone antibacterials, i.v. solutions, antivaricose therapy, hypnotics and sedatives, other analgesics and antipyretics, urologicals, other mineral supplements, i.v. solution additives, and other diagnostic agents.

Based on this information, we need to determine if the patient will be readmitted to the hospital within two weeks. To do this, we need to consider the patient's condition, the procedures performed, and the medications being used.

First, the patient has a malignant neoplasm of other specified sites of the bladder, which indicates a serious underlying condition. This suggests that the patient may require ongoing medical management and follow-up.

Second, the patient has experienced hemorrhage complicating a procedure. This indicates that the patient may have experienced bleeding during the transurethral excision or destruction of lesion or tissue of the bladder. This complication may require further intervention or monitoring.

Third, the patient is using various medications, including antiemetics and antinauseants, opioids, belladonna and derivatives, antipsychotics, antihistamines for systemic use, quinolone antibacterials, i.v. solutions, antivaricose therapy, hypnotics and sedatives, other analgesics and antipyretics, urologicals, other mineral supplements, i.v. solution additives, and other diagnostic agents. These medications may have side effects or interactions that could potentially lead to complications or the need for further medical attention.

Considering all of these factors, it is likely that the patient may require readmission to the hospital within two weeks......

### D.3.4  ROLE-PLAYING PROMPTING

Table 27: Example of LLMs with Role-Playing for Readmission Prediction.

*Input:*
Imagine that you are a doctor. Today, you're seeing a patient with the following profile:
Patient information:
Age: 68
Gender: male
Conditions: Malignant neoplasm of other specified sites of bladder, Hemorrhage complicating a procedure
Procedures: Other transurethral excision or destruction of lesion or tissue of bladder, Control of (postoperative) hemorrhage of bladder
Using Drugs: antiemetics and antinauseants, opioids, belladonna and derivatives, plain, antipsychotics, antihistamines for systemic use, quinolone antibacterials, i.v. solutions, antivaricose therapy, hypnotics and sedatives, other analgesics and antipyretics, urologicals, other mineral supplements, i.v. solution additives, other diagnostic agents
Will the patient be readmitted to the hospital within two weeks? Answer 1 for yes, 0 for no. Answer with only the number.
Answer:

*Output for Llama3-8B:*
1

*Output for Gemma2-9B:*
1

*Output for Meditron-7B:*
1

*Output for Internist-7B:*
1

### D.3.5 IN-CONTEXT LEARNING

Table 28: Example of LLMs with In Context Learning for Readmission Prediction.

---

*Input:*

Patient information:
Age: 31
Gender: female
Conditions: Acute respiratory failure, Myasthenia gravis with (acute) exacerbation, Other specified cardiac dysrhythmias, Diarrhea, Unspecified essential hypertension, Iron deficiency anemia, unspecified
Procedures: Continuous invasive mechanical ventilation for 96 consecutive hours or more, Insertion of endotracheal tube, Injection or infusion of immunoglobulin, Non-invasive mechanical ventilation, Enteral infusion of concentrated nutritional substances, Transfusion of packed cells
Using Drugs: immunosuppressants, corticosteroids for systemic use, plain, drugs for peptic ulcer and gastro-oesophageal reflux disease (gord), other nutrients, i.v. solutions, antivaricose therapy, anticholinergic agents, beta blocking agents, anesthetics, general, parasympathomimetics, other antidiarrheals, calcium, other analgesics and antipyretics, antithrombotic agents, antacids, potassium, i.v. solution additives, sulfonamides and trimethoprim, ace inhibitors, plain, antipropulsives, antidepressants, belladonna and derivatives, plain, anxiolytics, hypnotics and sedatives, other cardiac preparations, antiseptics and disinfectants, antiepileptics
Will the patient be readmitted to the hospital within two weeks?
Answer 1 for yes, 0 for no. Answer with only the number.
Answer: 0

Patient information:
Age: 62
Gender: male
Conditions: Acute myocardial infarction of unspecified site, initial episode of care, Congestive heart failure, unspecified, Unknown Diagnosis, Acute kidney failure with lesion of tubular necrosis, Other and unspecified complications of medical care, not elsewhere classified, Pneumonia, organism unspecified, Late effects of cerebrovascular disease, hemiplegia affecting unspecified side, Unspecified pleural effusion, Unknown Diagnosis, Anticoagulants causing adverse effects in therapeutic use, Diabetes mellitus without mention of complication, type II or unspecified type, not stated as uncontrolled, Anemia, unspecified, Atrial fibrillation, Attention to tracheostomy, Unspecified essential hypertension, Pure hypercholesterolemia, Coronary atherosclerosis of unspecified type of vessel, native or graft, Aortocoronary bypass status
Procedures: Thoracentesis, Enteral infusion of concentrated nutritional substances, Infusion of drotrecogin alfa (activated), Continuous invasive mechanical ventilation for less than 96 consecutive hours
Using Drugs: antithrombotic agents, beta blocking agents, antiarrhythmics, class i and iii, other nutrients, lipid modifying agents, plain, drugs for peptic ulcer and gastro-oesophageal reflux disease (gord), other analgesics and antipyretics, other beta-lactam antibacterials, other antibacterials, i.v. solution additives, hypnotics and sedatives, anesthetics, general, agents against amoebiasis and other protozoal diseases, quinolone antibacterials, ace inhibitors, plain, high-ceiling diuretics, other antidiarrheals, antacids, insulins and analogues, decongestants and other nasal preparations for topical use, intestinal antiinfectives, calcium, antifungals for topical use, anxiolytics, beta-lactam antibacterials, penicillins, adrenergics, inhalants, antidepressants, potassium
Will the patient be readmitted to the hospital within two weeks?
Answer 1 for yes, 0 for no. Answer with only the number.
Answer: 1

Patient information:

Table 28 – continued from previous page

Age: 68
Gender: male
Conditions: Malignant neoplasm of other specified sites of bladder, Hemorrhage complicating a procedure
Procedures: Other transurethral excision or destruction of lesion or tissue of bladder, Control of (postoperative) hemorrhage of bladder
Using Drugs: antiemetics and antinauseants, opioids, belladonna and derivatives, plain, antipsychotics, antihistamines for systemic use, quinolone antibacterials, i.v. solutions, antivaricose therapy, hypnotics and sedatives, other analgesics and antipyretics, urologicals, other mineral supplements, i.v. solution additives, other diagnostic agents
Will the patient be readmitted to the hospital within two weeks? Answer 1 for yes, 0 for no. Answer with only the number.
Answer:

*Output for Llama3-8B:*
0

*Output for Gemma2-9B:*
0

*Output for Meditron-7B:*
0

*Output for Internist-7B:*
0

