# OpenReview forum: "ClinicalBench: Can LLMs Beat Traditional ML Models in Clinical Prediction?"
_ICLR.cc/2025/Conference — Submitted to ICLR 2025_

### Official Review · Reviewer_YY2z · 2024-10-31

**Soundness:** 3
**Presentation:** 4
**Contribution:** 4
**Rating:** 8
**Confidence:** 3

**Summary:**

This paper compares the performances of multiple LLM and traditional ML models on three clinical tasks (length of stay, mortality, readmission) on MIMIC III and IV. This paper contributes key conclusions on the underperformance of LLMs compared to traditional ML models. Furthermore, the authors demonstrate that specialised models do not always perform better and that advanced prompting strategies also do not help. Only fine-tuning strategies significantly improve the LLM performance on these tasks.

**Strengths:**

The paper explores the use of LLM in medical tasks clearly and thoroughly.

**Weaknesses:**

- The paper could discuss more in depth the different works that have aimed to use LLM for prediction and how the proposed paper differs.
- One limitation could be the reliance on MIMIC used to train some of these models. However, the current underperformance of LLMs only further emphasises the limitations of these strategies, which may benefit from data leakage.

**Questions:**

None

---

> ### Author Response · Authors · 2024-12-04
> **We sincerely appreciate the comment  and suggestions. Thanks!**
>
> Dear Reviewer YY2z,
>
> We are genuinely  grateful for your acknowledgement of our contributions!
>
> > The paper could discuss more in depth the different works that have aimed to use LLM for prediction and how the proposed paper differs.
>
> Thanks for the comment! We have provided a detailed discussion on the differences between our proposed ClinicalBench and previous benchmarks in Section 7. Our work made the first attempt to comprehensively and systemically compare the clinical prediction capacities of LLMs and traditional ML models in a head-to-head way.
>
> > One limitation could be the reliance on MIMIC used to train some of these models. However, the current underperformance of LLMs only further emphasises the limitations of these strategies, which may benefit from data leakage.
>
> Thanks for the comment! We will further study the reasons for the under-performance of LLMs and the impact of potential data leakage in the future.
>
> We sincerely appreciate the comment and suggestions. Thanks!

---

### Official Review · Reviewer_8CVU · 2024-11-04

**Soundness:** 3
**Presentation:** 3
**Contribution:** 3
**Rating:** 8
**Confidence:** 4

**Summary:**

This paper introduces ClinicalBench, a benchmark designed to evaluate the performance of 14 general-purpose LLMs, 8 medical LLMs, and 11 traditional ML models (e.g., XGBoost, Logistic Regression, RNN etc.) on three clinical classification tasks: Length-of-Stay Prediction (three-class classification), Mortality Prediction (binary classification), and Readmission Prediction (binary classification). On both the MIMIC-III and MIMIC-IV datasets, this paper shows that most LLMs currently still fall short in making accurate predictions for these real-world clinical tasks, underperforming by an obvious margin compared to traditional ML models.

**Strengths:**

1. This paper is clearly written and easy to follow, and the figures are informative.
2. This paper offers a head-to-head comparison between LLMs and traditional ML methods on three real-world clinical tasks, helping to clarify the current capabilities of LLMs in this domain. The three ways of enabling LLMs to make predictions utilized in this paper - direct prompting, engineered prompting (e.g. Chain-of-Thought, Self-Reflection, Role-Playing, In-Context Learning), and supervised fine-tuning using task-specific data - reflects the primary approaches for applying LLMs nowadays, making its insights broadly applicable and valuable for future research.
3. The evaluation is quite thorough, and the findings offer interesting insights for this domain.

**Weaknesses:**

1. While named as a benchmark, this paper did not introduce or curate new datasets, or introduce novel ways of processing or derive new clinical tasks from the MIMIC-III and MIMIC-IV datasets. The main contribution is in developing the framework of comparing LLMs with traditional ML methods on standard clinical tasks in MIMIC-III and MIMIC-IV. Thus I would argue that the contribution may be slightly less significant compared to benchmarks which both introduce new clinical tasks closer to real-world clinical reasoning / decision-making and also develops the LLM evaluation framework.
2. For Section 5 fine-tuning, the details about how the LLMs are fine-tuned are unclear in the main paper - is it by adding a classification head to the output of the transformer, or by next-token prediction on target outputs (from Appendix C it seems to be the latter)? I think adding this detail to Section 5 would help the readers understand how SFT was conducted and better understand the performance discrepancy compared to traditional ML methods.
3. For In-context Learning, it seems that 3 examples are provided to the LLMs per the prompt example in Appendix D. Are these examples chosen randomly or selected to be representative of the patient population? I think it might be helpful to also analyze how the number of in-context examples and how the diversity of the in-context examples (e.g. different disease type, different age groups etc.) impact the LLM performance, as sometimes LLMs can be quite sensitive to that.
4. The impact of parameter scaling is studied for direct prompting but not for engineered prompting, i.e., only 7B, 8B and 9B models are studied for engineered prompting but not 34B or 70B. I would assume that large models have better instruction following capabilities and thus may be better at reasoning or role play than smaller models, and would like to know your insights on this and why larger models are not used for engineered prompting.

**Questions:**

Please refer to weaknesses.

---

> ### Author Response · Authors · 2024-12-04
> **We sincerely appreciate the comment and provide clarifications as follows (1)**
>
> >While named as a benchmark, this paper did not introduce or curate new datasets, or introduce novel ways of processing or derive new clinical tasks from the MIMIC-III and MIMIC-IV datasets. The main contribution is in developing the framework of comparing LLMs with traditional ML methods on standard clinical tasks in MIMIC-III and MIMIC-IV. Thus I would argue that the contribution may be slightly less significant compared to benchmarks which both introduce new clinical tasks closer to real-world clinical reasoning / decision-making and also develops the LLM evaluation framework.
>
> **We would like to clarify the main contribution of our benchmark.** Our work made the first attempt to comprehensively and systemically **compare the clinical prediction capacities of LLMs and traditional ML models in a head-to-head way.** We discover that **both general-purpose and medical LLMs, even with different model sizes, prompting or fine-tuning strategies, still cannot beat traditional ML models in clinical prediction yet.**
>
> **We also would like to emphasize the significance of our insights.** Previous results have shown that general-purpose and medical LLMs could have almost clinician-level performance in medical licensing exams and clinical case challenges. **But our findings demonstrate the surprising but critical deficiency of both general-purpose and medical LLMs in real-world clinical reasoning and decision-making.** We call for caution when adopting LLMs in practical clinical applications. ClinicalBench could be leveraged to bridge the gap between the development of LLMs for healthcare and real-world clinical practice.
>
> > For Section 5 fine-tuning, the details about how the LLMs are fine-tuned are unclear in the main paper - is it by adding a classification head to the output of the transformer, or by next-token prediction on target outputs (from Appendix C it seems to be the latter)? I think adding this detail to Section 5 would help the readers understand how SFT was conducted and better understand the performance discrepancy compared to traditional ML methods.
>
> Thanks for the suggestion. We do not add any additional parameters in the experiments. The details of the instruction tuning dataset have been described in Appendix C. We will make it more clear in the revision.
>
> > For In-context Learning, it seems that 3 examples are provided to the LLMs per the prompt example in Appendix D. Are these examples chosen randomly or selected to be representative of the patient population? I think it might be helpful to also analyze how the number of in-context examples and how the diversity of the in-context examples (e.g. different disease type, different age groups etc.) impact the LLM performance, as sometimes LLMs can be quite sensitive to that.
>
> Thanks for the suggestion. The 3 examples are randomly selected. Due to the limit of the length of the context, only three in-context examples can be incorporated in the context. It is our future work to conduct experiments on LLMs with longer context.

---

> ### Author Response · Authors · 2024-12-04
> **We sincerely appreciate the comment and provide clarifications as follows (2)**
>
> > The impact of parameter scaling is studied for direct prompting but not for engineered prompting, i.e., only 7B, 8B and 9B models are studied for engineered prompting but not 34B or 70B. I would assume that large models have better instruction following capabilities and thus may be better at reasoning or role play than smaller models, and would like to know your insights on this and why larger models are not used for engineered prompting.
>
> Thanks for the suggestion. We have provided the results of Llama3-70B (general-purpose LLM) and Med42-70B (medical LLM) with different prompting strategies as follows. The complete results are in https://anonymous.4open.science/r/llm4clinical_prediction-D09D/src/results
> - For example, the results of Llama3-70B with CoT on MIMIC-III for length-of-stay prediction task are in https://anonymous.4open.science/r/llm4clinical_prediction-D09D/src/results/length_pred/mimic3/length_pred_result_data_Meta-Llama-3-70B-Instruct_6_COT.csv
> - The results of  Med42-70B with ICL on MIMIC-III for length-of-stay prediction task are in
> https://anonymous.4open.science/r/llm4clinical_prediction-D09D/src/results/length_pred/mimic3/length_pred_result_data_Llama3-Med42-70B_6_ICL.csv
>
> We can clearly see that the results further strengthen our **Finding 2: The effectiveness of typical prompting engineering techniques is generally limited and they still cannot make general-purpose and medical LLMs beat traditional ML models yet.**
>
> |     Llama3-70B     | Length (mimic3) | Length (mimic4) | Mortality (mimic3) | Mortality (mimic4) | Readmission (mimic3) | Readmission (mimic4) |
> |----------|------------------|------------------|---------------------|---------------------|-----------------------|-----------------------|
> | **ORI**  | 18.88           | 21.76           | 29.82              | 17.80              | 13.72                | 22.03                |
> | **ICL**  | 41.04           | 47.52           | 25.22              | 12.41              | 12.47                | 25.43                |
> | **RP**   | 11.68           | 11.37           | 23.34              | 11.68              | 13.83                | 24.95                |
> | **COT**  | 10.00           | 9.02            | 0.00               | 0.00               | 0.00                 | 0.00                 |
> | **SR**   | 25.70           | 26.28           | 21.09              | 7.22               | 13.81                | 25.13                |
>
> |     Med42-70B    | Length (mimic3) | Length (mimic4) | Mortality (mimic3) | Mortality (mimic4) | Readmission (mimic3) | Readmission (mimic4) |
> |----------|------------------|------------------|---------------------|---------------------|-----------------------|-----------------------|
> | **ORI**  | 27.23           | 34.52           | 46.15              | 34.48              | 9.64                 | 9.90                 |
> | **ICL**  | 55.40           | 50.79           | 27.40              | 11.26              | 8.56                 | 19.35                |
> | **RP**   | 46.46           | 56.06           | 14.08              | 34.48              | 0.00                 | 2.70                 |
> | **COT**  | 9.39            | 6.92            | 4.24               | 0.43               | 11.34                | 18.66                |
> | **SR**   | 17.72           | 13.51           | 0.00               | 0.00               | 0.00                 | 1.69                 |

---

> ### Author Response · Authors · 2024-12-04
> **Dear Reviewer, if you have already found our responses satisfactory, we humbly remind you of a fitting update of the final rating. Thanks for your time and effort again!**
>
> Dear Reviewer 8CVU,
>
> We are genuinely grateful for your constructive feedback and the acknowledgement of our contributions. If you have already found our responses satisfactory, we humbly remind you of a fitting update of the final rating. Thanks for your time and effort again!
>
> The authors

---

### Official Review · Reviewer_H2Eh · 2024-11-04

**Soundness:** 2
**Presentation:** 3
**Contribution:** 1
**Rating:** 3
**Confidence:** 4

**Summary:**

The paper explores the capabilities of large language models (both general-purpose and medical-oriented models) in clinical prediction tasks (including length-of-stay, mortality, and readmission), comparing them with conventional machine learning-based models. The paper reveals that LLMs still cannot outperform traditional ML models in various settings (parameter size, different LLMs, prompting/finetuning strategies, etc.). The dataset employed in this research consists of code-based data extracted from MIMIC-III and MIMIC-IV. To better understand the code semantics, the authors convert codes to text.

**Strengths:**

The paper is well-presented and organized with clear motivation and writing flow. The paper addresses a key problem in the era of LLMs, where plenty of researchers are exploring the boundary capability of LLMs.

**Weaknesses:**

My main concern is the paper's novelty and contribution. There are already many works that explore LLMs' capabilities on diverse tasks, including those in the medical field. The findings are straightforward and predictable: machine learning models trained on specific medical tasks naturally surpass LLMs' zero-shot performance, and in real-world clinical practice, the simplest models (e.g., logistic regression, LASSO) are still widely used for their simplicity and interpretability. With the development of more advanced LLMs, the findings may change, as they only reflect current guidelines and cannot predict the future practical use of LLMs. The employed LLMs are mainly small in size; larger models of 70B parameters and beyond should be explored. These larger language models have much better prompt-following capabilities, so CoT and ICL approaches may work, rather than simply concluding that "The effectiveness of typical prompting engineering techniques is generally limited." Additionally, no variance is reported in the benchmarking table, although LLMs' outputs can be sensitive to input prompts, temperature settings, and the sequence of input instructions.

**Questions:**

See weaknesses section above.

---

> ### Author Response · Authors · 2024-12-04
> **We sincerely appreciate the comment and provide clarifications as follows (1)**
>
> Dear Reviewer H2Eh,
>
> We sincerely appreciate the comment and provide clarifications as follows:
>
> > My main concern is the paper's novelty and contribution. There are already many works that explore LLMs' capabilities on diverse tasks, including those in the medical field.
>
> Thanks for the comment! **We have provided a detailed discussion on the differences between our proposed ClinicalBench and previous benchmarks in Section 7**. As discussed in Line 483-498, The majority of existing clinical benchmarks for LLMs can generally be categorized from two perspectives. The first one aims to assess LLMs’ capacities in clinical text processing tasks (Harris et al., 2024; Jahan et al., 2024; Feng et al., 2024; Chen et al., 2023a; Wang et al., 2023a; Luo et al., 2024). The second one intends to evaluate LLMs performance in clinical question-answering tasks such as MedQA (Jin et al., 2021) and MedMCQA (Pal et al., 2022). `However, benchmarks on LLMs’
> clinical reasoning capacities with real-world complex scenarios are relatively lacking. ClinicalBench
> filled the gap through a comprehensive investigation involving three common clinical prediction
> tasks, two databases, 14 general-purpose LLMs, 8 medical LLMs and has provided valuable insights.`
>
>
> > The findings are straightforward and predictable: machine learning models trained on specific medical tasks naturally surpass LLMs' zero-shot performance, and in real-world clinical practice, the simplest models (e.g., logistic regression, LASSO) are still widely used for their simplicity and interpretability.
>
> We would like to clarify that we **not only evaluate the performance of zero-shot prompting, but also evaluate the performance of LLMs with different model sizes and temperatures, prompting or fine-tuning strategies**. We find that **they still cannot beat traditional ML models in clinical prediction yet.**
>
> > With the development of more advanced LLMs, the findings may change, as they only reflect current guidelines and cannot predict the future practical use of LLMs.
>
> First of all, **research has shown that the current general-purpose and medical LLMs could have almost clinician-level performance in medical licensing exams and clinical case challenges**. However, we discover that both general-purpose and medical LLMs, even with different model sizes, prompting or fine-tuning strategies, still cannot beat traditional ML models in clinical prediction yet, **shedding light on the surprising but critical deficiency of both general-purpose and medical LLMs in real-world clinical reasoning and decision-making**. We believe **`it is pressing to call for caution when practitioners adopt LLMs in practical clinical applications.`**
>
> Second, we have studied the impact of parameter scaling on LLMs' performance in clinical prediction. **We observe that the parameter scaling does not necessarily lead to better and could even hurt clinical prediction performance.**
>
> Third, **ClinicalBench could be adopted to facilitate the progress in enhancing LLMs’ clinical reasoning capacities** and minimize the gap between the development of LLMs for healthcare and clinical practice in the real world.
>
> > The employed LLMs are mainly small in size; larger models of 70B parameters and beyond should be explored.
>
> **We would like to clarify that we have thoroughly studied the impact of parameter scaling on LLMs' performance in clinical prediction in Table 2.** Actually, we have compared the performance of multiple model series: Llama3-70B and Llama3-8B; Qwen2-0.5B, Qwen2-1.5B and Qwen2-7B; Yi-v1.5-6B, Yi-v1.5-9B, Yi-v1.5-34B; Meditron-7B, Meditron-70B; Med42-8B, Med42-70B.

---

> ### Author Response · Authors · 2024-12-04
> **We sincerely appreciate the comment and provide clarifications as follows (2)**
>
> > These larger language models have much better prompt-following capabilities, so CoT and ICL approaches may work, rather than simply concluding that "The effectiveness of typical prompting engineering techniques is generally limited."
>
> Thanks for the suggestion. We have provided the results of Llama3-70B (general-purpose LLM) and Med42-70B (medical LLM) with different prompting strategies as follows. The complete results are in https://anonymous.4open.science/r/llm4clinical_prediction-D09D/src/results
> - For example, the results of Llama3-70B with CoT on MIMIC-III for length-of-stay prediction task are in https://anonymous.4open.science/r/llm4clinical_prediction-D09D/src/results/length_pred/mimic3/length_pred_result_data_Meta-Llama-3-70B-Instruct_6_COT.csv
> - The results of  Med42-70B with ICL on MIMIC-III for length-of-stay prediction task are in
> https://anonymous.4open.science/r/llm4clinical_prediction-D09D/src/results/length_pred/mimic3/length_pred_result_data_Llama3-Med42-70B_6_ICL.csv
>
> We can clearly see that the results further strengthen our **Finding 2: The effectiveness of typical prompting engineering techniques is generally limited and they still cannot make general-purpose and medical LLMs beat traditional ML models yet.**
>
> |     Llama3-70B     | Length (mimic3) | Length (mimic4) | Mortality (mimic3) | Mortality (mimic4) | Readmission (mimic3) | Readmission (mimic4) |
> |----------|------------------|------------------|---------------------|---------------------|-----------------------|-----------------------|
> | **ORI**  | 18.88           | 21.76           | 29.82              | 17.80              | 13.72                | 22.03                |
> | **ICL**  | 41.04           | 47.52           | 25.22              | 12.41              | 12.47                | 25.43                |
> | **RP**   | 11.68           | 11.37           | 23.34              | 11.68              | 13.83                | 24.95                |
> | **COT**  | 10.00           | 9.02            | 0.00               | 0.00               | 0.00                 | 0.00                 |
> | **SR**   | 25.70           | 26.28           | 21.09              | 7.22               | 13.81                | 25.13                |
>
> |     Med42-70B    | Length (mimic3) | Length (mimic4) | Mortality (mimic3) | Mortality (mimic4) | Readmission (mimic3) | Readmission (mimic4) |
> |----------|------------------|------------------|---------------------|---------------------|-----------------------|-----------------------|
> | **ORI**  | 27.23           | 34.52           | 46.15              | 34.48              | 9.64                 | 9.90                 |
> | **ICL**  | 55.40           | 50.79           | 27.40              | 11.26              | 8.56                 | 19.35                |
> | **RP**   | 46.46           | 56.06           | 14.08              | 34.48              | 0.00                 | 2.70                 |
> | **COT**  | 9.39            | 6.92            | 4.24               | 0.43               | 11.34                | 18.66                |
> | **SR**   | 17.72           | 13.51           | 0.00               | 0.00               | 0.00                 | 1.69                 |
>
> > Additionally, no variance is reported in the benchmarking table
>
> As stated in **"The numbers in bracket are the ranges of performance
> with 95% Confidence Interval (95% CI), which are calculated with the 5-run scores."** (line 214-215) We would like to clarify that we actually have included variance in Table 1.
>
> > although LLMs' outputs can be sensitive to input prompts, temperature settings, and the sequence of input instructions.
>
> Actually, **we have extensively studied the impact of decoding temperature on the performance of clinical prediction in Figure 3 in Section 3 and Figure 5 in Appendix B.3**.  We also have studied the effectiveness of typical prompting strategies in Table 3.

---

> ### Author Response · Authors · 2024-12-04
> **Dear Reviewer, if you have already found our responses satisfactory, we humbly remind you of a fitting update of the final rating. Thanks for your time and effort again!**
>
> Dear Reviewer H2Eh,
>
> We are genuinely grateful for your constructive feedback and the acknowledgement of our contributions. If you have already found our responses satisfactory, we humbly remind you of a fitting update of the final rating. Thanks for your time and effort again!
>
> The authors

---

### Official Review · Reviewer_6Gee · 2024-11-04

**Soundness:** 3
**Presentation:** 3
**Contribution:** 2
**Rating:** 3
**Confidence:** 2

**Summary:**

This paper presents ClinicalBench, comparing traditional ML models with LLMs on clinical prediction tasks. While the experimental setup is thorough, I have significant concerns about the scientific value and broader impact of this work given the dramatic performance gap demonstrated between LLMs and traditional ML models.

The results show that even state-of-the-art LLMs perform substantially worse than basic ML models like XGBoost and SVMs across all tasks. For example, traditional ML models achieve Macro F1 scores of 65-70% on Length-of-Stay prediction while LLMs struggle to exceed 30%. Similar large gaps exist for mortality and readmission prediction. Even with extensive prompt engineering and fine-tuning, LLMs fail to approach traditional ML performance.

Given these results, it's unclear what meaningful insights this work provides beyond demonstrating that LLMs are currently inappropriate for clinical prediction tasks. The paper essentially confirms what one might expect - that specialized ML models trained directly on structured clinical data outperform general language models on specific prediction tasks. While the thorough empirical validation may have some value, the magnitude of the performance gap suggests this comparison may be unnecessary.

**Strengths:**

Thorough experiments

**Weaknesses:**

The finding is obvious and negative. Unclear the value to the ML community.

**Questions:**

None

---

> ### Author Response · Authors · 2024-12-04
> **Thanks for the comment. We are more than happy to provide clarifications as follows.**
>
> Dear Reviewer 6Gee,
>
> We sincerely appreciate the comment and provide clarifications as follows:
>
> > The paper essentially confirms what one might expect - that specialized ML models trained directly on structured clinical data outperform general language models on specific prediction tasks.
>
> Thanks for the comment. **Actually we disagree that the results are expected.** We made the first attempt to comprehensively and systemically compare the clinical prediction capacities of LLMs and traditional ML models in a head-to-head way. We discover that both general-purpose and medical LLMs, **even with different model sizes, prompting or fine-tuning strategies, still cannot beat traditional ML models in clinical prediction yet**.
>
> > Given these results, it's unclear what meaningful insights this work provides beyond demonstrating that LLMs are currently inappropriate for clinical prediction tasks. ...
>
> Thanks for the comment. **The significance has been clearly stated in our contributions (line 97-101)**. Previous results have shown that general-purpose and medical LLMs could have almost clinician-level performance in medical licensing exams and clinical case challenges. **But our findings demonstrate the surprising but critical deficiency of both general-purpose and medical LLMs in real-world clinical reasoning and decision-making.** We call for caution when adopting LLMs in practical clinical applications. ClinicalBench could be leveraged to bridge the gap between the development of LLMs for healthcare and real-world clinical practice.

---

> ### Author Response · Authors · 2024-12-04
> **Dear Reviewer, if you have already found our responses satisfactory, we humbly remind you of a fitting update of the final rating. Thanks for your time and effort again!**
>
> Dear Reviewer 6Gee,
>
> We are genuinely grateful for your constructive feedback and the acknowledgement of our contributions. If you have already found our responses satisfactory, we humbly remind you of a fitting update of the final rating. Thanks for your time and effort again!
>
> The authors

---

### Meta-Review · Area_Chair_joqJ · 2024-12-21

**Metareview:**

The paper introduces ClinicalBench, comparing LLMs and traditional ML models in clinical prediction, revealing LLMs' deficiencies in reasoning and highlighting the need for caution in clinical AI adoption.

Strengths:
- The paper provides a head-to-head comparison of diverse LLMs with traditional ML models on real-world clinical prediction tasks.
- The reviewers agree that the paper is well presented in terms of its writing and structure

Weaknesses:
- The fact that traditional ML models outperform LLMs in structured prediction tasks is predictable and lack novelty, thus minimally contributing to advance the field.
- The benchmark relies on MIMIC only, creating a concern for data leakage and thus limited applicability of the findings to broader clinical scenarios.
- The paper does not fully explore the usage of LLMs in terms of in-context examples or experimenting with larger models with stronger reasoning capabilities.

**Additional Comments On Reviewer Discussion:**

During the discussio phase, the authors provided more details about fine-tuning and prompting strategies and additional epxeriment results with larger models. However, the core concern regarding the benchmark result being predictable, failure to explore more diverse options for in-context examples, and reliance on only MIMIC with existing tasks remained unresolved.

---

### Decision · Program_Chairs · 2025-01-22

Reject